# Kernel Regression in Structured Non-IID Settings: Theory and Implications for Denoising Score Learning

**Dechen Zhang**[1]   **Zhenmei Shi**[3]   **Yi Zhang**[1]   **Yingyu Liang**[1,2]   **Difan Zou**[1,2]

[1]Institute of Data Science, The University of Hong Kong
[2]School of Computing and Data Science, The University of Hong Kong
[3]Computer Sciences Department, University of Wisconsin-Madison
dechenzhang@connect.hku.hk, dzou@hku.hk

## Abstract

Kernel ridge regression (KRR) is a foundational tool in machine learning, with recent work emphasizing its connections to neural networks. However, existing theory primarily addresses the i.i.d. setting, while real-world data often exhibits structured dependencies - particularly in applications like denoising score learning where multiple noisy observations derive from shared underlying signals. We present the first systematic study of KRR generalization for non-i.i.d. data with signal-noise causal structure, where observations represent different noisy views of common signals. By developing a novel blockwise decomposition method that enables precise concentration analysis for dependent data, we derive excess risk bounds for KRR that explicitly depend on: (1) the kernel spectrum, (2) causal structure parameters, and (3) sampling mechanisms (including relative sample sizes for signals and noises). We further apply our results to denoising score learning, establishing generalization guarantees and providing principled guidance for sampling noisy data points. This work advances KRR theory while providing practical tools for analyzing dependent data in modern machine learning applications.

## 1  Introduction

Kernel ridge regression (KRR) occupies a central role in machine learning. Recently, driven by the insight that many deep neural networks (DNNs) can be viewed as converging to specific kernel regimes [31, 17], the research community has paid renewed attention directed toward the generalization behavior of KRR. A central question in KRR is to derive generalization guarantees with the regularization parameter $\lambda \geqslant 0$ under finite samples. In the special linear kernel case, Bartlett et al. [4] and Tsigler and Bartlett [68] established nearly tight upper and lower bounds on the excess risk for general $\lambda$. Their results demonstrate that non-vacuous generalization is achievable under specific conditions on the data covariance and global optimum. More recently, a series of works extended this analysis to nonlinear kernels, deriving the learning curve for KRR under power-law decay assumptions on the RKHS spectrum and mild assumptions on the target function[46, 37, 39, 11]. Their work shows that benign generalization occurs for a well-defined range of $\lambda$.

Despite these remarkable breakthroughs in characterizing the learning ability of KRR, a fundamental limitation persists: existing results are largely restricted to the i.i.d. setting. Specifically, they rely on the critical assumption that training samples are independently and identically distributed (i.i.d.) drawn from the underlying data distribution. However, in many real-world applications, collected data points often deviate from strict i.i.d. conditions due to inherent correlations introduced during data generation or collection processes. For example, consider data collected in a noisy environment, where each observation consists of an underlying signal $x$ corrupted by environmental noise $u$. When multiple samples are generated from the same signal $x$ but with different noise realizations $u_1, \ldots, u_k$,

39th Conference on Neural Information Processing Systems (NeurIPS 2025).

these samples become statistically dependent and thus the i.i.d. assumption will no longer hold. This dependency also occurs in denoising score matching [30, 69, 26], where multiple noisy versions of each clean data point are used to learn score functions, creating an inherently non-i.i.d. training set.

To the best of our knowledge, no prior work has systematically studied KRR with such causal-structured non-i.i.d. training samples (each i.i.d. signal is paired with $k$ i.i.d. noise). In particular, it remains an open question whether data dependencies benefit or hinder the generalization performance of KRR. The key technical barriers are two-fold: (1) the inapplicability of standard i.i.d. theory, and (2) the prevailing tendency in non-i.i.d. analysis to view data dependence unfavorably. This fundamental limitation poses significant challenges in establishing sharp theoretical guarantees for the causal-structured non-i.i.d. setting.

**Notations.** We use asymptotic notations $O(\cdot), o(\cdot), \Omega(\cdot)$ and $\Theta(\cdot)$, and use $\tilde{\Theta}(\cdot)$ to suppress logarithm terms. We also use the probability versions of the asymptotic notations such as $O_{\mathbb{P}}(\cdot)$. Moreover, following the notations in existing work [37, 39], we denote $a_n = O^{\mathrm{poly}}(b_n)$ if $a_n = O(n^p b_n)$ for any $p > 0$, $a_n = \Omega^{\mathrm{poly}}(b_n)$ if $a_n = \Omega(n^{-p}b_n)$ for any $p > 0$, $a_n = \Theta^{\mathrm{poly}}(b_n)$ if $a_n = O^{\mathrm{poly}}(b_n)$, and $a_n = \Omega^{\mathrm{poly}}(b_n)$; and we add a subscript $\mathbb{P}$ for their probability versions.

## 1.1 Our Main Results

In this paper, we initiate the generalization study of the KRR estimator $\hat{f}_\lambda$ for non-i.i.d. data. In particular, we consider the data model with a causal structure: $x \to g \leftarrow u$, where $g$ denotes the observed data point, and $x$ and $u$ denote the factors from the signal source $\mathcal{X}$ and noise source $\mathcal{U}$ respectively. Then, when generating the training samples, we first generate i.i.d. signals $x_1, \ldots, x_n$ from $\mathcal{X}$, then pair each $x_i$ with $k$ i.i.d. noise realizations $u_{i1}, \ldots, u_{ik}$ from $\mathcal{U}$, leading to $nk$ dependent observations $\{g_{ij}\}_{i=1,j=1}^{n,k}$ through the causal mechanism (see Section 3.1 for more details).

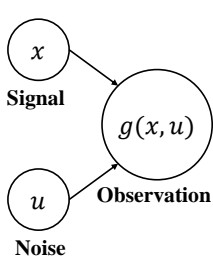

Figure 1: Causal structure of our data model.

Furthermore, as conventional concentration methods are ill-suited for the causal-structured data model, we introduce a novel methodology that systematically partitions correlated random sequences into independent blocks through iterative decomposition. Building on this approach, we establish a Bernstein-type concentration inequality for $k$-gap independent data (see definition in Section 3.1), which uncovers the benefit of dependency and nearly matches the rates of classical i.i.d. concentration bounds—up to logarithmic factors. Equipped with this technique, we characterize the excess risk of KRR in the structured non-i.i.d. setting, summarized as follows:

**Theorem 1.** (Informal statement of Theorem 4.1) *Under general assumptions, if the regularization parameter $\lambda = \Omega\left(n^{-\beta}\right)$, then the asymptotic rate (with respect to sample size $n$) of the generalization error (excess risk) $R(\lambda)$ is roughly*

$$R(\lambda) \leqslant \underbrace{\tilde{\Theta}_{\mathbb{P}}\left(\lambda^{\tilde{s}}\right)}_{\text{Bias}} + \underbrace{\tilde{\sigma}^2 O_{\mathbb{P}}^{\mathrm{poly}}\left(\lambda^{-\frac{1}{\beta}}\left(\frac{\tilde{r}}{n} + \frac{1-\tilde{r}}{nk}\right)\right)}_{\text{Variance}},$$

*where $\beta$ denotes the decay rate of the kernel eigenvalues, $\tilde{s}$ represents the smoothness of target function, $\tilde{\sigma}^2$ is the population noise level and $\tilde{r}$ quantifies the relevance of observations sampled from the same underlying signal but corrupted by different noise.*

Note that when $k = 1$, the setting reduces to the standard i.i.d. setting, recovering the prior results [37]. The theoretical result reveals the interplay between the data relevance $\tilde{r}$ and noise sample size $k$, which further implies the benefit of data relevance. In particular, while increasing noise samples enhances generalization, the improvement critically depends on the underlying signal relevance.

To further illustrate the theoretical result, we apply Theorem 1 to a single timestep of Denoising Diffusion Probabilistic Models (DDPM) [26], where the input data is generated by the weighted sum of the real-world observation and noise with weight $\sqrt{\alpha_t}$ and $\sqrt{1 - \alpha_t}$, i.e., $g_{ij} = \sqrt{\alpha_t}x_i + \sqrt{1 - \alpha_t}u_{ij}$. Under certain condition, our theoretical results show that the optimal value of the optimal noise multiplicity $k$ depends critically on the ratio $(1 - \alpha_t^{p/2})/\alpha_t^{p/2}$ where $p \in (0, 1]$ characterizes the Hölder-continuous property for kernel (see Assumption 1 and Theorem 4.4 for details). This

theoretical finding aligns well with the intuitive understanding that a larger $k$ is more beneficial when $\alpha_t$ is smaller—that is, when the noise component dominates.

Concretely, our contributions can be summarized as follows:

- We establish the first excess risk bound for KRR in structured non-i.i.d. setting (see Section 3.1 for details), characterizing the fundamental relationship between the data causal model and the sample sizes from different sources (signal and noise sources), which provides useful guidance for developing efficient data sampling strategies.

- We apply our framework to denoising diffusion probabilistic models (DDPMs) and derive the optimal noise sample size $k^*$ for each data point that minimizes the excess risk bound. Specifically, we show that the noise sampling schedule depends precisely on the time-varying noise-to-signal ratio $(1 - \alpha_t^{p/2})/\alpha_t^{p/2}$ at a each timestep $t$. This provides new insights for improving the training efficiency of diffusion models.

- We develop a novel Bernstein-type concentration inequality for $k$-gap independent data (see definition in Section 3.1) , which explicitly quantifies the benefit of data dependency. This general-purpose technique advances the theoretical toolkit for dependent data analysis and may find applications beyond our current setting, which is of independent interest to the community.

## 2 Related Works

**Theoretical Analysis of Kernel Regression.** Theoretical guarantees for the generalization property have attracted significant attention in machine learning. Seminal work by Bartlett et al. [4], Tsigler and Bartlett [68] derived nearly tight upper and lower excess risk bounds in linear (ridge) regression for general regularization schemes. Zou et al. [77, 76], Wu et al. [71] later extended this analysis to SGD and established sharp excess risk bound under substantially weaker assumptions on the spectrum of the data covariance. Their results demonstrate that benign overfitting is achievable under certain conditions on the data covariance and global optimum. For non-linear kernel, a large number of works [5, 56, 40, 38] studied the classical underparameterized (finite dimension) regime under specific polynomial decay kernel spectrum and smoothness of the ground-truth function. Specifically, Li et al. [40] proved the saturation effect that KRR fails to achieve the information theoretical lower bound when the smoothness of the underground truth function exceeds certain level. With regards to high-dimensional data, a line of work [42, 44, 48, 10] derived risk bounds by high-dimensional random matrix concentration for general kernel, while another line of research [22, 72, 50, 51, 46, 49, 11, 45] characterized the precise risk under specific conditions where the spectrum of kernel can be explicitly accessed. In particular, Mallinar et al. [46] and Medvedev et al. [49] demonstrated that the slow kernel eigenvalue decay and increasing dimensionality enable benign overfitting under Gaussian design assumption.

**Learning under Non-i.i.d. Data.** Standard i.i.d.-based concentration inequalities [1, 12, 13] fail to provide generalization guarantees for support vector machines (SVM) [62] or kernel methods under non-i.i.d. setting. To address this challenge, a line of work established the consistency under processes satisfying a law of large numbers [65], or satisfying empirical weak convergence [47]. However, the corresponding convergence rates typically remain unclear under such strong forms of non-i.i.d.-ness. Another line of research focused on the regression over trajectories generated by a dynamic system, including both linear cases [75] and non-linear [58] cases. However, the reliance on surrogate trajectory assumptions limits the applicability of these results to broader scenarios. A further body of literature examines learning under mixing conditions which characterize dependence via measures of correlation across time or sequence distance. Steinwart and Christmann [63], Hang and Steinwart [25] derived high-probability concentration bounds under geometric mixing, while Yu [73], Mohri and Rostamizadeh [53], Kuznetsov and Mohri [33] analyzed settings with algebraic mixing. Our $k$-gap independent case cannot be covered by these works, as the correlation between data points will remain high as long as they are from the same group with size $k$. Notably, although concentration results under general mixing framework (assuming the asymptotic mixing property) [53] could in principle accommodate our setting, our new results yield tighter bounds as we demonstrate the benefit of data relevance stands in contrast to a long line of work on learning from dependent data.

**Theoretical Analysis for Diffusion Model.** Recent theoretical advances in diffusion models have primarily addressed two fundamental aspects: (1) distribution estimation and (2) sampling guarantees.

For distribution estimation, seminal work by Song et al. [60] established the first statistical estimation bounds for diffusion models. Subsequent research by Chen et al. [7] demonstrated that when the target density lies on a low-dimensional manifold, the sample complexity scales only with the intrinsic dimension, thus avoiding the curse of dimensionality. Further studies have characterized the learning dynamics for specific data distributions, including Gaussian mixtures [61, 15, 57, 9, 21] and other structured distributions [36, 23, 24, 70]. On the sampling theory front, early convergence results required strong $\ell_\infty$ -accurate score estimates [16]. A significant advance by Lee et al. [34] established polynomial-time convergence under more practical $\ell_2$-accuracy assumptions, albeit requiring log-Sobolev inequalities. Later work relaxed these requirements to either bounded moment conditions [35, 8] or Lipschitz continuity of scores [8]. Recent developments have further improved computational efficiency through high-order discretization schemes [28, 27, 66] and exploitation of low-dimensional structures [29, 55, 41].

## 3 Theoretical Setup

### 3.1 Structured Non-i.i.d. Data

We consider the scenario that the same signal can differ owing to the existence of the random environment noise, leading to different but dependent observations. As shown in Figure 1, we formally define the data model as follows:

**Data model.** We consider the data model with two independent sources: signal source $\mathcal{X} \subset \mathbb{R}^d$ and noise source $\mathcal{U} \times \mathcal{Y} \subset \mathbb{R}^d \times \mathbb{R}^d$. Let $\mu_\mathcal{X}, \rho$ be a probability distribution on $\mathcal{X}, \mathcal{U} \times \mathcal{Y}$ respectively. The marginal distribution on $\mathcal{U}$ is denoted by $\mu_\mathcal{U}$. The data observation is formulated as a noisy realization of the signal, denoted as $g(x, u)$, where $g : \mathbb{R}^d \times \mathbb{R}^d \to \mathbb{R}^d$ is the realization function, $x \in \mathcal{X}$ and $u \in \mathcal{U}$ are independent signal and noise from their corresponding sources. Denote $\mathcal{G} := g(\mathcal{X}, \mathcal{U})$.

**Training data generation.** Following the causal structure in Figure 1, $n$ signals are first generated, then for each signal, we generate its $k$ noisy realizations via $k$ i.i.d. noise, yielding the training sample set $S = \{(g_{ij}, y_{ij})\}_{i,j=1}^{n,k}$ [1]. We call a sequence of random variables $(X_i)_{i \geqslant 1}$ is $k$-gap independent if any random variable $X_i$ is independent with $\sigma(X_{j \geqslant i+k}, X_{j \leqslant 1 \vee i-k})$. Obviously, $G := \{g_{ij}\}_{i,j=1}^{n,k}$ is $k$-gap independent. On the technical level, we develop concentration techniques under the general $k$-gap independence and apply our results on training samples $G$ (see Section 5.1 for details).

We assume an identical number of noisy realizations for all signals to simplify our analysis. However, our framework can be readily extended to accommodate varying numbers of realizations, though this would require somewhat more involved calculations. To further elucidate the data model and sampling methodology, we present two examples from real-world applications.

**Example 3.1. Signal processing in communication system.** The fundamental setting in signal processing is the communication system leveraging multiple transmissions [6]. Each source signal $x$ is transmitted $k$ times through a noisy channel where environmental disturbances $u_1, \ldots, u_k$ uniquely corrupt each transmission. This results in $k$ distinct received signals $g(x, u_1), g(x, u_2), \ldots, g(x, u_k)$ originating from the same source. More generally, for multiple source signals $x_1, x_2, \ldots, x_n$, the received signals are $\{g(x_i, u_{ij})\}_{i,j=1}^{n,k}$.

**Example 3.2. Denoising score learning.** In denoising score learning frameworks [30, 69, 26], a common strategy involves learning score functions using multiple noisy versions of clean data points. Specifically, for a single model at certain timestep $t$, each clean data points $x_i, i \in \{1, \ldots, n\}$ is perturbed with $k$ independent noises $u_{i1}, u_{i2}, \ldots, u_{ik}$. This perturbation follows a predefined function: $g(x, u) = \sqrt{\alpha_t} x + \sqrt{1 - \alpha_t} u$, yielding a noisy dataset $\{g(x_i, u_{ij})\}_{i,j=1}^{n,k}$.

### 3.2 Kernel Ridge Regression in Structural Non-i.i.d Setting

Let $k(\cdot, \cdot)$ be a continuous positive definite kernel over $\mathcal{G}$ and $\mathcal{H}$ be the separable reproducing kernel Hilbert space (RKHS) associated with $k(\cdot, \cdot)$. Denote the regularization parameter $\lambda \geqslant 0$, then the

---

[1]We consider the agnostic setting in this paper, i.e., we do not make any explicit assumption on the relationship between the data $g_{ij} = g(x_i, u_{ij})$ and its label $y_{ij}$.

kernel ridge regressor of each dimension can be represented as [2]

$$\hat{f}_\lambda^{(r)} = \arg\min_{f^{(r)} \in \mathcal{H}} \left( \frac{1}{nk} \sum_{i=1}^{n} \sum_{j=1}^{k} (y_{ij}^{(r)} - f^{(r)}(g_{ij}))^2 + \lambda \|f^{(r)}\|_{\mathcal{H}}^2 \right), \ r = 1, 2, \ldots, d.$$

Denote $f_\rho^{*(r)}, r = 1, ..., d$ as the population optimal solution, then the excess risk of $\hat{f}_\lambda$ is:

$$R(\lambda) = \sum_{r=1}^{d} \left\| \hat{f}_\lambda^{(r)} - f_\rho^{*(r)} \right\|_{L^2(\mathcal{G}, d\mu_\mathcal{G})}^2 = \sum_{r=1}^{d} \int \left( \hat{f}_\lambda^{(r)}(g) - f_\rho^{*(r)}(g) \right)^2 d\mu_\mathcal{G}(g),$$

where $\mu_\mathcal{G}$ is the probability measure on $\mathcal{G}$. By the optimality of $f_\rho^{*(r)}(g)$, it holds

$$\mathbb{E}\left[ \left( y^{(r)} - f_\rho^{*(r)}(g) \right) e_i(g) \right] = 0, \ i = 1, 2, \ldots; r = 1, \ldots, d. \tag{3.1}$$

To bound the excess risk, we further introduce the widely-used integral operator and the embedding index of RKHS [37, 39, 5, 43, 20]. Let $\mathcal{G} \subset \mathbb{R}^d$ be compact and $k(\cdot, \cdot)$ is continuous, we assume $k(\cdot, \cdot)$ is bounded [37]. Then the natural embedding $S_\mu : \mathcal{H} \to L^2$ is a Hilbert-Schmidt operator. Let $S_\mu^* : L^2 \to \mathcal{H}$ be the adjoint operator of $S_\mu$ and $T := S_\mu S_\mu^* : L^2 \to L^2$. Then, it is easy to show that $T$ is an integral operator given by $T(f) = \int_\mathcal{G} k(g, \cdot) f(g) d\mu_\mathcal{G}(g)$. By the spectral theorem of compact self-adjoint operators and the Mercer's theorem [64]:

$$T(f) = \sum_i \lambda_i \langle f, e_i \rangle_{L^2} e_i, \quad k(x, y) = \sum_i \lambda_i e_i(x) e_i(y),$$

where $\{\lambda_i\}_{i \geqslant 1}$ is the set of positive eigenvalues of the kernel in descending order and $\{e_i\}_{i \geqslant 1}$ is the corresponding eigenfunction, which forms an orthonormal basis of $\overline{\mathrm{Ran}\, S_\mu} \subset L^2$.

Besides, for $s \geqslant 0$, we define $T^s : L^2 \to L^2$ with $T^s(f) = \sum_i \lambda_i^s \langle f, e_i \rangle_{L^2} e_i$. Correspondingly, define the interpolation space [37]

$$[\mathcal{H}]^s = \mathrm{Ran}\, T^{s/2} = \left\{ \sum_{i \in N} a_i \lambda_i^{s/2} e_i \,\Big|\, \sum_{i \in N} a_i^2 < \infty \right\} \subseteq L^2,$$

with the norm $\left\| \sum_i a_i \lambda_i^{\frac{s}{2}} e_i \right\|_{[\mathcal{H}]^s} = \left( \sum_i a_i^2 \right)^{\frac{1}{2}}$. It is easy to verify that $[\mathcal{H}]^s$ is a Hilbert space with an orthonormal basis $\{\lambda_i^{s/2} e_i\}_{i \geqslant 1}$. Further, we define the embedding index $\alpha_0$ of $\mathcal{H}$, which characterizes the embedding property whether $[\mathcal{H}]^\alpha$ can be continuously embedded into $L^\infty(\mathcal{G}, \mu_\mathcal{G})$:

$$\alpha_0 = \inf \left\{ \alpha : \|[\mathcal{H}]^\alpha \hookrightarrow L^\infty(\mathcal{G}, \mu_\mathcal{G})\| := \mathrm{ess\, sup}_{g \in \mathcal{G}, \mu_\mathcal{G}} \sum_{i \in N} \lambda_i^\alpha e_i(g)^2 = M_\alpha < \infty \right\},$$

where ess sup is the essential supremum. For theoretical simplicity, we denote $\forall g \in \mathcal{G}$:

$$T_g f := \sum_i \lambda_i e_i(g) f(g) e_i, \quad T_G := \sum_{i=1}^{n} \sum_{j=1}^{k} T_{g_{ij}}, \quad T_\lambda = T + \lambda, \quad T_{G\lambda} = T_G + \lambda.$$

### 3.3 Assumptions and Definitions

**Assumption 1.** *We make the following assumptions on the data distribution and kernel function:*

- *Polynomial eigenvalue decay. There is some $\beta$ and constants $c_\beta, C_\beta$ such that*

$$c_\beta i^{-\beta} \leqslant \lambda_i \leqslant C_\beta i^{-\beta}, i = 1, 2, \ldots.$$

- *Relative smoothness of the regression function. For any $r = 1, 2, \ldots, d$, there are some $s > 1$ and a sequence $\left( a_i^{(r)} \right)_{i \geqslant 1}$ such that*

$$f_\rho^{*(r)} = \sum_{i=1}^{\infty} a_i^{(r)} \lambda_i^{\frac{s}{2}} i^{-\frac{1}{2}} e_i, \ 0 < c < |a_i^{(r)}| < C \text{ for some constants } c, C.$$

---

[2]This setting handles real-world vector-output tasks like denoising score learning where noise is assumed independent per dimension.

- *Sub-Gaussian noise. For each $r = 1, 2, \ldots, d$, noise $\epsilon^{(r)} := y^{(r)} - f_\rho^{*(r)}(g)$ is $\sigma_\epsilon^2$ sub-Gaussian conditionally on $g$, the second moment of $\epsilon^{(r)}$ conditionally on $g$ are bounded by $\sigma^2$:*

$$\left\| \epsilon^{(r)} | g \right\|_{\psi_2} \leqslant \sigma_\epsilon, \quad \mathbb{E}[\epsilon^{(r)^2} | g] \leqslant \sigma^2, \quad g, g' \in \mathcal{G} \text{ almost everywhere.}$$

  *For each $r = 1, \ldots, d$ and observation $g_{ij}$,*

$$\left\| \epsilon_{ij}^{(r)} | g_{ij}, g_{ij'} \right\|_{\psi_2} \leqslant \sigma_{\epsilon_{1,2}}, \quad \left\| \epsilon_{ij}^{(r)} | g_{i1}, \ldots, g_{ik} \right\|_{\psi_2} \leqslant \sigma_k, \quad \mathbb{E}\left[ \epsilon_{ij}^{(r)^2} | g_{ij}, g_{ij'} \right] \leqslant \sigma_G^2.$$

- *Hölder-continuous kernel. The kernel $k(\cdot, \cdot)$ is Hölder-continuous with index $p$, that is, there exist some $p \in (0, 1]$ and $L > 0$ such that*

$$|k(x_1, y_1) - k(x_2, y_2)| \leqslant L \left\| (x_1, y_1) - (x_2, y_2) \right\|_{\mathbb{R}^{d \times d}}^p, \ \forall x_1, y_1, x_2, y_2 \in \mathcal{G}.$$

These assumptions are largely consistent with existing work [37, 39], making our bound clearer and facilitating direct comparison with established results in the i.i.d. setting. The polynomial eigenvalue decay is satisfied by well-known kernels such as the Sobolev kernel [20], Laplace kernel, and neural tangent kernels for fully-connected multilayer neural networks. Notably, our framework is readily extensible to general spectra from a technical standpoint and the polynomial decay is assumed for theoretical simplicity (see Section F.1 for details).

The relative smoothness on $f_\rho^{*(r)}$ are also widely used [37, 39, 14, 32], showing that $f_\rho^{*(r)} \in [\mathcal{H}]^t$ for any $t < s$. In fact, our general theoretical bound still holds true under the relaxation from $s > 1$ to $s > 0$. The assumption $s > 1$ is used to estimate the relevance parameter for providing a concise bound and clear insights (see Section F.2 for details). Under this assumption,

The assumption on noise are widely used in Li et al. [37, 39], Bartlett et al. [4], Tsigler and Bartlett [68], Cheng et al. [11], all with respect to a single data point. For technical reasons to handle multiple signal realizations, we extend this assumption to hold conditionally on dependent data points.

Following Li et al. [37, 39], we assume the Hölder continuity with index $p$ to establish uniform concentration bounds via covering number estimates (see Section F.3 for details). This implies that $f_\rho^{*(r)} \in \mathcal{H}$ is Hölder-continuous with index $\frac{p}{2}$ for $r = 1, \ldots, d$ [19, 18]. Hence, there exists $L_\epsilon > 0$, such that

$$\left| \epsilon_{ij}^{(r)} - \epsilon_{i'j}^{(r)} \right| = \left| f_\rho^{*(r)}(g_{ij}) - f_\rho^{*(r)}(g_{i'j}) \right| \leqslant L_\epsilon \left\| g_{ij}^{(r)} - g_{i'j}^{(r)} \right\|^{\frac{p}{2}}, \ r = 1, \ldots, d,$$

where we construct $g_{i'j} := g(x_i', u_{ij}), \epsilon_{i'j}^{(r)} := y_{ij} - f_\rho^{*(r)}(g_{i'j})$ with $x_i'$ independent of $x_i$. We further assume that $\alpha_0 = \frac{1}{\beta}$, which is made in prior works [37, 39] and holds for numerous RKHSs. Examples include Sobolev RKHSs, those associated with periodic translation-invariant kernels, and those corresponding to dot-product kernels on spheres [37, 39, 74].

For detailed analysis in a structured non-i.i.d. setting, we summarize some key definitions characterizing the data dependency structure and the population noise level. To be specific, we extend the concept of population noise level $\sigma^2$ to structured non-i.i.d. settings by introducing the variance bound $\sigma_G^2$ conditioned on dependent data pairs. For technical reasons, we also take the smoothness of noise into account.

**Definition 3.1.** *Define the population noise level $\tilde{\sigma}^2 = L_\epsilon^2 \vee \sigma^2 \vee \sigma_G^2$.*

The population noise level captures the strength of both noise and its smoothness.

**Definition 3.2.** *(Data relevance) Denote $g_{i'j} = g(x_i', u_{ij}), \epsilon_{i'j}^{(r)} = y_{ij} - f_\rho^{*(r)}(g_{i'j})$ with $x_i'$ is independent of $x_i$. We respectively define the relevance of data, the relevance over the eigenfunction and the relevance under the integral operation :*

$$r_0 := \left( \frac{1}{2} - \frac{\sum_{r=1}^d \mathrm{Cov}\left( g_{ij}^{(r)}, g_{i'j}^{(r)} \right)}{2 \sum_{r=1}^d \mathrm{Var}\left( g_{ij}^{(r)} \right)} \right)^{\frac{p}{2}}, \ r_e := \frac{1}{2} - \frac{1}{2} \sup_r \mathbb{E}[e_r(g_{ij}) e_r(g_{i'j})],$$

$$r_T := \mathrm{ess} \sup_{g \in \mathcal{G}} \left| \frac{\mathbb{E} T_\lambda^{-1} k(g_{ij_1}, g) \epsilon_{ij_1}^{(r)} T_\lambda^{-1} k(g_{ij_2}, g) \epsilon_{ij_2}^{(r)}}{\left\| T_\lambda^{-1} k(g, \cdot) \right\|_{L^2}^2 \tilde{\sigma}^2} \right|,$$

*where the expectation is over $g_{ij}, g_{i'j}, g_{ij_1}, g_{ij_2}, \epsilon_{ij_1}, \epsilon_{ij_2}$.*

All these parameters $r_0, r_e, r_T \in [0,1]$ characterize how the signal source $x$ contribute to the observation $g(x, u)$. To be precise, $r_0$ describes the correlation between $g(x, u_1)$ and $g(x, u_2)$ (with independent $u_1$ and $u_2$), while $r_e$ and $r_T$ capture this correlation in the context of the eigenfunction $e_i$ and the integral operator $T_\lambda^{-1}$, respectively.

**Definition 3.3.** (Conditional orthogonality) *The conditional orthogonality holds for $r \neq s$ if*

$$\delta_{rs} := \mathbb{E}_u \left[ \mathbb{E}_x e_r(g) \mathbb{E}_x e_s(g) \right] = 0.$$

*We call an orthogonal basis $e_i(\cdot)$ that satisfies the conditional orthogonality if $\delta_{rs} = 0, \ \forall r \neq s$.*

There are many cases where the conditional orthogonality holds, which is discussed in Section E. Generally, Definition 3.2 and 3.3 capture the data dependency in a structured non-i.i.d. setting, which determines the impact of the noise sample size (see Section 4 for details).

## 4  Main Results

In this section, we will deliver the excess risk bound of the KRR estimator in our structured non-i.i.d. setting. In order to better explain the result, we first present the following bias-variance decomposition for the excess risk, which is commonly adopted in many recent works [4, 68, 59, 46, 49, 11, 37, 39] (see details in Section A.1). Denote

$$\text{Bias}^2(\lambda) = \sum_{r=1}^{d} \left\| T_{G\lambda}^{-1} T_G f_\rho^{*(r)} - f_\rho^{*(r)} \right\|_{L^2}^2, \text{Var}(\lambda) = \sum_{r=1}^{d} \left\| \frac{1}{nk} \sum_{i=1}^{n} \sum_{j=1}^{k} T_{G\lambda}^{-1} k(g_{ij}, \cdot) \epsilon_{ij}^{(r)} \right\|_{L^2}^2,$$

then

$$R(\lambda) \leqslant 2\text{Bias}^2(\lambda) + 2\text{Var}(\lambda).$$

We present our main theorem as follow.

**Theorem 4.1.** *Under Assumption 1, if $\lambda \asymp n^{-\theta}$, $\theta \in (0, \beta)$,*

$$R(\lambda) \leqslant \underbrace{\tilde{\Theta}_{\mathbb{P}} \left( n^{-\min(s,2)\theta} \right)}_{\text{Bias}^2(\lambda)} + \underbrace{\tilde{\sigma}^2 O_{\mathbb{P}}^{\text{poly}} \left( n^{\alpha_0 \theta} \left( \frac{r_T}{n} + \frac{1 - r_T}{nk} \right) \right)}_{\text{Var}(\lambda)}. \tag{4.1}$$

*Further, if the conditional orthogonality holds,*

$$R(\lambda) \leqslant \underbrace{\tilde{\Theta}_{\mathbb{P}} \left( n^{-\min(s,2)\theta} \right)}_{\text{Bias}^2(\lambda)} + \underbrace{\tilde{\sigma}^2 O_{\mathbb{P}}^{\text{poly}} \left( n^{\alpha_0 \theta} \left( \frac{r_0 \vee r_e}{n} + \frac{(1 - r_0) \wedge (1 - r_e)}{nk} \right) \right)}_{\text{Var}(\lambda)}. \tag{4.2}$$

**Remark 4.2.** Two novel concepts are introduced in our excess risk upper bound, the conditional orthogonality condition (Definition 3.3), which captures the dependency in structured non-i.i.d. setting and holds in many cases (Section E), and the parameters $r_0, r_e, r_T$, which characterize the data correlation between $g(x, u_1)$ and $g(x, u_2)$ under different conditions (Definition 3.2).

Overall, the non-vacuous generalization is attainable when $\theta \in (0, \beta)$, in agreement with the asymptotic results of Li et al. [37]. Denote the correlation level $\tilde{r} = r_e \vee r_0$. A key theoretical insight emerges: our bound explicitly blends $\frac{1}{n}$ and $\frac{1}{nk}$, weighted by $\tilde{r}$ and $1 - \tilde{r}$. Consequently, this result reveals a critical trade-off between relevance and noise sample size: when the correlation level $\tilde{r}$ is large, i.e., the signal dominates in the observed noisy data, increasing $k$ offers little benefits while increasing $k$ helps generalization when the noise component prevails.

In the regime $\theta \in [\beta, \infty)$, a theoretical lower bound in the i.i.d. setting is provided by some monotonicity properties with respect to $\lambda$ [37, 39], implying the generalization is vacuous in this case that $\lambda$ is small. For the reason that the correlation of data might not be positive, we can merely derive a lower bound for the variance term by taking conditional expectations over $\epsilon_{ij} | g_{ij}$ for $\text{Var}(\lambda)$.

**Theorem 4.3.** *Under Assumption 1, if $\lambda \asymp n^{-\theta}$, $\theta \in [\beta, \infty)$,*

$$\text{Bias}^2(\lambda) \leqslant O_{\mathbb{P}}^{poly} \left( n^{-\min(s,2)\beta} \right), \ \mathbb{E}_{\epsilon_{1,1}|g_{1,1}} \mathbb{E}_{\epsilon_{1,2}|g_{1,2}} \dots \mathbb{E}_{\epsilon_{n,k}|g_{n,k}} \left[ \text{Var}(\lambda) \right] \geqslant \Omega_{\mathbb{P}}^{\text{poly}} \left( \frac{\sigma_L^2}{k} \right),$$

*where $\sigma_L^2$ is the lower bound of $\mathbb{E}[\epsilon^{(r)2}|g]$ for $g \in \mathcal{G}$ almost everywhere.*

This lower bound for the variance term in case $\theta \in [\beta, \infty)$ demonstrates that the generalization will never be benign, aligning well with the prior results [37, 39].

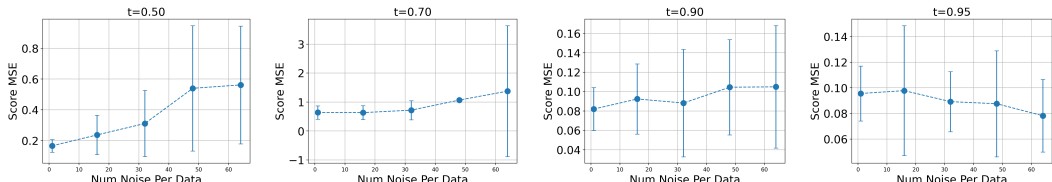

Figure 2: Score estimation error (mean $\pm$ s.d.) versus *the number of noise per data*, i.e., $k$, for four noise levels, where lower error implies better score learning.

## 4.1 Implication to Denoising Score Learning

At a single timestep $t$ in denoising score learning, the goal is to minimize the loss given the training set $S = \{(g_t(x_i, \xi_{ij}), \xi_{ij})\}_{i=1,j=1}^{n,k}$ where $g_t(x, \xi) = \sqrt{\alpha_t}x + \sqrt{1-\alpha_t}\xi$ applies data on noise.

$$\mathcal{L} := \frac{1}{nk} \sum_{i=1}^{n} \sum_{j=1}^{k} \|\xi_{ij} - f_\theta(g_t(x_i, \xi_{ij}))\|^2.$$

Since our structured non-i.i.d. framework does not make any assumptions on the relationship between the noisy data and its label, denoising score learning naturally fits our agnostic setting. Under the same notations in Assumption 1, we apply Theorem 4.1 to denoising score learning as follow.

**Theorem 4.4.** *Consider the denoising score learning at timestep $t$. Assuming that $\mathbb{E}[x^2] \leqslant \sigma_x^2$, $\mathbb{E}[\xi^2] \leqslant \sigma_\xi^2$, if $\lambda \asymp n^{-\theta}, \theta \in (0, \beta)$ and the conditional orthogonality holds, under Assumption 1 [3],*

$$R(\lambda) \leqslant \tilde{\Theta}_{\mathbb{P}}\left(n^{-\min(s,2)\theta}\right) + \tilde{\sigma}^2 O_{\mathbb{P}}^{\mathrm{poly}}\left(n^{\alpha_0\theta}\left(\frac{\alpha_t^{\frac{p}{2}} \vee r_e}{n} + \frac{(1-\alpha_t^{\frac{p}{2}}) \wedge (1-r_e)}{nk}\right)\right).$$

Theorem 4.4 can be easily derived by computing $r_0$ given $g(x, u) = \sqrt{\alpha_t}x + \sqrt{1-\alpha_t}u$. In practical denoising score learning, the main challenges arise from the underlying data distribution, the properties of the true score function $f_\rho^*$, and the spectral decay of the chosen kernel. Our theory provides a general framework to characterize the learnability of different data distributions. Practitioners can leverage this framework as follows: first, select a kernel appropriate to the problem domain; second, check the decay rate of the kernel's spectrum; and finally, apply Theorem 4.4 to rigorously determine (i) whether the distribution can be learned efficiently and (ii) the sample complexity required for convergence.

Furthermore, building on the theoretical trade-off that increasing $k$ helps generalization when the noise component prevails while increasing $k$ is useless when the signal dominates, a key inspiration for empirical study emerges: for a fixed batch size, if signal dominates, then setting $k = 1$ is enough; while when noise dominates, one is encouraged to increase $k$ up to roughly $(1 - \alpha_t)/\alpha_t$, or more precisely, $(1 - \alpha_t^{p/2})/\alpha_t^{p/2}$. This adaptive design for noise multiplicity $k$ may advance the empirical study for denoising score learning.

**Numerical Experiments.** We train a three-layer ReLU MLP (100 neurons each) to learn the score of a two-component MoG ($\mu = [-5, 5]$), $\sigma = 0.2$) at four noise levels $t \in \{0.50, 0.70, 0.90, 0.95\}$ via denoising score matching loss, where the networks are trained separately. Each network is optimized with SGD (lr $= 10^{-1}$, momentum $= 0.9$) and a 0.9 EMA. In each iteration, we consider a **fixed batch size** $nk = 128$, with a varying number of noises paired with each data, i.e., $k$, from 0 to 64. The results are displayed in Figure 2 over 100 independent runs. More experiments on real image diffusion training and experiments using kernel ridge regressor rather than neural network are detailed in Section G.1 and Section G.2.

From the experimental results, we demonstrate an important relationship between the noise level $t$ and the optimal noise-sample ratio $k$. For lower noise levels ($t = 0.5, 0.7$), we find that pairing each data point with a single noise sample ($k = 1$) yields optimal score learning performance.

---

[3]We present a general theoretical framework on denoising score learning under mild Assumption 1 here, while deferring derivations for specific data distribution and kernel to future work.

Conversely, at higher noise levels ($t = 0.9, 0.95$), better results are achieved by increasing $k$. These empirical findings directly support our theoretical analysis in Theorem 4.4, which shows that the optimal $k$ should scale with the noise level $t$ (or equivalently, inversely with $\alpha_t$). The results provide practical insights for optimizing the training efficiency of diffusion models, suggesting that adaptive noise-sample pairing strategies may offer significant computational benefits.

## 5  Proof Details

In this section, we outline the proof and present our key techniques, focusing particularly on the novel blockwise decomposition method developed to establish a Bernstein-type high-probability bound.

**Proof roadmap for Theorem 4.1.** For $\mathrm{Bias}(\lambda)$, standard concentration techniques—which rely heavily on the i.i.d. assumption—face significant challenges when applied to dependent data. Motivated by Banna et al. [3], we develop a novel blockwise decomposition method for $k$-gap independent random sequence and derive the Bernstein-type high probability bound tailed to structured non-i.i.d. bounded data. Overall, we adopt two-step concentration analysis using our developed technique to characterize $\mathrm{Bias}(\lambda)$. For $\mathrm{Var}(\lambda)$, instead of conditioning on $g$ to take expectations over the noise $\epsilon$ [4, 68, 46, 37, 39, 11], we perform direct concentration analysis on $\epsilon$, a necessity due to inherent data dependencies that invalidate standard conditional expectation techniques. Overall, we characterize $\mathrm{Var}(\lambda)$ by adopting three-step concentration analysis, where the concentration arguments in structured non-i.i.d. setting is similar as the analysis for $\mathrm{Bias}(\lambda)$.

### 5.1  Key Proof Techniques

As outlined in the proof roadmap, classical concentration inequalities crucially depend on the i.i.d. assumption, limiting their applicability to dependent data regimes. This dependence invalidates foundational steps in traditional concentration proofs, such as the decomposition of moment-generating functions (MGFs), where the equality $\mathbb{E}\left[e^{\sum_i X_i}\right] = \prod_i \mathbb{E}[e^{X_i}]$ fails to hold under non-i.i.d. conditions. We present our novel techniques in the following Lemma 5.1, which is stated under mild $k$-gap independent condition [4]. With this novel result Lemma 5.1, we can perform refined concentration inequalities in structured non-i.i.d. setting.

**Lemma 5.1.** *Consider a $k$-gap sequence of random variables $(\mathbb{X}_i)_{i=1}^{nk}$ taking values of self-adjoint Hilbert-Schmidt operators. Suppose that there exists a positive constant $M$ such that for any $i \geqslant 1$,*

$$\mathbb{E}(\mathbb{X}_i) = \mathbf{0} \quad and \quad \lambda_{\max}(\mathbb{X}_i) \leqslant M \quad almost \ surely.$$

*Denote* $v^2 = \sup\limits_{K \subseteq \{1,\ldots,nk\}} \frac{1}{\mathrm{Card}K} \lambda_{\max}\left(\mathbb{E}\left[\left(\sum_{i \in K} \mathbb{X}_i\right)^2\right]\right)$, $\mathrm{intd} = \mathrm{intdim}\left(\mathbb{E}\mathbb{X}^2\right)$, *where* $\mathrm{intdim}(A) = \frac{\mathrm{tr}(A)}{\|A\|}$ *is the intrinsic dimension of $A$. For any positive $t$ such that $tM < \frac{1}{k}\frac{1}{\log n}$,*

$$\log \mathbb{E}\mathrm{tr}\left(\exp\left(t\sum_{i=1}^{nk} \mathbb{X}_i\right) - \mathbf{I}\right) \leqslant \log n \log 3 + \log\left(\frac{nk}{2}\mathrm{intd}\right) + t^2 nkv^2 \frac{169}{1 - tMk\log n}.$$

**Key proof insight of Lemma 5.1.** Lemma 5.1 is proved by iteratively partitioning the random sequence into mutually independent blocks and incorporating the separated bounds. On high level, we develop a different block scheme compared with current mixing techniques, enabling us to capture the underlying data independence and fully utilize the intra-block randomness. This refined study helps discover some benefits of data dependency.

**Step 1: Partition and derive the separated bound.** We firstly partition $A_0 := \{1, \ldots, nk\}$ into three fragments, delete the middle fragment to guarantee the remaining two fragments are independent; secondly partition each of the two remaining terms into three fragments, delete the middle fragment to guarantee the remaining two fragments are independent. After repeating this procedure $\ell$ times, we denote the remaining terms as $K_{A_0}$. From the partition, $\sum_{i \in K_{A_0}} \mathbb{X}_i$ can be represented as the sum

---

[4]Our novel concentration bound can extend to the case where the number of noisy realizations $k_i$ varies per signal $x_i$, by replacing $k$ with $k_{\max} = \max_i k_i$. Our result can also generalize to the case where block gaps are only approximately independent, e.g., some weakly dependent process assuming specific mixing property, by quantifying block dependence [52, 3].

of $2^\ell$ mutually independent random fragments. Consequently, we derive the separated bound (see details in Proposition D.1).

**Step 2: Incorporate the separated bounds.** After obtaining $K_{A_0}$, we can also undertake the same partition for the remaining elements $\{i_1, \ldots, i_{A_1}\} = \{1, \ldots, A\} \backslash K_A$. Repeating $L \leqslant O(\log n)$ times, the sum $\sum_{i=1}^{nk} \mathbb{X}_i$ can be represented as the sum of $L + 1$ fragments which can be bounded by the analysis in Step 1. Finally, we incorporate the separated bound by a simple incorporating lemma and prove the Lemma 5.1 (see details in Proposition D.2).

Consider the case $d = 1$, Lemma 5.1 can be simplified as: There exists a constant $C''$ such that for any positive $t$ such that $tM < \frac{1}{k \log n}$,

$$
\log \mathbb{E} \exp \left( t \sum_{i=1}^{nk} \mathbb{X}_i \right) \leqslant \frac{C'' t^2 nk v^2}{1 - tMk \log n}. \tag{5.1}
$$

For simplicity, we discuss our technique via the case $d = 1$. In particular, the novel concentration inequality can be obtained by (5.1):

$$
\mathbb{P} \left[ \left| \sum_{i=1}^{nk} \mathbb{X}_i \right| \geqslant \epsilon \right] \leqslant 2 \exp \left( -\frac{\epsilon^2 / 2}{2C'' nk v^2 + \epsilon \cdot Mk \log n} \right). \tag{5.2}
$$

**Comparison to concentration bound under general mixing assumption.** For simplicity, we consider $M$ is on constant level. Under general mixing assumption $\lim_{i \to \infty} \phi(i) = 0$ where $\phi(\cdot)$ is the mixing coefficient for zero-mean random variables $W_1, \ldots, W_{nk}, \ldots$, previous concentration techniques treat the data dependency as a bad effect, yielding the following concentration inequality (Theorem 8 in [53]) $\left| \frac{1}{nk} \sum_{i=1}^{nk} W_i \right| \leqslant \tilde{O}_{\mathbb{P}} \left( \frac{1 + \sum_{i=1}^{nk} \phi(i)}{\sqrt{nk}} \right) = \tilde{O}_{\mathbb{P}} \left( \sqrt{\frac{k}{n}} \right)$ [5]. In contrast, our result (5.2) implies $\left| \frac{1}{nk} \sum_{i=1}^{nk} \mathbb{X}_i \right| \leqslant \tilde{O}_{\mathbb{P}} \left( \sqrt{\frac{v^2}{nk}} \right) = \tilde{O}_{\mathbb{P}} \left( \sqrt{\frac{kr+1-r}{nk}} \right)$ [6], which decreases as $k$ increases. The intuition is that we do not treat the data dependency a bad effect but aim to discover some benefits for reducing the error, otherwise it would be intractable to prove the vanishing generalization error in our setting for general $k$.

# 6 Conclusion and Limitations

In this work, we provide a refined analysis on the excess risk of kernel ridge regression in structured non-i.i.d. setting, by deriving a novel Bernstein-type concentration inequality for $k$-block independent data. Our theoretical upper bound of excess risk demonstrates that when the noise dominates in the observed noisy data, increasing $k$ helps generalization. In practical denoising score learning, empirical findings directly support our theoretical insight and further inspire adaptive noise-sample pairing strategies for optimizing the training efficiency of diffusion models.

Our limitations are twofold: (i) we only establish the learnability of individual score function but no fully characterization of the sampling error throughout the entire denoising process, and (ii) our analysis is confined to structured non-i.i.d. settings, leaving a rigorous theoretical characterization of general non-i.i.d. scenarios as an open problem.

# Acknowledgements

We would like to thank the anonymous reviewers and area chairs for their helpful comments. This work is supported by NSFC 62306252, Hong Kong ECS award 27309624, Guangdong NSF 2024A1515012444, and the central fund from HKU IDS.

---

[5]Here we invoke the $k$-gap condition, which implies that $\phi(i) = 0$ for all $i \geqslant k$, and we note that $|\phi(i)|$ is bounded by a constant for all $i$ [2].

[6]Here $r \in [0, 1]$ denotes the relevance between two data points with sequence distance smaller than $k$.

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

# Appendix

We provide detailed proofs for Theorem 4.1, Theorem 4.3 and Theorem 4.4 in Section A. In particular, we develop novel Bernstein-type concentration techniques in Section D and apply this to establish concentration lemmas in Section B. These concentration lemmas are essential for the derivation of our theory. In Section E, we provide two specific examples to illustrate when the conditional orthogonality holds. We further provide comprehensive discussion on our assumptions in Section F. To supplement our numerical experiments, we perform experiments on image diffusion and kernel ridge regressor in Section G.

The following proof dependency graph visually encapsulates the logical structure and organizational architecture of the theoretical results in our paper. This graph serves as a map for navigating the paper's proofs, allowing readers to quickly grasp the global structure, identify core technical components, and understand the interrelationships that underpin our main findings. In particular, the arrow from element $X$ to element $Y$ means the proof of $Y$ relies on $X$.

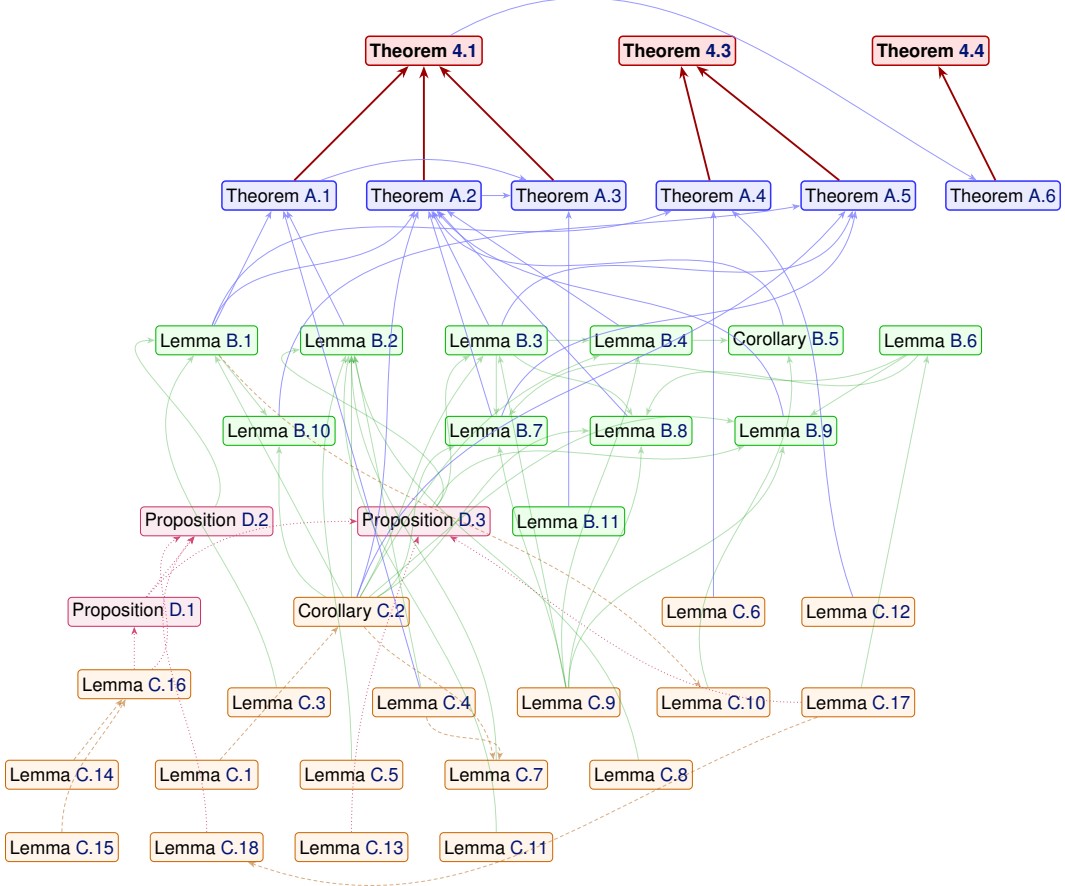

# Appendix Contents

# A Detailed Proofs

In this section, we present detailed proofs for our main results. To be specific, we prove Theorem 4.1 in Section A.2, Theorem 4.3 in Section A.3 and Theorem 4.4 in Section A.4. For simplicity, we denote $\tilde{s} = \min(s, 2)$. Before presenting the detailed proofs, we firstly perform bias-variance decomposition.

## A.1 Bias-Variance Decomposition

We first undertake bias-variance decomposition, which is commonly used in analyzing excess risk [4, 68, 59, 46, 49, 11, 37, 39]. By the definition of the integral operator, we express the kernel ridge regressor as

$$\hat{f}_\lambda^{(r)} = (T_G + \lambda)^{-1} \frac{1}{nk} \sum_{i=1}^n \sum_{j=1}^k k(g_{ij}, \cdot) y_{ij}^{(r)}, \ r = 1, 2, \ldots, d.$$

We denote the conditional kernel ridge regressor

$$\tilde{f}_\lambda^{(r)} := \mathbb{E}\left[\hat{f}_\lambda^{(r)} | G\right] = (T_G + \lambda)^{-1} T_G f_\rho^{*(r)}, \quad r = 1, \ldots, d,$$

where we use (3.1). Hence, the excess risk

$$
\begin{aligned}
R(\lambda) &= \sum_{r=1}^d \left\| \hat{f}_\lambda^{(r)} - f_\rho^{*(r)} \right\|_{L^2}^2 \\
&= \sum_{r=1}^d \left\| \tilde{f}_\lambda^{(r)} - f_\rho^{*(r)} + \frac{1}{nk} \sum_{i=1}^n \sum_{j=1}^k (T_G + \lambda)^{-1} k(g_{ij}, \cdot) \epsilon_{ij}^{(r)} \right\|_{L^2}^2 \\
&\leqslant 2 \sum_{r=1}^d \left( \left\| \tilde{f}_\lambda^{(r)} - f_\rho^{*(r)} \right\|_{L^2}^2 + \left\| \frac{1}{nk} \sum_{i=1}^n \sum_{j=1}^k (T_G + \lambda)^{-1} k(g_{ij}, \cdot) \epsilon_{ij}^{(r)} \right\|_{L^2}^2 \right).
\end{aligned}
$$

For each $r = 1, \ldots, d$, we define

$$\mathrm{B}_r^2(\lambda) := \left\| \tilde{f}_\lambda^{(r)} - f_\rho^{*(r)} \right\|_{L^2}^2, \quad \mathrm{V}_r(\lambda) := \left\| \frac{1}{nk} \sum_{i=1}^n \sum_{j=1}^k (T_G + \lambda)^{-1} k(g_{ij}, \cdot) \epsilon_{ij}^{(r)} \right\|_{L^2}^2. \tag{A.1}$$

The key part of our proof is to provide bounds for $\mathrm{B}_r(\lambda)$ and $\mathrm{V}_r(\lambda)$.

## A.2 Large Regularization Induces Non-vacuous Generalization

In this section, we prove Theorem 4.1. In particular, we derive upper bounds for bias and variance respectively in Section A.2.1 and Section A.2.2.

### A.2.1 Bounds for the Bias Term

**Theorem A.1.** *Under Assumption 1, if $\lambda \asymp n^{-\theta}, \theta \in (0, \beta)$, then*

$$\mathrm{B}_r(\lambda) \leqslant \tilde{\Theta}_{\mathbb{P}} \left( n^{-\min(s,2)\theta/2} \right), \quad r = 1, \ldots, d.$$

*Proof.* We analyze bounds in each dimension. For simplicity, we ignore script $r$. We decompose the bias term by introducing the expectation of $\tilde{f}_\lambda = (T_G + \lambda)^{-1} T_G f_\rho^*$:

$$f_\lambda = (T + \lambda)^{-1} T f_\rho^*,$$

then

$$\left\| f_\lambda - f_\rho^* \right\|_{L^2} - \left\| \tilde{f}_\lambda - f_\lambda \right\|_{L^2} \leqslant \mathrm{B}(\lambda) = \left\| \tilde{f}_\lambda - f_\rho^* \right\|_{L^2} \leqslant \left\| f_\lambda - f_\rho^* \right\|_{L^2} + \left\| \tilde{f}_\lambda - f_\lambda \right\|_{L^2}.$$

We first compute the term $\left\| f_\lambda - f_\rho^* \right\|_{L^2}$. Apply Lemma C.4 and take $\gamma = 0$ we have

$$\left\| f_\lambda - f_\rho^* \right\|_{L^2} = \tilde{\Theta}\left( n^{-\min(s,2)\theta/2} \right).$$

Secondly, with regards to $\left\| \tilde{f}_\lambda - f_\lambda \right\|_{L^2}$, the key is to perform concentration analysis. To this end, we decompose $\left\| \tilde{f}_\lambda - f_\lambda \right\|_{L^2}$ into two components related to $G$ and analyze each component by concentration.

$$
\begin{aligned}
\left\| \tilde{f}_\lambda - f_\lambda \right\|_{L^2} &= \left\| T^{\frac{1}{2}}\left( \tilde{f}_\lambda - f_\lambda \right) \right\|_{\mathcal{H}} \\
&= \left\| T^{\frac{1}{2}} T_{G\lambda}^{-1}\left( T_G f_\rho^* - T_{G\lambda} f_\lambda \right) \right\|_{\mathcal{H}} \\
&\leqslant \left\| T^{\frac{1}{2}} T_\lambda^{-\frac{1}{2}} \right\| \cdot \left\| T_\lambda^{\frac{1}{2}} T_{G\lambda}^{-1} T_\lambda^{\frac{1}{2}} \right\| \cdot \left\| T_\lambda^{-\frac{1}{2}}\left( T_G f_\rho^* - T_{G\lambda} f_\lambda \right) \right\|_{\mathcal{H}} \\
&\leqslant \left\| T_\lambda^{\frac{1}{2}} T_{G\lambda}^{-1} T_\lambda^{\frac{1}{2}} \right\| \cdot \left\| T_\lambda^{-\frac{1}{2}}\left( T_G f_\rho^* - T_{G\lambda} f_\lambda \right) \right\|_{\mathcal{H}} \\
&= \left\| T_\lambda^{\frac{1}{2}} T_{G\lambda}^{-1} T_\lambda^{\frac{1}{2}} \right\| \cdot \left\| T_\lambda^{-\frac{1}{2}}\left( T_G f_\rho^* - \left( T_G + \lambda + T - T \right) f_\lambda \right) \right\|_{\mathcal{H}} \\
&= \left\| T_\lambda^{\frac{1}{2}} T_{G\lambda}^{-1} T_\lambda^{\frac{1}{2}} \right\| \cdot \left\| T_\lambda^{-\frac{1}{2}}\left[ \left( T_G f_\rho^* - T_G f_\lambda \right) - \left( T f_\rho^* - T f_\lambda \right) \right] \right\|_{\mathcal{H}} \\
&\leqslant \left| 1 - \left\| T_\lambda^{-\frac{1}{2}}(T - T_G) T_\lambda^{-\frac{1}{2}} \right\| \right|^{-1} \cdot \left\| T_\lambda^{-\frac{1}{2}}\left[ \left( T_G f_\rho^* - T_G f_\lambda \right) - \left( T f_\rho^* - T f_\lambda \right) \right] \right\|_{\mathcal{H}},
\end{aligned}
$$

where the last inequality utilizes the fact that:

$$
\begin{aligned}
\left\| T_\lambda^{\frac{1}{2}} T_{G\lambda}^{-1} T_\lambda^{\frac{1}{2}} \right\| &= \left\| \left\{ T_\lambda^{-\frac{1}{2}} T_{G\lambda} T_\lambda^{-\frac{1}{2}} \right\}^{-1} \right\| \\
&= \left\| \left\{ 1 - T_\lambda^{-\frac{1}{2}}\left( T - T_G \right) T_\lambda^{-\frac{1}{2}} \right\}^{-1} \right\| \\
&\leqslant \left| 1 - \left\| T_\lambda^{-\frac{1}{2}}\left( T - T_G \right) T_\lambda^{-\frac{1}{2}} \right\| \right|^{-1}.
\end{aligned}
$$

By Lemma B.1, for $\alpha > \alpha_0$ being sufficiently close, with high probability,

$$\left\| T_\lambda^{-\frac{1}{2}}(T - T_G) T_\lambda^{-\frac{1}{2}} \right\| \leqslant \tilde{O}_{\mathbb{P}}\left( \sqrt{\frac{\lambda^{-\alpha}}{n}} \right) = \tilde{o}_{\mathbb{P}}(1).$$

By Lemma B.2, with high probability,

$$\left\| T_\lambda^{-\frac{1}{2}}\left[ \left( T_G f_\rho^* - T_G f_\lambda \right) - \left( T f_\rho^* - T f_\lambda \right) \right] \right\|_{\mathcal{H}} \leqslant \tilde{o}_{\mathbb{P}}\left( \lambda^{\frac{\bar{s}}{2}} \right).$$

Hence,

$$\left\| \tilde{f}_\lambda - f_\lambda \right\|_{L^2} \leqslant \tilde{o}_{\mathbb{P}}\left( \lambda^{\frac{\bar{s}}{2}} \right).$$

Therefore,

$$\mathrm{B}(\lambda) = \tilde{\Theta}_{\mathbb{P}}\left( n^{-\min(s,2)\theta/2} \right).$$

$\square$

### A.2.2 Bounds for the Variance Term

**Theorem A.2.** *Under Assumption 1, if $\lambda \asymp n^{-\theta}, \theta \in (0, \beta)$, then*

$$\mathrm{V}_r(\lambda) \leqslant \tilde{\sigma}^2 O_{\mathbb{P}}^{\mathrm{poly}}\left( n^{\alpha_0 \theta}\left( \frac{r_T}{n} + \frac{1 - r_T}{nk} \right) \right), \quad r = 1, \ldots, d.$$

*Proof.* By definition,

$$V_r(\lambda) = \left\| \frac{1}{nk} \sum_{i=1}^{n} \sum_{j=1}^{k} (T_G + \lambda)^{-1} k(g_{ij}, \cdot) \epsilon_{ij}^{(r)} \right\|_{L^2}^2$$

$$= \int_{\mathcal{G}} \frac{1}{n^2 k^2} \left[ \sum_{i=1}^{n} \sum_{j=1}^{k} (T_G + \lambda)^{-1} k(g_{ij}, g) \epsilon_{ij}^{(r)} \right]^2 d\mu_{\mathcal{G}}(g).$$

For simplicity, we first ignore script $r$ in $\epsilon_{ij}^{(r)}$. The proof for upper bounding $V(\lambda)$ undertakes several steps of concentration. We first separate $T_G$ by interpolating its expectation $T$, and then analyze the discrepancy between $T_G$ and $T$ and the remaining term respectively.

$$\frac{1}{n^2 k^2} \left[ \sum_{i=1}^{n} \sum_{j=1}^{k} T_{G\lambda}^{-1} k(g_{ij}, g) \epsilon_{ij} \right]^2$$

$$= \underbrace{\frac{1}{n^2 k^2} \left[ \sum_{i=1}^{n} \sum_{j=1}^{k} T_{G\lambda}^{-1} k(g_{ij}, g) \epsilon_{ij} \right]^2 - \frac{1}{n^2 k^2} \left[ \sum_{i=1}^{n} \sum_{j=1}^{k} T_\lambda^{-1} k(g_{ij}, g) \epsilon_{ij} \right]^2}_{\Delta G} \tag{A.2}$$

$$+ \underbrace{\frac{1}{n^2 k^2} \left[ \sum_{i=1}^{n} \sum_{j=1}^{k} T_\lambda^{-1} k(g_{ij}, g) \epsilon_{ij} \right]^2}_{V}.$$

Intuitively, $V$ is consist of the covariance between sampled observations, which can be categorized by the expectation value into the covariance of an individual observation itself, the covariance across noises per signal and the covariance across signals. We further perform decomposition according to these intuition:

$$V = \frac{1}{n^2 k^2} \sum_{i_1=1}^{n} \sum_{i_2=1}^{n} \sum_{j_1=1}^{k} \sum_{j_2=1}^{k} T_\lambda^{-1} k(g_{i_1 j_1}, g) T_\lambda^{-1} k(g_{i_2 j_2}, g) [\epsilon_{i_1 j_1} \epsilon_{i_2 j_2}]$$

$$= \underbrace{\frac{1}{n^2 k^2} \sum_{i=1}^{n} \sum_{j=1}^{k} \left[ T_\lambda^{-1} k(g_{ij}, g) \epsilon_{ij} \right]^2}_{V_1}$$

$$+ \underbrace{\frac{1}{n^2 k^2} \sum_{i=1}^{n} \sum_{j_1=1}^{k} \sum_{j_2=1, j_2 \neq j_1}^{k} T_\lambda^{-1} k(g_{ij_1}, g) T_\lambda^{-1} k(g_{ij_2}, g) \epsilon_{ij_1} \epsilon_{ij_2}}_{V_2}$$

$$+ \underbrace{\frac{1}{n^2 k^2} \sum_{i_1=1}^{n} \sum_{i_2=1, i_2 \neq i_1}^{n} \sum_{j_1=1}^{k} \sum_{j_2=1}^{k} T_\lambda^{-1} k(g_{i_1 j_1}, g) T_\lambda^{-1} k(g_{i_2 j_2}, g) \epsilon_{i_1 j_1} \epsilon_{i_2 j_2}}_{V_3},$$

where $V_1$ characterizes the covariance of observation, $V_2$ captures the covariance across noises per signal and $V_3$ reflects the covariance across signals. We first bound $V_1$. By Lemma B.3, for $\alpha > \alpha_0$ being sufficiently close, for $g \in \mathcal{G}$ almost everywhere,

$$\left| \frac{1}{nk} \sum_{i=1}^{n} \sum_{j=1}^{k} \left[ T_\lambda^{-1} k(g_{ij}, g) \right]^2 - \left\| T_\lambda^{-1} k(g, \cdot) \right\|_{L^2}^2 \right| \leq \tilde{O}_{\mathbb{P}} \left( \lambda^{-\alpha} \sqrt{\frac{\lambda^{-\alpha}}{n}} \right).$$

By Lemma B.7, for $\alpha > \alpha_0$ being sufficiently close, for $g \in \mathcal{G}$ almost everywhere,

$$\left| \frac{1}{nk} \sum_{i=1}^{n} \sum_{j=1}^{k} \left[ T_\lambda^{-1} k(g_{ij}, g) \epsilon_{ij} \right]^2 - \frac{1}{nk} \sum_{i=1}^{n} \sum_{j=1}^{k} \left[ T_\lambda^{-1} k(g_{ij}, g) \right]^2 \mathbb{E} \left[ \epsilon_{ij}^2 | g_{ij} \right] \right| \leq \tilde{O}_{\mathbb{P}} \left( \sigma_\epsilon^2 \lambda^{-\alpha} \sqrt{\frac{\lambda^{-\alpha}}{n}} \right).$$

Jointly, for $\alpha > \alpha_0$ being sufficiently close, with high probability, for $g \in \mathcal{G}$ almost everywhere,

$$
\begin{aligned}
V_1 &= \frac{1}{n^2k^2} \sum_{i=1}^{n} \sum_{j=1}^{k} \left[ T_\lambda^{-1} k(g_{ij}, g) \epsilon_{ij} \right]^2 \\
&\leqslant \frac{1}{nk} \left[ \frac{1}{nk} \sum_{i=1}^{n} \sum_{j=1}^{k} \left[ T_\lambda^{-1} k(g_{ij}, g) \right]^2 \mathbb{E} \left[ \epsilon_{ij}^2 | g_{ij} \right] + \tilde{O}_\mathbb{P} \left( \sigma_\epsilon^2 \lambda^{-\alpha} \sqrt{\frac{\lambda^{-\alpha}}{n}} \right) \right] \\
&\leqslant \frac{\sigma^2}{nk} \left[ \left\| T_\lambda^{-1} k(g, \cdot) \right\|_{L^2}^2 + \tilde{O}_\mathbb{P} \left( \lambda^{-\alpha} \sqrt{\frac{\lambda^{-\alpha}}{n}} \right) \right] + \frac{1}{nk} \tilde{O}_\mathbb{P} \left( \sigma_\epsilon^2 \lambda^{-\alpha} \sqrt{\frac{\lambda^{-\alpha}}{n}} \right) \\
&= \frac{\sigma^2}{nk} \left[ \left\| T_\lambda^{-1} k(g, \cdot) \right\|_{L^2}^2 + \tilde{o}_\mathbb{P} \left( \lambda^{-\alpha} \right) \right] \\
&= \frac{\sigma^2}{nk} \tilde{O}_\mathbb{P} \left( \left\| T_\lambda^{-1} k(g, \cdot) \right\|_{L^2}^2 \right),
\end{aligned}
$$

where the last equality results from Corollary C.2 that

$$
\left\| T_\lambda^{-1} k(g, \cdot) \right\|_{L^2}^2 \leqslant M_\alpha^2 \lambda^{-\alpha} = O\left( \lambda^{-\alpha} \right).
$$

We then bound $V_2$. By Lemma B.8, for $\alpha > \alpha_0$ being sufficiently close, with high probability, for $g \in \mathcal{G}$ almost everywhere,

$$
\begin{aligned}
&\left| \frac{1}{nk(k-1)} \sum_{i=1}^{n} \sum_{j_1=1}^{k} \sum_{j_2=1, j_2 \neq j_1}^{k} T_\lambda^{-1} k(g_{ij_1}, g) \epsilon_{ij_1} T_\lambda^{-1} k(g_{ij_2}, g) \epsilon_{ij_2} - \mathbb{E} T_\lambda^{-1} k(g_{ij_1}, g) \epsilon_{ij_1} T_\lambda^{-1} k(g_{ij_2}, g) \epsilon_{ij_2} \right| \\
&\leqslant (\sigma_{\epsilon_{1,2}}^2 + \sigma_G^2) \tilde{o}_\mathbb{P}(\lambda^{-\alpha}).
\end{aligned}
$$

Therefore, togeter with Corollary C.2, we have, with high probability, for $g \in \mathcal{G}$ almost everywhere,

$$
V_2 \leqslant r_T \frac{\tilde{\sigma}^2 (k-1)}{nk} \tilde{O}_\mathbb{P} \left( \left\| T_\lambda^{-1} k(g, \cdot) \right\|_{L^2}^2 \right).
$$

We at last bound $V_3$. By Lemma B.9, with high probability, for $g \in \mathcal{G}$ almost everywhere,

$$
\begin{aligned}
V_3 &\leqslant \left| \frac{1}{n^2k^2} \sum_{i_1=1}^{n} \sum_{i_2=1, i_2 \neq i_1}^{n} \sum_{j_1=1}^{k} \sum_{j_2=1}^{k} T_\lambda^{-1} k(g_{i_1 j_1}, g) T_\lambda^{-1} k(g_{i_2 j_2}, g) \epsilon_{i_1 j_1} \epsilon_{i_2 j_2} \right| \\
&\leqslant \tilde{\sigma}^2 \tilde{O}_\mathbb{P} \left( \frac{\left\| T_\lambda^{-1} k(g, \cdot) \right\|_{L^2}^2}{n} \left( r_T + \frac{1 - r_T}{k} \right) \right).
\end{aligned}
$$

Here we complete the analysis for $V$. For $\Delta G$,

$$
\begin{aligned}
\Delta G &= \frac{1}{n^2k^2} \left[ \sum_{i=1}^{n} \sum_{j=1}^{k} T_{G\lambda}^{-1} k(g_{ij}, g) \epsilon_{ij} \right]^2 - \frac{1}{n^2k^2} \left[ \sum_{i=1}^{n} \sum_{j=1}^{k} T_\lambda^{-1} k(g_{ij}, g) \epsilon_{ij} \right]^2 \\
&= \left[ \left( \frac{1}{nk} \sum_{i=1}^{n} \sum_{j=1}^{k} T_{G\lambda}^{-1} k(g_{ij}, g) \epsilon_{ij} \right) - \left( \frac{1}{nk} \sum_{i=1}^{n} \sum_{j=1}^{k} T_\lambda^{-1} k(g_{ij}, g) \epsilon_{ij} \right) \right] \\
&\quad \left[ \left( \frac{1}{nk} \sum_{i=1}^{n} \sum_{j=1}^{k} T_{G\lambda}^{-1} k(g_{ij}, g) \epsilon_{ij} \right) + \left( \frac{1}{nk} \sum_{i=1}^{n} \sum_{j=1}^{k} T_\lambda^{-1} k(g_{ij}, g) \epsilon_{ij} \right) \right].
\end{aligned}
$$

We first apply Corollary B.5 to deal with $\left( \frac{1}{nk} \sum_{i=1}^{n} \sum_{j=1}^{k} T_{G\lambda}^{-1} k(g_{ij}, g) \epsilon_{ij} \right) - \left( \frac{1}{nk} \sum_{i=1}^{n} \sum_{j=1}^{k} T_\lambda^{-1} k(g_{ij}, g) \epsilon_{ij} \right)$. For $\alpha > \alpha_0$ being sufficiently close, with high probabil-

ity, for $g \in \mathcal{G}$ almost everywhere,

$$\left| \left( \frac{1}{nk} \sum_{i=1}^{n} \sum_{j=1}^{k} T_{G\lambda}^{-1} k(g_{ij}, g) \epsilon_{ij} \right) - \left( \frac{1}{nk} \sum_{i=1}^{n} \sum_{j=1}^{k} T_{\lambda}^{-1} k(g_{ij}, g) \epsilon_{ij} \right) \right|$$

$$= \left| \frac{1}{nk} \sum_{i=1}^{n} \sum_{j=1}^{k} \left( T_{G\lambda}^{-1} - T_{\lambda}^{-1} \right) k(g_{ij}, g) \epsilon_{ij} \right|$$

$$\leqslant \tilde{O}_{\mathbb{P}} \left( \sqrt{\frac{\sigma_k^2}{n} \frac{1}{nk} \sum_{i=1}^{n} \sum_{j=1}^{k} \left[ \left( T_{G\lambda}^{-1} - T_{\lambda}^{-1} \right) k(g_{ij}, g) \right]^2} \right).$$

Note that

$$\sqrt{\frac{\sigma_k^2}{n} \frac{1}{nk} \sum_{i=1}^{n} \sum_{j=1}^{k} \left[ \left( T_{G\lambda}^{-1} - T_{\lambda}^{-1} \right) k(g_{ij}, g) \right]^2} = \sqrt{\frac{\sigma_k^2}{n}} \left\| T_G^{\frac{1}{2}} \left( T_{G\lambda}^{-1} - T_{\lambda}^{-1} \right) k(g, \cdot) \right\|_{\mathcal{H}},$$

we then separate it to several components related to $G$ as in the proof of Theorem A.1 to perform concentration.

$$\tilde{O} \left( \sqrt{\frac{\sigma_k^2}{n}} \left\| T_G^{\frac{1}{2}} \left( T_{G\lambda}^{-1} - T_{\lambda}^{-1} \right) k(g, \cdot) \right\|_{\mathcal{H}} \right)$$

$$= \tilde{O}_{\mathbb{P}} \left( \sqrt{\frac{\sigma_k^2}{n}} \left\| T_G^{\frac{1}{2}} T_{G\lambda}^{-1} \left( T_G - T \right) T_{\lambda}^{-1} k(g, \cdot) \right\|_{\mathcal{H}} \right)$$

$$= \tilde{O}_{\mathbb{P}} \left( \sqrt{\frac{\sigma_k^2}{n}} \left\| T_G^{\frac{1}{2}} T_{G\lambda}^{-\frac{1}{2}} T_{G\lambda}^{-\frac{1}{2}} T_{\lambda}^{\frac{1}{2}} T_{\lambda}^{-\frac{1}{2}} \left( T_G - T \right) T_{\lambda}^{-\frac{1}{2}} T_{\lambda}^{-\frac{1}{2}} k(g, \cdot) \right\|_{\mathcal{H}} \right)$$

$$\leqslant \tilde{O}_{\mathbb{P}} \left( \sqrt{\frac{\sigma_k^2}{n}} \left\| T_G^{\frac{1}{2}} T_{G\lambda}^{-\frac{1}{2}} \right\| \left\| T_{G\lambda}^{-\frac{1}{2}} T_{\lambda}^{\frac{1}{2}} \right\| \left\| T_{\lambda}^{-\frac{1}{2}} \left( T_G - T \right) T_{\lambda}^{-\frac{1}{2}} \right\| \left\| T_{\lambda}^{-\frac{1}{2}} k(g, \cdot) \right\|_{\mathcal{H}} \right).$$

By Lemma B.1, for $\alpha > \alpha_0$ being sufficiently close, with high probability,

$$\left\| T_{G\lambda}^{-\frac{1}{2}} T_{\lambda}^{\frac{1}{2}} \right\| \leqslant O_{\mathbb{P}}(1),$$

$$\left\| T_{\lambda}^{-\frac{1}{2}} \left( T_G - T \right) T_{\lambda}^{-\frac{1}{2}} \right\| \leqslant \tilde{O}_{\mathbb{P}} \left( \sqrt{\frac{\lambda^{-\alpha}}{n}} \right).$$

By Corollary C.2,

$$\left\| T_{\lambda}^{-\frac{1}{2}} k(g, \cdot) \right\|_{\mathcal{H}} \leqslant M_{\alpha} \lambda^{-\frac{\alpha}{2}}.$$

Note that

$$\left\| T_G^{\frac{1}{2}} T_{G\lambda}^{-\frac{1}{2}} \right\| \leqslant \sup_{t \geqslant 0} \sqrt{\frac{t}{t+\lambda}} \leqslant 1,$$

we have for $\alpha > \alpha_0$ being sufficiently close, with high probability, for $g \in \mathcal{G}$ almost everywhere,

$$\left| \left( \frac{1}{nk} \sum_{i=1}^{n} \sum_{j=1}^{k} T_{G\lambda}^{-1} k(g_{ij}, g) \epsilon_{ij} \right) - \left( \frac{1}{nk} \sum_{i=1}^{n} \sum_{j=1}^{k} T_{\lambda}^{-1} k(g_{ij}, g) \epsilon_{ij} \right) \right| \leqslant \sigma_k \tilde{O}_{\mathbb{P}} \left( \sqrt{\frac{\lambda^{-2\alpha}}{n^2}} \right). \quad \text{(A.3)}$$

Secondly, regarding $\left( \frac{1}{nk} \sum_{i=1}^{n} \sum_{j=1}^{k} T_{G\lambda}^{-1} k(g_{ij}, g) \epsilon_{ij} \right) + \left( \frac{1}{nk} \sum_{i=1}^{n} \sum_{j=1}^{k} T_{\lambda}^{-1} k(g_{ij}, g) \epsilon_{ij} \right)$, we intend to handle this term by Equation (A.3). For $\alpha > \alpha_0$ being sufficiently close, with high probability,

for $g \in \mathcal{G}$ almost everywhere,

$$\left| \left( \frac{1}{nk} \sum_{i=1}^{n} \sum_{j=1}^{k} T_{G\lambda}^{-1} k(g_{ij}, g) \epsilon_{ij} \right) + \left( \frac{1}{nk} \sum_{i=1}^{n} \sum_{j=1}^{k} T_{\lambda}^{-1} k(g_{ij}, g) \epsilon_{ij} \right) \right|$$

$$\leqslant 2 \left| \frac{1}{nk} \sum_{i=1}^{n} \sum_{j=1}^{k} T_{\lambda}^{-1} k(g_{ij}, g) \epsilon_{ij} \right| + \left| \left( \frac{1}{nk} \sum_{i=1}^{n} \sum_{j=1}^{k} T_{G\lambda}^{-1} k(g_{ij}, g) \epsilon_{ij} \right) - \left( \frac{1}{nk} \sum_{i=1}^{n} \sum_{j=1}^{k} T_{\lambda}^{-1} k(g_{ij}, g) \epsilon_{ij} \right) \right|$$

$$\leqslant \tilde{O}_{\mathbb{P}} \left( \sqrt{\frac{\lambda^{-\alpha}}{n} \sigma_{\epsilon}^2} \right) + \sigma_k \tilde{O}_{\mathbb{P}} \left( \sqrt{\frac{\lambda^{-2\alpha}}{n^2}} \right),$$

$$\tag{A.4}$$

where the last inequality results from Lemma B.4 and Equation (A.3).

Therefore, for $\alpha > \alpha_0$ being sufficiently close, with high probability, for $g \in \mathcal{G}$ almost everywhere,

$$\Delta G \leqslant \sigma_k \tilde{O}_{\mathbb{P}} \left( \sqrt{\frac{\lambda^{-2\alpha}}{n^2}} \right) \cdot \left[ \tilde{O}_{\mathbb{P}} \left( \sqrt{\frac{\lambda^{-\alpha}}{n} \sigma_{\epsilon}^2} \right) + \sigma_k \tilde{O}_{\mathbb{P}} \left( \sqrt{\frac{\lambda^{-2\alpha}}{n^2}} \right) \right]$$

$$= (\sigma_{\epsilon}^2 \vee \sigma_k^2) \tilde{O}_{\mathbb{P}} \left( \sqrt{\frac{\lambda^{-3\alpha}}{n^3}} \right).$$

Combining the bounds for $V$ and $\Delta G$, with high probability, for $g \in \mathcal{G}$ almost everywhere,

$$\frac{1}{n^2 k^2} \left[ \sum_{i=1}^{n} \sum_{j=1}^{k} T_{G\lambda}^{-1} k(g_{ij}, g) \epsilon_{ij} \right]^2 \leqslant \tilde{O}_{\mathbb{P}} \left( \|T_{\lambda}^{-1} k(g, \cdot)\|_{L^2}^2 \right) \tilde{\sigma}^2 \left( \frac{r_T}{n} + \frac{1 - r_T}{nk} \right).$$

Hence,

$$V(\lambda) = \int_{\mathcal{G}} \frac{1}{n^2 k^2} \left[ \sum_{i=1}^{n} \sum_{j=1}^{k} T_{G\lambda}^{-1} k(g_{ij}, g) \epsilon_{ij} \right]^2 d\mu_{\mathcal{G}}(g)$$

$$\leqslant \tilde{\sigma}^2 O_{\mathbb{P}}^{\text{poly}} \left( n^{\alpha_0 \theta} \left( \frac{r_T}{n} + \frac{1 - r_T}{nk} \right) \right).$$

$\square$

### A.2.3 Excess Risk Bounds under Conditional Orthogonality Condition

**Theorem A.3.** *Under Assumption 1 and the conditional orthogonality, if $\lambda \asymp n^{-\theta}, \theta \in (0, \beta)$,*

$$R(\lambda) \leqslant \underbrace{\tilde{\Theta}_{\mathbb{P}} \left( n^{-\min(s, 2)\theta} \right)}_{\text{Bias}^2(\lambda)} + \underbrace{\tilde{\sigma}^2 O_{\mathbb{P}}^{\text{poly}} \left( n^{\alpha_0 \theta} \left( \frac{r_0 \vee r_e}{n} + \frac{(1 - r_0) \wedge (1 - r_e)}{nk} \right) \right)}_{\text{Var}(\lambda)}.$$

*Proof.* This can be directly proved by applying Lemma B.11 to Theorem A.1 and Theorem A.2. $\square$

## A.3 Small Regularization Induces Vacuous Generalization

In this section, we prove Theorem 4.3. To be specific, we derive upper bounds for bias and variance under small regularization respectively in Section A.3.1 and Section A.3.2.

### A.3.1 Bounds for the Bias term

**Theorem A.4.** *Under Assumption 1, if $\lambda \asymp n^{-\theta}$, $\theta \in [\beta, \infty)$, then*

$$B_r(\lambda) \leqslant O_{\mathbb{P}}^{poly} \left( n^{-\min(s, 2)\beta/2} \right), \quad r = 1, \ldots, d.$$

*Proof.* Similar to [37], the bias term can also be written as

$$B_r(\lambda) = \|\lambda(T_G + \lambda)^{-1} f_\rho^{*(r)}\|_{L^2}.$$

For simplicity, we first ignore script $r$. Similar to [37], we assume that $f_\rho^* = T^{\frac{t}{2}} g$ for some $g \in L^2$ with $\|g\|_{L^2} \leqslant C$, and restrict further that $t \leqslant 2$. Let $\tilde{\lambda} \asymp n^{-l}$ for $l \in (0, \beta)$. Using the notations of Lemma C.12, denote $\psi_\lambda = \lambda(T_G + \lambda)^{-1}$, by the definition of $B(\lambda)$, we have

$$B(\lambda) = \|\psi_\lambda f_\rho^*\|_{L^2} = \left\| T^{1/2} \psi_\lambda T^{\frac{t-1}{2}} \cdot T^{1/2} g \right\|_{\mathcal{H}}.$$

Utilizing Lemma C.12,

$$
\begin{aligned}
\left\| T^{1/2} \psi_\lambda T^{\frac{t-1}{2}} \cdot T^{1/2} g \right\|_{\mathcal{H}} &\leqslant \left\| T^{1/2} \psi_\lambda T^{(t-1)/2} \right\| \cdot \left\| T^{1/2} g \right\|_{\mathcal{H}} \\
&\leqslant C \| T^{1/2} \psi_\lambda^{1/2} \| \cdot \| \psi_\lambda^{1/2} T^{\frac{t-1}{2}} \| \\
&\leqslant C \| T^{1/2} \psi_{\tilde{\lambda}}^{1/2} \| \cdot \| \psi_{\tilde{\lambda}}^{1/2} T^{\frac{t-1}{2}} \| \\
&\leqslant C \| T^{1/2} \psi_{\tilde{\lambda}}^{1/2} \| \cdot \| \psi_{\tilde{\lambda}}^{(2-t)/2} \| \cdot \| \psi_{\tilde{\lambda}}^{\frac{t-1}{2}} T^{\frac{t-1}{2}} \| \\
&= C \tilde{\lambda}^{t/2} \| \psi_{\tilde{\lambda}}^{(2-t)/2} \| \cdot \| T^{1/2} T_{G\tilde{\lambda}}^{-1/2} \| \cdot \| T^{\frac{t-1}{2}} T_{G\tilde{\lambda}}^{-\frac{t-1}{2}} \| \\
&\leqslant C \tilde{\lambda}^{t/2} \| T^{1/2} T_{G\tilde{\lambda}}^{-1/2} \|^t,
\end{aligned}
$$

where the third inequality uses Lemma C.12, the last equality uses the definition of $\psi$ and the last inequality uses Lemma C.6. Finally, by Lemma B.1, with high probability,

$$\left\| T^{\frac{1}{2}} T_{G\tilde{\lambda}}^{-\frac{1}{2}} \right\| = \left\| T^{\frac{1}{2}} T_\lambda^{-\frac{1}{2}} T_\lambda^{\frac{1}{2}} T_{G\tilde{\lambda}}^{-\frac{1}{2}} \right\| \leqslant \left\| T^{\frac{1}{2}} T_\lambda^{-\frac{1}{2}} \right\| \left\| T_\lambda^{\frac{1}{2}} T_{G\tilde{\lambda}}^{-\frac{1}{2}} \right\| \leqslant O(1).$$

Since $t < \min(s, 2)$ and $l < \beta$ can be arbitrarily close,

$$B(\lambda) = O_{\mathbb{P}}\left( \tilde{\lambda}^{t/2} \right) = O_{\mathbb{P}}^{\mathrm{poly}}\left( n^{-\min(s,2)\beta/2} \right).$$

$\square$

### A.3.2 Bounds for the Variance term

**Theorem A.5.** *Under Assumption 1, if $\lambda \asymp n^{-\theta}$, $\theta \in [\beta, \infty)$,*

$$\mathbb{E}_{\epsilon_{1,1}|g_{1,1}} \mathbb{E}_{\epsilon_{1,2}|g_{1,2}} \cdots \mathbb{E}_{\epsilon_{n,k}|g_{n,k}} \left[ V_r(\lambda) \right] \geqslant \Omega_{\mathbb{P}}^{\mathrm{poly}}\left( \frac{\sigma_L^2}{k} \right), \quad r = 1, \ldots, d,$$

*where $\sigma_L^2$ is the lower bound of $\mathbb{E}[\epsilon^{(r)2}|g]$ for $g \in \mathcal{G}$ almost everywhere.*

*Proof.* The proof is similar to the proof of lower bound in [37]. Note that $V_r(\lambda)$ is monotonically decreasing with respect to $\lambda$, it holds

$$V_r(\lambda) \geqslant V_r(n^{-\beta}), \quad r = 1, \ldots, d.$$

Following the notations in Section A.2.2 and ignoring the script $r$, we have

$$V(\lambda) = \int_{\mathcal{G}} \frac{1}{n^2 k^2} \left[ \sum_{i=1}^n \sum_{j=1}^k T_{G\lambda}^{-1} k(g_{ij}, g) \epsilon_{ij} \right]^2 d\mu_{\mathcal{G}}(g).$$

By the optimality (3.1), if we further assume as Li et al. [37, 39],

$$\mathbb{E}[\epsilon|g] = 0, \quad g \in \mathcal{G} \ \text{almost everywhere},$$

then, for $\alpha > \alpha_0$ being sufficiently close, with high probability,

$$\mathbb{E}_{\epsilon_{1,1}|g_{1,1}}\mathbb{E}_{\epsilon_{1,2}|g_{1,2}}\ldots\mathbb{E}_{\epsilon_{n,k}|g_{n,k}}\left[\frac{1}{n^2k^2}\left(\sum_{i=1}^{n}\sum_{j=1}^{k}T_{G\lambda}^{-1}k(g_{ij},g)\epsilon_{ij}\right)^2\right]$$

$$=\mathbb{E}_{\epsilon_{1,1}|g_{1,1}}\mathbb{E}_{\epsilon_{1,2}|g_{1,2}}\ldots\mathbb{E}_{\epsilon_{n,k}|g_{n,k}}\left[\frac{1}{n^2k^2}\sum_{i=1}^{n}\sum_{j=1}^{k}\left(T_{G\lambda}^{-1}k(g_{ij},g)\epsilon_{ij}\right)^2\right]$$

$$=\mathbb{E}_{\epsilon_{1,1}|g_{1,1}}\mathbb{E}_{\epsilon_{1,2}|g_{1,2}}\ldots\mathbb{E}_{\epsilon_{n,k}|g_{n,k}}\left[\frac{1}{n^2k^2}\sum_{i=1}^{n}\sum_{j=1}^{k}\left(T_{G\lambda}^{-1}k(g_{ij},g)\epsilon_{ij}\right)^2\right]$$

$$\geqslant\frac{\sigma_L^2}{nk}\left[\frac{1}{nk}\sum_{i=1}^{n}\sum_{j=1}^{k}\left(T_{G\lambda}^{-1}k(g_{ij},g)\right)^2\right],$$

where we use Lemma B.7. By Lemma B.10, for $\alpha > \alpha_0$ being sufficiently close, with high probability,

$$\left|\left\|T_G^{\frac{1}{2}}T_{G\lambda}^{-1}k(g,\cdot)\right\|_{\mathcal{H}}-\left\|T_G^{\frac{1}{2}}T_\lambda^{-1}k(g,\cdot)\right\|_{\mathcal{H}}\right|\leqslant\tilde{O}_{\mathbb{P}}\left(\lambda^{-\frac{\alpha}{2}}\sqrt{\frac{\lambda^{-\alpha}}{n}}\right):=R_1.$$

By Lemma B.3, for $\alpha > \alpha_0$ being sufficiently close, with high probability, for $g \in \mathcal{G}$ almost everywhere,

$$\left|\frac{1}{nk}\sum_{i=1}^{n}\sum_{j=1}^{k}\left[T_\lambda^{-1}k(g_{ij},g)\right]^2-\left\|T_\lambda^{-1}k(g,\cdot)\right\|_{L^2}^2\right|\leqslant\tilde{O}_{\mathbb{P}}\left(\lambda^{-\alpha}\sqrt{\frac{\lambda^{-\alpha}}{n}}\right).$$

Hence, with high probability, for $g \in \mathcal{G}$ almost everywhere,

$$\frac{1}{nk}\sum_{i=1}^{n}\sum_{j=1}^{k}\left(T_{G\lambda}^{-1}k(g_{ij},g)\right)^2$$

$$=\left\|T_G^{\frac{1}{2}}T_{G\lambda}^{-1}k(g,\cdot)\right\|_{\mathcal{H}}^2$$

$$=\left\|T_G^{\frac{1}{2}}T_{G\lambda}^{-1}k(g,\cdot)\right\|_{\mathcal{H}}^2-\left\|T_G^{\frac{1}{2}}T_\lambda^{-1}k(g,\cdot)\right\|_{\mathcal{H}}^2+\left\|T_G^{\frac{1}{2}}T_\lambda^{-1}k(g,\cdot)\right\|_{\mathcal{H}}^2$$

$$=\left(\left\|T_G^{\frac{1}{2}}T_{G\lambda}^{-1}k(g,\cdot)\right\|_{\mathcal{H}}-\left\|T_G^{\frac{1}{2}}T_\lambda^{-1}k(g,\cdot)\right\|_{\mathcal{H}}\right)\left(\left\|T_G^{\frac{1}{2}}T_{G\lambda}^{-1}k(g,\cdot)\right\|_{\mathcal{H}}+\left\|T_G^{\frac{1}{2}}T_\lambda^{-1}k(g,\cdot)\right\|_{\mathcal{H}}\right)+\left\|T_G^{\frac{1}{2}}T_\lambda^{-1}k(g,\cdot)\right\|_{\mathcal{H}}^2$$

$$\geqslant-R_1\left(R_1+2\left\|T_G^{\frac{1}{2}}T_\lambda^{-1}k(g,\cdot)\right\|_{\mathcal{H}}\right)+\left\|T_G^{\frac{1}{2}}T_\lambda^{-1}k(g,\cdot)\right\|_{\mathcal{H}}^2$$

$$=-R_1\left(R_1+2\sqrt{\frac{1}{nk}\sum_{i=1}^{n}\sum_{j=1}^{k}\left[T_\lambda^{-1}k(g_{ij},g)\right]^2}\right)+\frac{1}{nk}\sum_{i=1}^{n}\sum_{j=1}^{k}\left[T_\lambda^{-1}k(g_{ij},g)\right]^2$$

$$\geqslant\tilde{\Omega}_{\mathbb{P}}\left(\left\|T_\lambda^{-1}k(g,\cdot)\right\|_{L^2}^2\right).$$

Therefore, with high probability, by Corollary C.2,

$$\mathbb{E}_{\epsilon_{1,1}|g_{1,1}}\mathbb{E}_{\epsilon_{1,2}|g_{1,2}}\ldots\mathbb{E}_{\epsilon_{n,k}|g_{n,k}}\mathbb{E}_{\epsilon_{ij}|g_{ij}}\mathrm{V}(n^{-\beta})\geqslant\sigma_L^2\Omega_{\mathbb{P}}^{\mathrm{poly}}\left(\frac{n^{\alpha_0\beta}}{nk}\right)=\Omega_{\mathbb{P}}^{\mathrm{poly}}\left(\frac{\sigma_L^2}{k}\right).$$

$$\square$$

## A.4 Excess Risk Bounds in Denoising Score Learning

We prove Theorem 4.4 in this section. We rewrite it as the theorem as follow.

**Theorem A.6.** *Consider the denoising score learning at timestep $t$. Assuming that*

$$\mathbb{E}[x^2]\leqslant\sigma_x^2,\quad\mathbb{E}[\xi^2]\leqslant\sigma_\xi^2,$$

then if $\lambda \asymp n^{-\theta}, \theta \in (0, \beta)$ and the conditional orthogonality holds, under Assumption 1, the excess risk satisfies

$$R(\lambda) \leqslant \underbrace{\tilde{\Theta}_{\mathbb{P}}\left(n^{-\min(s,2)\theta}\right)}_{\text{Bias}^2(\lambda)} + \underbrace{\tilde{\sigma}^2 O_{\mathbb{P}}^{\text{poly}}\left(n^{\alpha_0\theta}\left(\frac{\alpha_t^{\frac{p}{2}} \vee r_e}{n} + \frac{(1 - \alpha_t^{\frac{p}{2}}) \wedge (1 - r_e)}{nk}\right)\right)}_{\text{Var}(\lambda)}.$$

*Proof.* We prove by simply computing $r_0$. In the setting of denoising score learning at timestep $t$,

$$g_{ij} = \sqrt{\alpha_t}x_i + \sqrt{1 - \alpha_t}\xi_{ij}.$$

Therefore,

$$r_0 = \left(\frac{1}{2} - \frac{\sum_{r=1}^d \text{Cov}\left(g_{ij}^{(r)}, g_{i'j}^{(r)}\right)}{2\sum_{r=1}^d \text{Var}\left(g_{ij}^{(r)}\right)}\right)^{\frac{p}{2}} = \left(\frac{\alpha_t\sigma_x^2}{2(1 - \alpha_t)\sigma_\xi^2 + 2\alpha_t\sigma_x^2}\right)^{\frac{p}{2}} = \alpha_t^{\frac{p}{2}}\Theta(1).$$

Hence, the proof is completed by applying Theorem 4.1. $\qquad\square$

## B  Concentration Lemmas

For simplicity, we sometimes use $\{g_i\}_{i=1}^{nk}$ to represent $\{g_{ij}\}_{i=1,j=1}^{n,k}$, where the first $k$ components $g_1, \ldots, g_k$ represent $g_{1,1}, \ldots, g_{1,k}$, the second $k$ components $g_{k+1}, \ldots, g_{2k}$ represent $g_{2,1}, \ldots, g_{2,k}$, and iteratively, $g_{(n-1)k+1}, \ldots, g_{nk}$ represent $g_{n,1}, \ldots, g_{n,k}$. We always ignore the script $r$ in this section.

**Lemma B.1.** *Under Assumption 1, if $\lambda \asymp n^{-\theta}, \theta \in (0, \beta)$, such that for $\alpha > \alpha_0$ being sufficiently close, with high probability,*

$$\left\|T_\lambda^{-\frac{1}{2}}(T - T_G)T_\lambda^{-\frac{1}{2}}\right\| \leqslant \tilde{O}_{\mathbb{P}}\left(\sqrt{\frac{\lambda^{-\alpha}}{n}}\right).$$

*Further,*

$$\left\|T_{G\lambda}^{-\frac{1}{2}}T_\lambda^{\frac{1}{2}}\right\|^2 = \left\|T_\lambda^{\frac{1}{2}}T_{G\lambda}^{-1}T_\lambda^{\frac{1}{2}}\right\| \leqslant O_{\mathbb{P}}(1).$$

*Proof.* The proof is standard in concentration inequalities, while we utilize our novel Bernstein-type bound, i.e., Proposition D.2, to handle the $k$-gap independent random sequence. Denote $A(g) = T_\lambda^{-\frac{1}{2}}(T - T_g)T_\lambda^{-\frac{1}{2}}$, $E[A(g)] = 0$, then $T_\lambda^{-\frac{1}{2}}(T - T_G)T_\lambda^{-\frac{1}{2}} = \frac{1}{nk}\sum_{i=1}^{nk} A(g_i)$. For simplicity, we denote $A_i := A(g_i)$. As the first step, for any positive $x, t$,

$$\mathbb{P}\left(\lambda_{\max}\left(\sum_{i=1}^{nk} A_i\right) \geqslant x\right) \leqslant \frac{1}{e^{tx} - tx - 1}\mathbb{E}\text{tr}\left(\exp\left(t\sum_{i=1}^{nk} A_i\right) - \mathbf{I}\right),$$

where $\mathbf{I}$ is the identity. Then we prove by Proposition D.2. We first bound $\|A(g)\|$ by Corollary C.2,

$$\left\|T_\lambda^{-\frac{1}{2}}T_gT_\lambda^{-\frac{1}{2}}\right\| = \left\|T_\lambda^{-\frac{1}{2}}k(g, \cdot)\right\|_{\mathcal{H}}^2 \leqslant M_\alpha^2\lambda^{-\alpha},$$

which implies that

$$\|A(g)\| \leqslant 2M_\alpha^2\lambda^{-\alpha}.$$

Therefore, by Proposition D.2, for any positive $t$ such that $t \cdot 2M_\alpha^2\lambda^{-\alpha} < \frac{1}{k}\frac{1}{\log n}$,

$$\log\mathbb{E}\text{tr}\left(\exp\left(t\sum_{i=1}^{nk} A_i\right) - \mathbf{I}\right) \leqslant \log n\log 3 + \log\left(\frac{nk}{2}\text{intd}\right) + t^2nkv^2\frac{169}{1 - t \cdot 2M_\alpha^2\lambda^{-\alpha}\log n},$$

where

$$v^2 = \sup_{K \subseteq \{1,\ldots,nk\}}\frac{1}{\text{Card}K}\lambda_{\max}\left(\mathbb{E}\left[\left(\sum_{i \in K} A(g_i)\right)^2\right]\right), \quad \text{intd} = \text{intdim}\left(\mathbb{E}\left[A(g)^2\right]\right).$$

Hence, for any positive $x$,

$$\mathbb{P}\left(\lambda_{\max}\left(\sum_{i=1}^{nk} A_i\right) \geq x\right)$$

$$\leq \inf_{t>0:t2M_\alpha^2\lambda^{-\alpha}<\frac{1}{k}\frac{1}{\log n}} \frac{\exp\left(\log n \log 3 + \log\left(\frac{nk}{2}\text{intd}\right) + t^2 nkv^2 \frac{169}{1-t\cdot 2M_\alpha^2\lambda^{-\alpha}\log n}\right)}{e^{tx}-tx-1}$$

$$\leq \exp\left(\log n \log 3 + \log\left(\frac{nk}{2}\text{intd}\right)\right) \inf_{t>0:t2M_\alpha^2\lambda^{-\alpha}<\frac{1}{k}\frac{1}{\log n}} \left(1 + \frac{3}{x^2 t^2}\right) \exp\left(-tx + \frac{169t^2 nkv^2}{1-t\cdot 2M_\alpha^2\lambda^{-\alpha}k\log n}\right),$$

where the last inequality holds by the basic inequality:

$$\frac{1}{e^x - x - 1} \leq \left(1 + \frac{3}{x^2}\right) e^{-x}, \quad x > 0.$$

As the second step, we select $t$. Denote $\theta = 169nkv^2, \phi = 2M_\alpha^2\lambda^{-\alpha}k\log n$ and let $t = \frac{x}{2\theta+\phi x} < \frac{1}{\phi}$ then

$$\mathbb{P}\left(\lambda_{\max}\left(\sum_{i=1}^{nk} A_i\right) \geq x\right) \leq \exp\left(\log n \log 3 + \log\left(\frac{nk}{2}\text{intd}\right)\right)\left(1 + 3\frac{(2\theta+\phi x)^2}{x^4}\right)\exp\left(-\frac{x^2/2}{2\theta+\phi x}\right).$$

If $x^2 \geq 2\theta + \phi x$ then

$$\mathbb{P}\left(\lambda_{\max}\left(\sum_{i=1}^{nk} A_i\right) \geq x\right) \leq 4\exp\left(\log n \log 3 + \log\left(\frac{nk}{2}\text{intd}\right)\right)\exp\left(-\frac{x^2/2}{2\theta+\phi x}\right).$$

Therefore,

$$\mathbb{P}\left(\left\|\frac{1}{nk}\sum_{i=1}^{nk} A_i\right\| \geq \frac{1}{nk}x\right) \leq 4\exp\left(\log n \log 3 + \log\left(\frac{nk}{2}\text{intd}\right)\right)\exp\left(-\frac{x^2/2}{2\theta+\phi x}\right).$$

By setting $\delta = 4\exp\left(\log n \log 3 + \log\left(\frac{nk}{2}\text{intd}\right)\right)\exp\left(-\frac{x^2/2}{2\theta+\phi x}\right)$ and solving

$$x \leq 2\phi\log\left(\frac{4\exp\left(\log n \log 3 + \log\left(\frac{nk}{2}\text{intd}\right)\right)}{\delta}\right) + \sqrt{4\theta\log\left(\frac{4\exp\left(\log n \log 3 + \log\left(\frac{nk}{2}\text{intd}\right)\right)}{\delta}\right)},$$

we have, with high probability,

$$\left\|\frac{1}{nk}\sum_{i=1}^{nk} A_i\right\| \leq \frac{2\phi\log\left(\frac{4\exp\left(\log n \log 3 + \log\left(\frac{nk}{2}\text{intd}\right)\right)}{\delta}\right)}{nk} + \left(\frac{4\theta\log\left(\frac{4\exp\left(\log n \log 3 + \log\left(\frac{nk}{2}\text{intd}\right)\right)}{\delta}\right)}{n^2k^2}\right)^{1/2}.$$

We next consider the bound for $\dfrac{4\theta\log\left(\frac{4\exp\left(\log n \log 3 + \log\left(\frac{nk}{2}\text{intd}\right)\right)}{\delta}\right)}{n^2k^2}$. Note that

$$v^2 = \sup_{K\subseteq\{1,\ldots,k\}} \frac{1}{\text{Card}K}\lambda_{\max}\left(\mathbb{E}\left[\left(\sum_{i\in K} A(g_i)\right)^2\right]\right)$$

$$\leq \sup_{K\subseteq\{1,\ldots,k\}} \text{Card}K \cdot \lambda_{\max}\left(\mathbb{E}\left[A(g_i)^2\right]\right)$$

$$\leq k\left\|\mathbb{E}\left[A(g_i)^2\right]\right\|,$$

further note that

$$\left\|\mathbb{E}\left[A(g)^2\right]\right\|\log\frac{\text{tr}\left(\mathbb{E}\left[A(g)^2\right]\right)}{\left\|\mathbb{E}\left[A(g)^2\right]\right\|}$$

increases monotonically with respect to $\mathbb{E}\left[A(g)^2\right]$, and by Corollary C.2,

$$\mathbb{E}\left[A(g)^2\right] \leq \mathbb{E}\left[\left(T_\lambda^{-\frac{1}{2}} T_g T_\lambda^{-\frac{1}{2}}\right)^2\right] \leq M_\alpha^2 \lambda^{-\alpha} \mathbb{E}\left[T_\lambda^{-\frac{1}{2}} T_g T_\lambda^{-\frac{1}{2}}\right] = M_\alpha^2 \lambda^{-\alpha} T T_\lambda^{-1},$$

we have

$$\frac{4\theta \log\left(\frac{4\exp\left(\log n \log 3 + \log\left(\frac{nk}{2}\mathrm{intd}\right)\right)}{\delta}\right)}{n^2 k^2} = \frac{676 v^2 \log\left(\frac{4\exp\left(\log n \log 3 + \log\left(\frac{nk\,\mathrm{tr}\left(\mathbb{E}\left[A(g)^2\right]\right)}{2\|\mathbb{E}\left[A(g)^2\right]\|}\right)\right)}{\delta}\right)}{nk}$$

$$\leq \frac{676 M_\alpha^2 \lambda^{-\alpha} \log\left(\frac{4\exp\left(\log n \log 3 + \log\left(\frac{nk\,\mathrm{tr}\left(TT_\lambda^{-1}\right)}{2\|TT_\lambda^{-1}\|}\right)\right)}{\delta}\right)}{n}.$$

By Lemma C.3,

$$\frac{\mathrm{tr}\left(TT_\lambda^{-1}\right)}{\|TT_\lambda^{-1}\|} \leq O\left(\frac{\lambda^{-\alpha_0}}{\|TT_\lambda^{-1}\|}\right) = O\left(\frac{(\|T\|+\lambda)\lambda^{-\alpha_0}}{\|T\|}\right) \leq O\left(\lambda^{-\alpha-\alpha_0}\right).$$

Therefore,

$$\frac{4\theta \log\left(\frac{4\exp\left(\log n \log 3 + \log\left(\frac{nk}{2}\mathrm{intd}\right)\right)}{\delta}\right)}{n^2 k^2} \leq \tilde{O}\left(\frac{\lambda^{-\alpha}}{n}\right).$$

We finally obtain that for $\alpha > \alpha_0$,

$$\left\|\frac{1}{nk}\sum_{i=1}^{nk} A_i\right\| \leq \tilde{O}_\mathbb{P}\left(\sqrt{\frac{\lambda^{-\alpha}}{n}}\right).$$

Further, we have

$$\left\|T_\lambda^{1/2}(T_G + \lambda)^{-1}T_\lambda^{1/2}\right\| = \left\|\left\{T_\lambda^{-1/2}(T_G + \lambda)T_\lambda^{-1/2}\right\}^{-1}\right\|$$

$$= \left\|\left\{I - T_\lambda^{-1/2}(T - T_G)T_\lambda^{-1/2}\right\}^{-1}\right\|$$

$$\leq \left(1 - \left\|T_\lambda^{-\frac{1}{2}}(T - T_G)T_\lambda^{-\frac{1}{2}}\right\|\right)^{-1}$$

$$\leq O_\mathbb{P}(1).$$

$\square$

**Lemma B.2.** *Under Assumption 1, if $\lambda \asymp n^{-\theta}, \theta \in (0, \beta)$,*

$$\left\|T_\lambda^{-\frac{1}{2}}\left[(T_G f_\rho^* - T_G f_\lambda) - (T f_\rho^* - T f_\lambda)\right]\right\|_\mathcal{H} \leq \tilde{o}_\mathbb{P}\left(\lambda^{\frac{\tilde{s}}{2}}\right).$$

*Proof.* The proof is standard in concentration inequalities, while we utilize our novel Bernstein-type bound, i.e., Proposition D.3, to handle the $k$-gap independent random sequence. Denote $\xi(g) = T_\lambda^{-\frac{1}{2}}(T_g f_\rho^* - T_g f_\lambda), \xi_i = \xi(g_i)$, then

$$\left\|T_\lambda^{-\frac{1}{2}}\left[(T_G f_\rho^* - T_G f_\lambda) - (T f_\rho^* - T f_\lambda)\right]\right\|_\mathcal{H} = \left\|\frac{1}{nk}\sum_{i=1}^{nk}\xi_i - \mathbb{E}\xi\right\|_\mathcal{H}.$$

Further let $X = \xi - \mathbb{E}\xi, \ X_i = \xi_i - \mathbb{E}\xi_i$, then

$$\left\|T_\lambda^{-\frac{1}{2}}\left[(T_G f_\rho^* - T_G f_\lambda) - (T f_\rho^* - T f_\lambda)\right]\right\|_\mathcal{H} = \left\|\frac{1}{nk}\sum_{i=1}^{nk}X_i\right\|_\mathcal{H}.$$

For any positive $x, \theta$ we have

$$\mathbb{P}\left(\left\|\sum_{i=1}^{nk} X_i\right\|_{\mathcal{H}} \geqslant x\right) \leqslant e^{-\theta x}\mathbb{E}\left[e^{\theta\left\|\sum_{i=1}^{nk} X_i\right\|_{\mathcal{H}}}\right]$$

$$\leqslant 2e^{-\theta x}\mathbb{E}\cosh\left(\theta\left\|\sum_{i=1}^{nk} X_i\right\|_{\mathcal{H}}\right)$$

$$\overset{(*)}{\leqslant} 2e^{-\theta x}\mathbb{E}\prod_{i=1}^{nk} e^{\theta\|X_i\|_{\mathcal{H}}} - \theta\|X_i\|_{\mathcal{H}} \qquad (B.1)$$

$$\leqslant 2e^{-\theta x}\mathbb{E}\prod_{i=1}^{nk} e^{\theta\|X_i\|_{\mathcal{H}}},$$

where (*) is the result of Lemma C.8. Then we are going to apply Proposition D.3. Similar to [37], we separate to two cases: $s > \alpha_0$ and $s \leqslant \alpha_0$, where we use truncation technique to handle the more difficult $s \leqslant \alpha_0$ case.

Firstly, we consider the case $s > \alpha_0$. By Lemma C.7, for $\alpha > \alpha_0$ being sufficiently close,

$$\|X\|_{\mathcal{H}} \leqslant \tilde{O}\left(\lambda^{-\alpha+\frac{\tilde{s}}{2}}\right) := M_X.$$

Hence, by Proposition D.3, there exists a constant $C''$ such that for any positive $t$ such that $tM_X < \frac{1}{k\log n}$,

$$\log\mathbb{E}\exp\left(t\sum_{i=1}^{nk} X_i\right) \leqslant \frac{C''t^2 nkv^2}{1 - tM_X k\log n},$$

where

$$v^2 = \sup_{K\subseteq\{1,\dots,k\}}\frac{1}{\mathrm{Card}K}\mathbb{E}\left[\left(\sum_{j\in K}\|X_j\|_{\mathcal{H}}\right)^2\right].$$

Apply into (B.1) and take $\theta = \frac{x}{2C''nkv^2 + x\cdot C'' M_X k\log nk}$ we have

$$\mathbb{P}\left(\left\|\sum_{i=1}^{nk} X_i\right\|_{\mathcal{H}} \geqslant x\right) \leqslant 2e^{-\theta x}\mathbb{E}\prod_{i=1}^{nk} e^{\theta\|X_i\|_{\mathcal{H}}}$$

$$\leqslant 2e^{-\theta x}\exp\left(\frac{C''\theta^2 nkv^2}{1 - \theta M_X k\log n}\right)$$

$$= 2\exp\left(-\frac{x^2/2}{2C''nkv^2 + x\cdot M_X k\log n}\right)$$

$$:= 2\exp\left(-\frac{x^2/2}{U + xV}\right),$$

where $U = 2C''nkv^2, V = M_X k\log n$. Let $\delta = 2\exp\left(-\frac{x^2/2}{U+xV}\right)$ we have

$$x \leqslant 2V\log\frac{2}{\delta} + \sqrt{2U\log\frac{2}{\delta}} = O_{\mathbb{P}}\left(M_X k\log n + \sqrt{nkv^2}\right).$$

Then with high probability,

$$\left\|\frac{1}{nk}\sum_{i=1}^{nk} X_i\right\|_{\mathcal{H}} \leqslant O_{\mathbb{P}}\left(\frac{M_X\log n}{n} + \sqrt{\frac{v^2}{nk}}\right) \leqslant \tilde{O}_{\mathbb{P}}\left(\lambda^{\frac{\tilde{s}}{2}}\frac{\lambda^{-\alpha}\log n}{n} + \sqrt{\frac{v^2}{nk}}\right). \qquad (B.2)$$

Secondly, we consider the case $s \leqslant \alpha_0$. For any $t > 0$, denote $\Omega_t = \{g \in \mathcal{G} : |f_\rho^*(g)| \leqslant t\}$ and $\bar{\xi}(g) = \xi(g)\mathbf{1}_{\{g\in\Omega_t\}}$, $\bar{X} = \bar{\xi} - \mathbb{E}\bar{\xi}$, $\bar{X}_i = \bar{\xi}_i - \mathbb{E}\bar{\xi}$. Then similar to Lemma C.7, for $\alpha > \alpha_0$ being

sufficiently close,

$$\|\bar{X}\|_{\mathcal{H}} \leqslant M_\alpha \lambda^{-\frac{\alpha}{2}} \|\mathbf{1}_{\{g \in \Omega_t\}}(f_\rho^* - f_\lambda)\|_{L^\infty}$$
$$\leqslant M_\alpha \lambda^{-\frac{\alpha}{2}} \left(\|f_\lambda\|_{L^\infty} + t\right)$$
$$\leqslant M_\alpha \lambda^{-\frac{\alpha}{2}} \left(M_\alpha \|f_\lambda\|_{[\mathcal{H}]^\alpha} + t\right)$$
$$\leqslant \tilde{O}\left(\lambda^{-\alpha + \frac{s}{2}} + t\lambda^{-\frac{\alpha}{2}}\right) := M_X,$$

where the last inequality uses Lemma C.5. Further we decompose

$$\left\|\frac{1}{nk}\sum_{i=1}^{nk}\xi_i - \mathbb{E}\xi\right\|_{\mathcal{H}} \leqslant \left\|\frac{1}{nk}\sum_{i=1}^{nk}\bar{\xi}_i - \mathbb{E}\bar{\xi}\right\|_{\mathcal{H}} + \left\|\frac{1}{nk}\sum_{i=1}^{nk}\xi_i \mathbf{1}_{\{g_i \notin \Omega_t\}}\right\|_{\mathcal{H}} + \left\|\mathbb{E}\xi\mathbf{1}_{\{g \notin \Omega_t\}}\right\|_{\mathcal{H}}. \tag{B.3}$$

Regarding the first term in (B.3), we set $t \asymp n^l, l < 1 - \frac{\alpha+s}{2}\theta < \frac{\alpha-s}{2}\theta$, then, similar to (B.2), with high probability,

$$\left\|\frac{1}{nk}\sum_{i=1}^{nk}\bar{\xi}_i - \mathbb{E}\bar{\xi}\right\|_{\mathcal{H}} \leqslant O_{\mathbb{P}}\left(\frac{M_X \log n}{n} + \sqrt{\frac{\bar{v}^2}{nk}}\right) \leqslant \tilde{O}_{\mathbb{P}}\left(\lambda^{\frac{s}{2}}\frac{\lambda^{-\alpha}\log n}{n} + \sqrt{\frac{\bar{v}^2}{nk}}\right),$$

where

$$\bar{v}^2 = \sup_{K \subseteq \{1,\dots,k\}} \frac{1}{\mathrm{Card}K}\mathbb{E}\left[\left(\sum_{j \in K}\|\bar{X}_j\|_{\mathcal{H}}\right)^2\right].$$

To bound the second term in (B.3), we only need to consider the case $g_i \notin \Omega_t$. Since the Markov's inequality yields

$$\mathbb{P}_{g \sim \mu_{\mathcal{G}}}\left(g \notin \Omega_t\right) \leqslant t^{-q}\|f_\rho^*\|_{L^q}^q,$$

where $q = \frac{2\alpha}{\alpha-s}$ (referring to Lemma C.11), we have

$$\mathbb{P}\left(g_{i,1} \notin \Omega_t, g_{i,2} \notin \Omega_t, \dots, g_{i,k} \notin \Omega_t\right) \leqslant \mathbb{P}\left(g_{i,1} \notin \Omega_t\right) \leqslant t^{-q}\|f_\rho^*\|_{L^q}^q.$$

Then we get

$$\mathbb{P}\left(g_i \in \Omega_t, \forall i\right) \geqslant \left(1 - t^{-q}\|f_\rho^*\|_{L^q}^q\right)^n.$$

So the second vanishes with high probability as long as $l > \frac{1}{q}$.

For the third term in (B.3),

$$\left\|\mathbb{E}\xi(g)\mathbf{1}_{\{g \notin \Omega_t\}}\right\|_{\mathcal{H}} \leqslant \mathbb{E}\|\xi(g)\mathbf{1}_{\{g \notin \Omega_t\}}\|_{\mathcal{H}}$$
$$= \mathbb{E}\left[\mathbf{1}_{\{g \notin \Omega_t\}}\left(f_\rho^*(g) - f_\lambda(g)\right)\left\|T_\lambda^{-1/2}k(g,\cdot)\right\|_{\mathcal{H}}\right]$$
$$\leqslant M_\alpha \lambda^{-\alpha/2}\mathbb{E}\left[\mathbf{1}_{\{x \notin \Omega_t\}}|f_\rho^*(g) - f_\lambda(g)|\right]$$
$$\leqslant M_\alpha \lambda^{-\alpha/2}\mathbb{E}\left[(f_\rho^*(g) - f_\lambda(g))^2\right]^{\frac{1}{2}}\left[\mathbb{P}\{g \notin \Omega_t\}\right]^{\frac{1}{2}}$$
$$\leqslant M_\alpha \lambda^{-\alpha/2}\tilde{\Theta}\left(\lambda^{\tilde{s}/2}\right)t^{-q/2}\|f_\rho^*\|_{L^q}^{q/2},$$

where the second inequality holds by Corollary C.2, and the last inequality uses Lemma C.4. If $l > \frac{\alpha\theta}{q}$, then

$$\left\|\mathbb{E}\xi(g)\mathbf{1}_{\{g \notin \Omega_t\}}\right\|_{\mathcal{H}} \leqslant \tilde{o}\left(\frac{\lambda^{-\frac{\alpha}{2}}}{\sqrt{n}}\lambda^{\frac{\tilde{s}}{2}}\right).$$

Finally, the three requirements of $l$ are

$$l < 1 - \frac{\alpha+s}{2}\theta, \quad l > \frac{1}{q}, \quad \text{and} \quad l > \frac{\theta\alpha}{q},$$

where $q = \frac{2\alpha}{\alpha-s}$. It is easy to verify these three requirements hold.

Combine the two cases, with high probability,

$$\left\| \frac{1}{nk} \sum_{i=1}^{nk} X_i \right\|_{\mathcal{H}} \leqslant \tilde{O}_{\mathbb{P}} \left( \lambda^{\frac{\tilde{s}}{2}} \frac{\lambda^{-\alpha}}{n} + \sqrt{\frac{v^2}{nk}} \right) + \tilde{o} \left( \frac{\lambda^{-\frac{\alpha}{2}}}{\sqrt{n}} \lambda^{\frac{\tilde{s}}{2}} \right),$$

where

$$v^2 = \sup_{K \subseteq \{1,\dots,k\}} \frac{1}{\mathrm{Card}K} \mathbb{E} \left[ \left( \sum_{j \in K} \|X_j\|_{\mathcal{H}} \right)^2 \right].$$

The last thing is to handle $v^2$. It is easy to verify that

$$v^2 \leqslant \mathrm{Card}K \cdot \mathbb{E} \|X\|_{\mathcal{H}}^2 \leqslant k \mathbb{E} \|X\|_{\mathcal{H}}^2.$$

Note that

$$\mathbb{E} \|X\|_{\mathcal{H}}^2 \leqslant \mathbb{E} \|\xi\|_{\mathcal{H}}^2 = \sup_g \left\| T_\lambda^{-\frac{1}{2}} k(g, \cdot) \right\|_{\mathcal{H}}^2 \mathbb{E} \left[ \left( f_\rho^*(g) - f_\lambda(g) \right)^2 \right] \leqslant M_\alpha^2 \lambda^{-\alpha} \mathbb{E} \left[ \left( f_\rho^*(g) - f_\lambda(g) \right)^2 \right],$$

then by Lemma C.4, we have

$$\mathbb{E} \left[ \left( f_\rho^*(g) - f_\lambda(g) \right)^2 \right] = \tilde{\Theta}(\lambda^{\tilde{s}}).$$

Hence,

$$v^2 \leqslant k \tilde{O}(\lambda^{-\alpha} \lambda^{\tilde{s}}).$$

Therefore,

$$\left\| T_\lambda^{-\frac{1}{2}} \left[ (T_G f_\rho^* - T_G f_\lambda) - (T f_\rho^* - T f_\lambda) \right] \right\|_{\mathcal{H}} \leqslant \tilde{o}_{\mathbb{P}} \left( \lambda^{\frac{\tilde{s}}{2}} \right).$$

$\square$

**Lemma B.3.** *Under Assumption 1, if $\lambda \asymp n^{-\theta}, \theta \in (0, \beta)$, for $\alpha > \alpha_0$ being sufficiently close, with high probability, for $g \in \mathcal{G}$ almost everywhere,*

$$\left| \frac{1}{nk} \sum_{i=1}^n \sum_{j=1}^k \left[ T_\lambda^{-1} k(g_{ij}, g) \right]^2 - \left\| T_\lambda^{-1} k(g, \cdot) \right\|_{L^2}^2 \right| \leqslant \tilde{O}_{\mathbb{P}} \left( \lambda^{-\alpha} \sqrt{\frac{\lambda^{-\alpha}}{n}} \right).$$

*Proof.* The proof is standard in concentration inequalities, while we use the net theory to obtain a union high probability bound as Li et al. [37, 39] and utilize our novel Bernstein-type bound, i.e., Proposition D.3, to handle the $k$-gap independent random sequence. Denote $X_{ij} = \left[ T_\lambda^{-1} k(g_{ij}, g) \right]^2$, with its expectation over each data $g_{ij}$

$$\mathbb{E} \left[ T_\lambda^{-1} k(g_{ij}, g) \right]^2 = \| T_\lambda^{-1} k(\cdot, g) \|_{L^2}^2 := \mu.$$

For any positive $s, \epsilon$,

$$\mathbb{P} \left[ \left| \frac{1}{nk} \sum_{i=1}^n \sum_{j=1}^k X_{ij} - \| T_\lambda^{-1} k(\cdot, g) \|_{L^2}^2 \right| \geqslant \epsilon \right] \leqslant 2e^{-s\epsilon} \mathbb{E} \exp \left( \frac{s}{nk} \sum_{i=1}^n \sum_{j=1}^k (X_{ij} - \mu) \right).$$

We next prove by applying Proposition D.3. Note that by Corollary C.2,

$$|X_{ij}| \leqslant \left\| T_\lambda^{-1} k(g, \cdot) \right\|_{L^\infty}^2 \leqslant M_\alpha^4 \lambda^{-2\alpha} := B,$$

then by Proposition D.3, for any $0 < s < \frac{nk}{2Bk \log n}$

$$\mathbb{E} \exp \left( \frac{s}{nk} \sum_{i=1}^n \sum_{j=1}^k (X_{ij} - \mu) \right) \leqslant \exp \left( \frac{C'' s^2 nk v^2}{n^2 k^2 \left( 1 - \frac{s}{nk} 2Bk \log n \right)} \right),$$

where $v^2 = \sup_{K \subseteq \{1,\dots,k\}} \frac{1}{\mathrm{Card}K} \mathbb{E} \left[ \left( \sum_{j \in K} (X_{ij} - \mu) \right)^2 \right]$. Then by setting $s = \frac{\epsilon nk}{2C'' v^2 + 2Bk \log n\epsilon}$,

$$\mathbb{P} \left[ \left| \frac{1}{nk} \sum_{i=1}^n \sum_{j=1}^k X_{ij} - \| T_\lambda^{-1} k(\cdot, g) \|_{L^2}^2 \right| \geqslant \epsilon \right] \leqslant 2e^{-s\epsilon} \exp \left( \frac{C'' s^2 nk v^2}{n^2 k^2 \left( 1 - \frac{s}{nk} 2Bk \log n \right)} \right)$$

$$\leqslant 2 \exp \left( - \frac{\epsilon^2 nk}{4C'' v^2 + 4Bk \log n\epsilon} \right).$$

Let $\delta = 2\exp\left(-\frac{\epsilon^2 nk}{4C''v^2 + 4Bk\log n\epsilon}\right)$ and solve $\epsilon$ we obtain

$$\epsilon \leqslant \frac{4\log\frac{2}{\delta}B\log n}{n} + \sqrt{\frac{16C''v^2}{nk}\log\frac{2}{\delta}}.$$

For the reason that we are considering the union bound for any $g \in \mathcal{G}$, we denote $\mathcal{K}_\lambda = \left\{T_\lambda^{-1}k(g,\cdot)\right\}_{g\in\mathcal{G}}$ and utilize the net theory. By Lemma C.9, we can find an $\epsilon$-net $\mathcal{F} \subset \mathcal{K}_\lambda \subset \mathcal{H}$ such that

$$|\mathcal{F}| \leqslant C'''(\lambda\varepsilon)^{-\frac{2d}{p}},$$

where $\varepsilon = \varepsilon(n) = \frac{1}{n}$. Then, with probability at least $1 - \delta$, $\forall f \in \mathcal{F}$ [7]

$$\left|\frac{1}{nk}\sum_{i=1}^{n}\sum_{j=1}^{k}X_{ij} - \|T_\lambda^{-1}k(\cdot,g)\|_{L^2}^2\right| \leqslant \frac{4\log\frac{2|\mathcal{F}|}{\delta}B\log n}{n} + \sqrt{\frac{16C''v^2}{nk}\log\frac{2|\mathcal{F}|}{\delta}}.$$

Note that by Corollary C.2, we have

$$v^2 = \sup_{K\subseteq\{1,\dots,k\}}\frac{1}{\mathrm{Card}K}\mathbb{E}\left[\left(\sum_{j\in K}(X_{ij}-\mu)\right)^2\right] \leqslant kB\left\|T_\lambda^{-1}k(\cdot,g)\right\|_{L^2}^2,$$

which implies that

$$\sqrt{\frac{\lambda^{-\alpha}}{n}}\sqrt{\frac{v^2}{nk}} = O\left(\frac{\lambda^{-2\alpha}}{n}\right).$$

Hence with high probability, $\forall f \in \mathcal{F}$

$$\left|\frac{1}{nk}\sum_{i=1}^{n}\sum_{j=1}^{k}X_{ij} - \|T_\lambda^{-1}k(\cdot,g)\|_{L^2}^2\right| \leqslant \tilde{O}_\mathbb{P}\left(\lambda^{-\alpha}\sqrt{\frac{\lambda^{-\alpha}}{n}}\right).$$

At last, by the definition of $\mathcal{F}$, for any $g \in \mathcal{G}$, there exists some $f \in \mathcal{F}$, such that

$$\left\|T_\lambda^{-1}k(g,\cdot) - f\right\|_{L^\infty} \leqslant \varepsilon,$$

which implies that

$$\left|\left\|T_\lambda^{-1}k(\cdot,g)\right\|_{L^2}^2 - \|f\|_{L^2}^2\right| \leqslant \varepsilon O(\lambda^{-\alpha}), \quad \left|\frac{1}{nk}\sum_{i=1}^{n}\sum_{j=1}^{k}X_{ij} - \frac{1}{nk}\sum_{i=1}^{n}\sum_{j=1}^{k}f^2(g_{ij})\right| \leqslant \varepsilon O(\lambda^{-\alpha}).$$

Therefore, for $\alpha > \alpha_0$ being sufficiently close, with high probability, $\forall g \in \mathcal{G}$ almost everywhere,

$$\left|\frac{1}{nk}\sum_{i=1}^{n}\sum_{j=1}^{k}X_{ij} - \|T_\lambda^{-1}k(\cdot,g)\|_{L^2}^2\right| \leqslant \tilde{O}_\mathbb{P}\left(\lambda^{-\alpha}\sqrt{\frac{\lambda^{-\alpha}}{n}}\right).$$

$\square$

**Lemma B.4.** *Under Assumption 1, if $\lambda \asymp n^{-\theta}, \theta \in (0,\beta)$, for $\alpha > \alpha_0$ being sufficiently close, with high probability, for $g \in \mathcal{G}$ almost everywhere,*

$$\left|\frac{1}{nk}\sum_{i=1}^{n}\sum_{j=1}^{k}T_\lambda^{-1}k(g,g_{ij})\epsilon_{ij}\right| \leqslant \tilde{O}_\mathbb{P}\left(\sqrt{\frac{\lambda^{-\alpha}}{n}\sigma_\epsilon^2}\right).$$

*Proof.* The proof is mainly based on the fact that $\epsilon_{ij}|g_{ij}$ is sub-Gaussian with norm $\sigma_\epsilon$.

$$\mathbb{P}\left(\left|\frac{1}{nk}\sum_{i=1}^{n}\sum_{j=1}^{k}T_\lambda^{-1}k(g,g_{ij})\epsilon_{ij}|g_{ij}\right| \geqslant t\right) \leqslant 2\exp\left(-\frac{t^2}{\left\|\frac{1}{nk}\sum_{i=1}^{n}\sum_{j=1}^{k}T_\lambda^{-1}k(g,g_{ij})\epsilon_{ij}|g_{ij}\right\|_{\psi_2}^2}\right).$$

---

[7] Here $\forall f \in \mathcal{F}$ means $\forall g \in \mathcal{G}$ such that $T_\lambda^{-1}k(g,\cdot) \in \mathcal{F}$.

Hence, consider the net $\mathcal{F}$ constructed in Lemma B.3, with high probability, $\forall f \in \mathcal{F}$,

$$\left| \frac{1}{nk} \sum_{i=1}^n \sum_{j=1}^k T_\lambda^{-1} k(g, g_{ij}) \epsilon_{ij} \right| \leqslant \tilde{O}_{\mathbb{P}} \left( \left\| \frac{1}{nk} \sum_{i=1}^n \sum_{j=1}^k T_\lambda^{-1} k(g, g_{ij}) \epsilon_{ij} | g_{ij} \right\|_{\psi_2} \right)$$

$$\leqslant \tilde{O}_{\mathbb{P}} \left( \frac{1}{nk} \sqrt{ \sum_{i=1}^n \left\| \sum_{j=1}^k T_\lambda^{-1} k(g, g_{ij}) \epsilon_{ij} | g_{ij} \right\|_{\psi_2}^2 } \right)$$

$$\leqslant \tilde{O}_{\mathbb{P}} \left( \frac{1}{nk} \sqrt{ \sum_{i=1}^n \left( \sum_{j=1}^k \left\| T_\lambda^{-1} k(g, g_{ij}) \epsilon_{ij} | g_{ij} \right\|_{\psi_2} \right)^2 } \right)$$

$$\leqslant \tilde{O}_{\mathbb{P}} \left( \frac{1}{nk} \sqrt{ \sum_{i=1}^n \left( \sum_{j=1}^k \left| T_\lambda^{-1} k(g, g_{ij}) \right| \left\| \epsilon_{ij} | g_{ij} \right\|_{\psi_2} \right)^2 } \right)$$

$$\leqslant \tilde{O}_{\mathbb{P}} \left( \sqrt{ \frac{\sigma_\epsilon^2}{n} \frac{1}{nk} \sum_{i=1}^n \sum_{j=1}^k \left[ T_\lambda^{-1} k(g, g_{ij}) \right]^2 } \right).$$

Further, $\forall g \in \mathcal{G}$, there exists $f \in \mathcal{F}$ such that

$$\left| T_\lambda^{-1} k(g, g_{ij}) - f(g_{ij}) \right|_{L^\infty} \leqslant \varepsilon = \frac{1}{n}.$$

Hence,

$$\left| \frac{1}{nk} \sum_{i=1}^n \sum_{j=1}^k \left[ T_\lambda^{-1} k(g, g_{ij}) - f(g_{ij}) \right] \epsilon_{ij} \right| \leqslant \frac{1}{n} \frac{1}{nk} \sum_{i=1}^n \sum_{j=1}^k |\epsilon_{ij}|.$$

Note that $\epsilon_{ij} | g_{ij}$ is $\sigma_\epsilon^2$ sub-Gaussian, then

$$\mathbb{P} \left( \sum_{i=1}^n \sum_{j=1}^k |\epsilon_{ij}| \geqslant t \right) \leqslant \exp \left( - \frac{t^2}{2nk^2 \sigma_\epsilon^2} \right).$$

Hence, with high probability,

$$\frac{1}{nk} \sum_{i=1}^n \sum_{j=1}^k |\epsilon_{ij}| \leqslant O_{\mathbb{P}} \left( \sqrt{\frac{\sigma_\epsilon^2}{n}} \right),$$

which implies that

$$\left| \frac{1}{nk} \sum_{i=1}^n \sum_{j=1}^k \left[ T_\lambda^{-1} k(g, g_{ij}) - f(g_{ij}) \right] \epsilon_{ij} \right| \leqslant \frac{1}{n} \frac{1}{nk} \sum_{i=1}^n \sum_{j=1}^k |\epsilon_{ij}| \leqslant O_{\mathbb{P}} \left( \sqrt{\frac{\sigma_\epsilon^2}{n} \frac{1}{n}} \right).$$

Therefore, for $\alpha > \alpha_0$ being sufficiently close, with high probability for $g \in \mathcal{G}$ almost everywhere,

$$\left| \frac{1}{nk} \sum_{i=1}^n \sum_{j=1}^k T_\lambda^{-1} k(g, g_{ij}) \epsilon_{ij} \right| \leqslant \tilde{O}_{\mathbb{P}} \left( \sqrt{\frac{\lambda^{-\alpha}}{n} \sigma_\epsilon^2} \right),$$

where we use Lemma B.3 and Corollary C.2. $\qquad \square$

**Corollary B.5.** *Under Assumption 1, if $\lambda \asymp n^{-\theta}, \theta \in (0, \beta)$, then for $\alpha > \alpha_0$ being sufficiently close, with high probability, for $g \in \mathcal{G}$ almost everywhere,*

$$\left| \frac{1}{nk} \sum_{i=1}^n \sum_{j=1}^k \left( T_{G\lambda}^{-1} - T_\lambda^{-1} \right) k(g, g_{ij}) \epsilon_{ij} \right| \leqslant \tilde{O}_{\mathbb{P}} \left( \sqrt{ \frac{\sigma_k^2}{n} \frac{1}{nk} \sum_{i=1}^n \sum_{j=1}^k \left[ \left( T_{G\lambda}^{-1} - T_\lambda^{-1} \right) k(g_{ij}, g) \right]^2 } \right).$$

*Proof.* Corollary B.5 can be easily proved by Lemma B.4 and Lemma C.10. $\qquad \square$

**Lemma B.6.** (Concentration on $\epsilon^2$) *Under Assumption 1, if $\lambda \asymp n^{-\theta}, \theta \in (0, \beta)$, with high probability,*

$$\frac{1}{nk} \sum_{i=1}^{n} \sum_{j=1}^{k} \epsilon_{ij}^2 \leqslant \sigma^2 + \tilde{O}_{\mathbb{P}}\left(\sigma_\epsilon^2 n^{-\frac{1}{2}}\right).$$

*Proof.* The proof is mainly based on the sub-Gaussianity of $\epsilon|g$. Define $X_{ij} = \epsilon_{ij}^2|g_{ij} - \mathbb{E}[\epsilon_{ij}^2|g_{ij}]$. Hence $\mathbb{E}X_{ij} = 0$. For any $\theta, t > 0$,

$$\mathbb{P}\left(\left|\frac{1}{nk} \sum_{i=1}^{n} \sum_{j=1}^{k} X_{ij}\right| \geqslant t\right) \leqslant 2\exp(-\theta t) \exp \log \mathbb{E} \exp\left(\frac{\theta}{nk} \sum_{i=1}^{n} \sum_{j=1}^{k} X_{ij}\right).$$

The key part is to bound

$$\log \mathbb{E} \exp\left(\frac{\theta}{nk} \sum_{i=1}^{n} \sum_{j=1}^{k} X_{ij}\right) = \log \mathbb{E} \prod_{i=1}^{n} \exp\left(\frac{\theta}{nk} \sum_{j=1}^{k} X_{ij}\right) = \sum_{i=1}^{n} \log \mathbb{E} \exp\left(\frac{\theta}{nk} \sum_{j=1}^{k} X_{ij}\right).$$

Note that $X_{ij}$ is sub-exponential, then for $\frac{\theta}{nk} < \frac{1}{C_K \sigma_\epsilon^2}$,

$$\log \mathbb{E} \exp\left(\frac{\theta}{nk} X_{ij}\right) \leqslant \frac{C_K^2 \sigma_\epsilon^4 \frac{\theta^2}{n^2 k^2}}{1 - C_K \sigma_\epsilon^2 \frac{\theta}{nk}}.$$

By Lemma C.17,

$$\log \mathbb{E} \exp\left(\frac{\theta}{nk} \sum_{j=1}^{k} X_{ij}\right) \leqslant \frac{C_K^2 \sigma_\epsilon^4 \frac{\theta^2}{n^2}}{1 - C_K \sigma_\epsilon^2 \frac{\theta}{n}}.$$

Hence,

$$\log \mathbb{E} \exp\left(\frac{\theta}{nk} \sum_{i=1}^{n} \sum_{j=1}^{k} X_{ij}\right) \leqslant \frac{C_K^2 \sigma_\epsilon^4 \frac{\theta^2}{n}}{1 - C_K \sigma_\epsilon^2 \frac{\theta}{n}}.$$

Set $\theta = \frac{t}{2C_K^2 \sigma_\epsilon^4 \frac{1}{n} + C_K \sigma_\epsilon^2 \frac{1}{n} t}$ then

$$\mathbb{P}\left(\left|\frac{1}{nk} \sum_{i=1}^{n} \sum_{j=1}^{k} X_{ij}\right| \geqslant t\right) \leqslant 2\exp\left(-\frac{1}{2}\frac{t^2}{2\frac{C_K^2 \sigma_\epsilon^4}{n} + C_K \sigma_\epsilon^2 \frac{t}{n}}\right).$$

Let $\delta = 2\exp\left(-\frac{1}{2}\frac{t^2}{2\frac{C_K^2 \sigma_\epsilon^4}{n} + C_K \sigma_\epsilon^2 \frac{t}{n}}\right)$ then we can solve that

$$t \leqslant \tilde{O}_{\mathbb{P}}\left(\sigma_\epsilon^2 n^{-\frac{1}{2}}\right).$$

Therefore, with high probability,

$$\left|\frac{1}{nk} \sum_{i=1}^{n} \sum_{j=1}^{k} X_{ij}\right| \leqslant \tilde{O}_{\mathbb{P}}\left(\sigma_\epsilon^2 n^{-\frac{1}{2}}\right).$$

Note that the conditional expectation

$$\frac{1}{nk} \sum_{i=1}^{n} \sum_{j=1}^{k} \mathbb{E}\left[\epsilon_{ij}^2 | g_{ij}\right] \leqslant \sigma^2,$$

then with high probability,

$$\frac{1}{nk} \sum_{i=1}^{n} \sum_{j=1}^{k} \epsilon_{ij}^2 \leqslant \sigma^2 + \tilde{O}_{\mathbb{P}}\left(\sigma_\epsilon^2 n^{-\frac{1}{2}}\right).$$

$\square$

**Lemma B.7.** *Under Assumption 1, if $\lambda \asymp n^{-\theta}, \theta \in (0, \beta)$, for $\alpha > \alpha_0$ being sufficiently close, with high probability, for $g \in \mathcal{G}$ almost everywhere,*

$$\left| \frac{1}{nk} \sum_{i=1}^{n} \sum_{j=1}^{k} \left[ T_\lambda^{-1} k(g_{ij}, g) \epsilon_{ij} \right]^2 - \frac{1}{nk} \sum_{i=1}^{n} \sum_{j=1}^{k} \left[ T_\lambda^{-1} k(g_{ij}, g) \right]^2 \mathbb{E} \left[ \epsilon_{ij}^2 | g_{ij} \right] \right| \leqslant \tilde{O}_{\mathbb{P}} \left( \lambda^{-\alpha} \sqrt{\frac{\lambda^{-\alpha}}{n}} \sigma_\epsilon^2 \right).$$

*Proof.* The proof can be easily extended from the proof of Lemma B.6. We first focus on the finite net constructed in Lemma B.3. Similarly, we denote $\mathcal{K}_\lambda = \left\{ T_\lambda^{-1} k(g, \cdot) \right\}_{g \in \mathcal{G}}$. By Lemma C.9, we can find an $\epsilon$-net $\mathcal{F} \subset \mathcal{K}_\lambda \subset \mathcal{H}$ such that

$$|\mathcal{F}| \leqslant C'''(\lambda \varepsilon)^{-\frac{2d}{p}},$$

where $\varepsilon = \varepsilon(n) = \frac{1}{n}$. Denote $X_{ij} = \left[ T_\lambda^{-1} k(g_{ij}, g) \right]^2 \left[ \epsilon_{ij}^2 - \mathbb{E}\epsilon_{ij}^2 | g_{ij} \right] | g_{ij}$ with $\mathbb{E}_{\epsilon_{ij} | g_{ij}} X_{ij} = 0$. Then

$$\frac{1}{nk} \sum_{i=1}^{n} \sum_{j=1}^{k} \left[ T_\lambda^{-1} k(g_{ij}, g) \epsilon_{ij} \right]^2 - \frac{1}{nk} \sum_{i=1}^{n} \sum_{j=1}^{k} \left[ T_\lambda^{-1} k(g_{ij}, g) \right]^2 \mathbb{E} \left[ \epsilon_{ij}^2 | g_{ij} \right] = \frac{1}{nk} \sum_{i=1}^{n} \sum_{j=1}^{k} X_{ij}.$$

By the fact that $\epsilon_{ij}^2 | g_{ij}$ is $\sigma_\epsilon^2$ sub-exponential, there exists a constant $C_K$, such that for $\frac{\theta}{nk} < \frac{1}{C_K \sigma_\epsilon^2}$,

$$\log \mathbb{E} \exp \left( \frac{\theta}{nk} \left[ \epsilon_{ij}^2 | g_{ij} - \mathbb{E}\epsilon_{ij}^2 | g_{ij} \right] \right) \leqslant C_K^2 \sigma_\epsilon^4 \frac{\theta^2}{n^2 k^2}.$$

Hence, for $\frac{\theta \left[ T_\lambda^{-1} k(g_{ij}, g) \right]^2}{nk} < \frac{1}{C_K \sigma_\epsilon^2}$,

$$\log \mathbb{E} \exp \left( \frac{\theta \left[ T_\lambda^{-1} k(g_{ij}, g) \right]^2}{nk} \left[ \epsilon_{ij}^2 | g_{ij} - \mathbb{E}\epsilon_{ij}^2 | g_{ij} \right] \right) \leqslant C_K^2 \sigma_\epsilon^4 \frac{\theta^2 \left[ T_\lambda^{-1} k(g_{ij}, g) \right]^4}{n^2 k^2}.$$

In this sense, we get an equivalent $C_K^{(ij)} = C_K \sigma_\epsilon^2 \left[ T_\lambda^{-1} k(g_{ij}, g) \right]^2$. That is, $X_{ij}$ is sub-exponential with sub-exponential norm $C_K^{(ij)}$. Hence,

$$\log \mathbb{E} \exp \left( \frac{\theta}{nk} \sum_{i=1}^{n} \sum_{j=1}^{k} X_{ij} \right) \leqslant \frac{\sum_{i=1}^{n} \left( \sum_{j=1}^{k} C_K^{(ij)} \right)^2 \frac{\theta^2}{n^2 k^2}}{1 - \max_i \sum_{j=1}^{k} C_K^{(ij)} \frac{\theta}{nk}}.$$

Denote $A = \frac{1}{n^2 k^2} \sum_{i=1}^{n} \left( \sum_{j=1}^{k} C_K^{(ij)} \right)^2$, $B = \frac{1}{nk} \max_i \sum_{j=1}^{k} C_K^{(ij)}$ and for any positive $t$, take $\theta = \frac{t}{2A + Bt}$ we have

$$\mathbb{P} \left( \left| \frac{1}{nk} \sum_{i=1}^{n} \sum_{j=1}^{k} X_{ij} \right| \geqslant t \right) \leqslant 2 \exp \left( -\frac{t^2/2}{2A + Bt} \right).$$

Set $\delta = 2 \exp \left( -\frac{t^2/2}{2A + Bt} \right)$ then with probability at least $1 - \delta$,

$$t \leqslant O_{\mathbb{P}} \left( B \log \left( \frac{2}{\delta} \right) + \sqrt{A \log \left( \frac{2}{\delta} \right)} \right).$$

Hence, with probability at least $1 - \delta, \forall f \in \mathcal{F}$,

$$\left| \frac{1}{nk} \sum_{i=1}^{n} \sum_{j=1}^{k} X_{ij} \right| \leqslant O \left( B \log \left( \frac{2|\mathcal{F}|}{\delta} \right) + \sqrt{A \log \left( \frac{2|\mathcal{F}|}{\delta} \right)} \right).$$

We next derive bounds for $A$ and $B$. By Corollary C.2,

$$B \leqslant \sigma_\epsilon^2 O \left( \frac{\lambda^{-2\alpha}}{n} \right).$$

To bound $A$, by definition, we have

$$A = \frac{1}{n^2 k^2} \sum_{i=1}^{n} \left( \sum_{j=1}^{k} C_K^{(ij)} \right)^2$$

$$\leqslant \frac{C_K^2}{n^2 k} \sum_{i=1}^{n} \sum_{j=1}^{k} \left[ T_\lambda^{-1} k(g, g_{ij}) \right]^4 \sigma_\epsilon^4$$

$$\leqslant \left\| T_\lambda^{-1} k(g, \cdot) \right\|_{L^\infty}^2 \frac{C_K^2}{n} \frac{1}{nk} \sum_{i=1}^{n} \sum_{j=1}^{k} \left[ T_\lambda^{-1} k(g, g_{ij}) \right]^2 \sigma_\epsilon^4.$$

By Lemma B.3, with high probability, $\forall f \in \mathcal{F}$,

$$\left| \frac{1}{nk} \sum_{i=1}^{n} \sum_{j=1}^{k} \left[ T_\lambda^{-1} k(g_{ij}, g) \right]^2 - \left\| T_\lambda^{-1} k(g, \cdot) \right\|_{L^2}^2 \right| \leqslant \tilde{O}_{\mathbb{P}} \left( \lambda^{-\alpha} \sqrt{\frac{\lambda^{-\alpha}}{n}} \right).$$

By Lemma C.2,

$$\left\| T_\lambda^{-1} k(g, \cdot) \right\|_{L^\infty}^2 \leqslant M_\alpha^4 \lambda^{-2\alpha}, \quad \left\| T_\lambda^{-1} k(g, \cdot) \right\|_{L^2}^2 \leqslant M_\alpha^2 \lambda^{-\alpha},$$

then

$$A \leqslant \tilde{O}_{\mathbb{P}} \left( \frac{\lambda^{-3\alpha}}{n} \sigma_\epsilon^4 \right).$$

Jointly, with high probability for all $f \in \mathcal{F}$,

$$\left| \frac{1}{nk} \sum_{i=1}^{n} \sum_{j=1}^{k} \left[ T_\lambda^{-1} k(g_{ij}, g) \epsilon_{ij} \right]^2 - \frac{1}{nk} \sum_{i=1}^{n} \sum_{j=1}^{k} \left[ T_\lambda^{-1} k(g_{ij}, g) \right]^2 \mathbb{E} \left[ \epsilon_{ij}^2 | g_{ij} \right] \right|$$

$$= \tilde{O}_{\mathbb{P}} \left( \lambda^{-\alpha} \sqrt{\frac{\lambda^{-\alpha}}{n}} \sigma_\epsilon^2 \right).$$

Further, by the net theory, $\forall g \in \mathcal{G}$, there exists $f \in \mathcal{F}$ such that

$$\left| \left[ T_\lambda^{-1} k(g_{ij}, g) \right]^2 - f^2(g_{ij}) \right| \leqslant \varepsilon O(\lambda^{-\alpha}) = O \left( \frac{\lambda^{-\alpha}}{n} \right).$$

Hence,

$$\left| \frac{1}{nk} \sum_{i=1}^{n} \sum_{j=1}^{k} \left( \left[ T_\lambda^{-1} k(g_{ij}, g) \right]^2 - f(g_{ij})^2 \right) \epsilon_{ij}^2 \right| \leqslant O \left( \frac{\lambda^{-\alpha}}{n} \right) \frac{1}{nk} \sum_{i=1}^{n} \sum_{j=1}^{k} \epsilon_{ij}^2$$

$$\leqslant \sigma^2 O_{\mathbb{P}} \left( \frac{\lambda^{-\alpha}}{n} \right),$$

where the last inequality is the result of Lemma B.6. Jointly, with high probability, for all $g \in \mathcal{G}$,

$$\left| \frac{1}{nk} \sum_{i=1}^{n} \sum_{j=1}^{k} \left[ T_\lambda^{-1} k(g_{ij}, g) \epsilon_{ij} \right]^2 - \frac{1}{nk} \sum_{i=1}^{n} \sum_{j=1}^{k} \left[ T_\lambda^{-1} k(g_{ij}, g) \right]^2 \mathbb{E} \left[ \epsilon_{ij}^2 | g_{ij} \right] \right| \leqslant \tilde{O}_{\mathbb{P}} \left( \lambda^{-\alpha} \sqrt{\frac{\lambda^{-\alpha}}{n}} \sigma_\epsilon^2 \right).$$

$\square$

The following Lemma B.8 can be viewed as an extension from the combination of Lemma B.3 and Lemma B.7.

**Lemma B.8.** *Under Assumption 1, if $\lambda \asymp n^{-\theta}, \theta \in (0, \beta)$, for $\alpha > \alpha_0$ being sufficiently close, with high probability, for $g \in \mathcal{G}$ almost everywhere,*

$$\left| \frac{1}{nk(k-1)} \sum_{i=1}^{n} \sum_{j_1=1}^{k} \sum_{j_2=1, j_2 \neq j_1}^{k} T_\lambda^{-1} k(g_{ij_1}, g) \epsilon_{ij_1} T_\lambda^{-1} k(g_{ij_2}, g) \epsilon_{ij_2} - \mathbb{E} T_\lambda^{-1} k(g_{ij_1}, g) \epsilon_{ij_1} T_\lambda^{-1} k(g_{ij_2}, g) \epsilon_{ij_2} \right|$$

$$\leqslant \sigma_{\epsilon_{1,2}}^2 \tilde{O}_{\mathbb{P}} \left( \lambda^{-\alpha} \sqrt{\frac{\lambda^{-\alpha}}{n}} \right) + \sigma_G^2 \tilde{O}_{\mathbb{P}} \left( \lambda^{-\alpha} \sqrt{\frac{\lambda^{-\alpha}}{n}} \right) = (\sigma_{\epsilon_{1,2}}^2 + \sigma_G^2) \tilde{o}_{\mathbb{P}}(\lambda^{-\alpha}).$$

*Proof.* We first concentrate $\epsilon|g$ and then concentrate $g$. Denote

$$X_{ij_1j_2} = T_\lambda^{-1}k(g_{ij_1}, g)\epsilon_{ij_1}T_\lambda^{-1}k(g_{ij_2}, g)\epsilon_{ij_2}|g_{ij_1}, g_{ij_2} - T_\lambda^{-1}k(g_{ij_1}, g)T_\lambda^{-1}k(g_{ij_2}, g)\mathbb{E}\epsilon_{ij_1}\epsilon_{ij_2}|g_{ij_1}, g_{ij_2}.$$

Note that $\epsilon_{ij_1}|g_{ij_1}, g_{ij_2} \cdot \epsilon_{ij_2}|g_{ij_1}, g_{ij_2}$ is the product of two sub-Gaussian random variables, then $\epsilon_{ij_1}|g_{ij_1}, g_{ij_2} \cdot \epsilon_{ij_2}|g_{ij_1}, g_{ij_2}$ is sub-exponential. Similar to the proof for Lemma B.7, for $\frac{\theta}{nk(k-1)} \leqslant \frac{1}{C_K^{(ij_1j_2)}}$,

$$\log\mathbb{E}\exp\left(\frac{\theta}{nk(k-1)}\sum_{i=1}^n\sum_{j_1=1}^k\sum_{j_2=1,j_2\neq j_1}^k X_{ij_1j_2}\right) \leqslant \frac{\sum_{i=1}^n\left(\sum_{j_1=1}^k\sum_{j_2=1,j_2\neq j_1}^k C_K^{(ij_1j_2)}\right)^2\frac{\theta^2}{n^2k^2(k-1)^2}}{1 - \max_i\sum_{j_1=1}^k\sum_{j_2=1,j_2\neq j_1}^k C_K^{(ij_1j_2)}\frac{\theta}{nk(k-1)}},$$

where

$$C_K^{(ij_1j_2)} = C_K T_\lambda^{-1}k(g_{ij_1}, g)T_\lambda^{-1}k(g_{ij_2}, g)\sigma_{\epsilon_{1,2}}^2.$$

Denote

$$A = \frac{1}{n^2k^2(k-1)^2}\sum_{i=1}^n\left(\sum_{j_1=1}^k\sum_{j_2=1,j_2\neq j_1}^k C_K^{(ij_1j_2)}\right)^2, \quad B = \frac{1}{nk(k-1)}\max_i\sum_{j_1=1}^k\sum_{j_2=1,j_2\neq j_1}^k C_K^{(ij_1j_2)},$$

and take $\theta = \frac{t}{2A+Bt}$ we have

$$\mathbb{P}\left(\left|\frac{1}{nk(k-1)}\sum_{i=1}^n\sum_{j_1=1}^k\sum_{j_2=1,j_2\neq j_1}^k X_{ij_1j_2}\right| \geqslant t\right) \leqslant 2\exp\left(-\frac{t^2/2}{2A+Bt}\right).$$

Set $\delta = 2\exp\left(-\frac{t^2/2}{2A+Bt}\right)$ then we can solve that

$$t \leqslant O_\mathbb{P}\left(B\log\frac{2}{\delta} + \sqrt{A\log\frac{2}{\delta}}\right).$$

Hence, using the net $\mathcal{F}$ constructed in Lemma B.7, with probability at least $1 - \delta, \forall f \in \mathcal{F}$,

$$\left|\frac{1}{nk(k-1)}\sum_{i=1}^n\sum_{j_1=1}^k\sum_{j_2=1,j_2\neq j_1}^k X_{ij_1j_2}\right| \leqslant O\left(B\log\left(\frac{2|\mathcal{F}|}{\delta}\right) + \sqrt{A\log\left(\frac{2|\mathcal{F}|}{\delta}\right)}\right).$$

By Corollary C.2,

$$B \leqslant \sigma_{\epsilon_{1,2}}^2 O\left(\frac{\lambda^{-2\alpha}}{n}\right).$$

To bound $A$, note that

$$A = \frac{C_K^2}{n^2k^2(k-1)^2}\sum_{i=1}^n\left(\sum_{j_1=1}^k\sum_{j_2=1,j_2\neq j_1}^k C_K^{(ij_1j_2)}\right)^2$$

$$\leqslant \frac{C_K^2}{n^2k(k-1)}\sum_{i=1}^n\sum_{j_1=1}^k\sum_{j_2=1,j_2\neq j_1}^k\left[T_\lambda^{-1}k(g, g_{ij_1})\right]^2\left[T_\lambda^{-1}k(g, g_{ij_2})\right]^2\sigma_{\epsilon_{1,2}}^4$$

$$\leqslant \left\|T_\lambda^{-1}k(g, \cdot)\right\|_{L^\infty}^2\frac{C_K^2}{n}\frac{1}{nk}\sum_{i=1}^n\sum_{j=1}^k\left[T_\lambda^{-1}k(g, g_{ij})\right]^2\sigma_{\epsilon_{1,2}}^4.$$

By Lemma B.3, with high probability, $\forall f \in \mathcal{F}$,

$$\left|\frac{1}{nk}\sum_{i=1}^n\sum_{j=1}^k\left[T_\lambda^{-1}k(g_{ij}, g)\right]^2 - \left\|T_\lambda^{-1}k(g, \cdot)\right\|_{L^2}^2\right| \leqslant \tilde{O}_\mathbb{P}\left(\lambda^{-\alpha}\sqrt{\frac{\lambda^{-\alpha}}{n}}\right).$$

By Lemma C.2,

$$\left\|T_\lambda^{-1}k(g, \cdot)\right\|_{L^\infty}^2 \leqslant M_\alpha^4\lambda^{-2\alpha}, \quad \left\|T_\lambda^{-1}k(g, \cdot)\right\|_{L^2}^2 \leqslant M_\alpha^2\lambda^{-\alpha},$$

then
$$A \leqslant \tilde{O}_{\mathbb{P}} \left( \frac{\lambda^{-3\alpha}}{n} \sigma_{\epsilon_{1,2}}^4 \right).$$

Jointly, with high probability for all $f \in \mathcal{F}$,

$$\left| \frac{1}{nk(k-1)} \sum_{i=1}^{n} \sum_{j_1=1}^{k} \sum_{j_2=1,j_2\neq j_1}^{k} T_\lambda^{-1}k(g_{ij_1},g)T_\lambda^{-1}k(g_{ij_2},g)\left(\epsilon_{ij_1}\epsilon_{ij_2} - \mathbb{E}\left[\epsilon_{ij_1}\epsilon_{ij_2}|g_{ij_1},g_{ij_2}\right]\right) \right|$$
$$\leqslant \tilde{O}_{\mathbb{P}} \left( \lambda^{-\alpha}\sqrt{\frac{\lambda^{-\alpha}}{n}}\sigma_{\epsilon_{1,2}}^2 \right).$$

Further, by the net theory, $\forall g \in \mathcal{G}$, there exists $f \in \mathcal{F}$ such that

$$\left| T_\lambda^{-1}k(g_{ij_1},g)T_\lambda^{-1}k(g_{ij_2},g) - f(g_{ij_1})f(g_{ij_2}) \right| \leqslant \varepsilon O(\lambda^{-\alpha}) = O\left( \frac{\lambda^{-\alpha}}{n} \right).$$

Hence,

$$\left| \frac{1}{nk(k-1)} \sum_{i=1}^{n} \sum_{j_1=1}^{k} \sum_{j_2=1,j_2\neq j_1}^{k} \left[ T_\lambda^{-1}k(g_{ij_1},g)T_\lambda^{-1}k(g_{ij_2},g) - f(g_{ij_1})f(g_{ij_2}) \right] \epsilon_{ij_1}\epsilon_{ij_2} \right|$$
$$\leqslant O\left( \frac{\lambda^{-\alpha}}{n} \right) \frac{1}{nk(k-1)} \sum_{i=1}^{n} \sum_{j_1=1}^{k} \sum_{j_2=1,j_2\neq j_1}^{k} \left| \epsilon_{ij_1}\epsilon_{ij_2} \right|$$
$$\leqslant O\left( \frac{\lambda^{-\alpha}}{n} \right) \frac{1}{nk} \sum_{i=1}^{n} \sum_{j=1}^{k} \epsilon_{ij}^2$$
$$\leqslant \sigma^2 O\left( \frac{\lambda^{-\alpha}}{n} \right),$$

where the last inequality uses Lemma B.6. Jointly, with high probability for all $g \in \mathcal{G}$,

$$\left| \frac{1}{nk(k-1)} \sum_{i=1}^{n} \sum_{j_1=1}^{k} \sum_{j_2=1,j_2\neq j_1}^{k} T_\lambda^{-1}k(g_{ij_1},g)T_\lambda^{-1}k(g_{ij_2},g)\left(\epsilon_{ij_1}\epsilon_{ij_2} - \mathbb{E}\left[\epsilon_{ij_1}\epsilon_{ij_2}|g_{ij_1},g_{ij_2}\right]\right) \right|$$
$$\leqslant \tilde{O}_{\mathbb{P}} \left( \lambda^{-\alpha}\sqrt{\frac{\lambda^{-\alpha}}{n}}\sigma_{\epsilon_{1,2}}^2 \right).$$

As the second step, we concentrate $g$. This step is similar to the proof of Lemma B.3. For simplicity, we directly deduce

$$\left| \frac{1}{nk(k-1)} \sum_{i=1}^{n} \sum_{j_1=1}^{k} \sum_{j_2=1,j_2\neq j_1}^{k} T_\lambda^{-1}k(g_{ij_1},g)T_\lambda^{-1}k(g_{ij_2},g)\mathbb{E}\left[\epsilon_{ij_1}\epsilon_{ij_2}|g_{ij_1},g_{ij_2}\right] - \mathbb{E}\left[T_\lambda^{-1}k(g_{ij_1},g)T_\lambda^{-1}k(g_{ij_2},g)\epsilon_{ij_1}\epsilon_{ij_2}\right] \right|$$
$$\leqslant \sigma_G^2 \tilde{O}_{\mathbb{P}} \left( \lambda^{-\alpha}\sqrt{\frac{\lambda^{-\alpha}}{n}} \right).$$

Therefore, for $\alpha > \alpha_0$ being sufficiently close, with high probability, for $g \in \mathcal{G}$ almost everywhere,

$$\left| \frac{1}{nk(k-1)} \sum_{i=1}^{n} \sum_{j_1=1}^{k} \sum_{j_2=1,j_2\neq j_1}^{k} T_\lambda^{-1}k(g_{ij_1},g)\epsilon_{ij_1}T_\lambda^{-1}k(g_{ij_2},g)\epsilon_{ij_2} - \mathbb{E}T_\lambda^{-1}k(g_{ij_1},g)\epsilon_{ij_1}T_\lambda^{-1}k(g_{ij_2},g)\epsilon_{ij_2} \right|$$
$$\leqslant \left| \frac{1}{nk(k-1)} \sum_{i=1}^{n} \sum_{j_1=1}^{k} \sum_{j_2=1,j_2\neq j_1}^{k} T_\lambda^{-1}k(g_{ij_1},g)T_\lambda^{-1}k(g_{ij_2},g)\left(\epsilon_{ij_1}\epsilon_{ij_2} - \mathbb{E}\left[\epsilon_{ij_1}\epsilon_{ij_2}|g_{ij_1},g_{ij_2}\right]\right) \right|$$
$$+ \left| \frac{1}{nk(k-1)} \sum_{i=1}^{n} \sum_{j_1=1}^{k} \sum_{j_2=1}^{k} T_\lambda^{-1}k(g_{ij_1},g)T_\lambda^{-1}k(g_{ij_2},g)\mathbb{E}\left[\epsilon_{ij_1}\epsilon_{ij_2}|g_{ij_1},g_{ij_2}\right] - \mathbb{E}T_\lambda^{-1}k(g_{ij_1},g)\epsilon_{ij_1}T_\lambda^{-1}k(g_{ij_2},g)\epsilon_{ij_2} \right|$$
$$\leqslant \sigma_{\epsilon_{1,2}}^2 \tilde{O}_{\mathbb{P}} \left( \lambda^{-\alpha}\sqrt{\frac{\lambda^{-\alpha}}{n}} \right) + \sigma_G^2 \tilde{O}_{\mathbb{P}} \left( \lambda^{-\alpha}\sqrt{\frac{\lambda^{-\alpha}}{n}} \right).$$

$\square$

**Lemma B.9.** *Under Assumption 1, if $\lambda \asymp n^{-\theta}, \theta \in (0, \beta)$, with high probability, for $g \in \mathcal{G}$ almost everywhere,*

$$\left| \frac{1}{n^2 k^2} \sum_{i_1=1}^{n} \sum_{i_2=1, i_2 \neq i_1}^{n} \sum_{j_1=1}^{k} \sum_{j_2=1}^{k} T_\lambda^{-1} k(g_{i_1 j_1}, g) T_\lambda^{-1} k(g_{i_2 j_2}, g) \epsilon_{i_1 j_1} \epsilon_{i_2 j_2} \right|$$

$$\leqslant \tilde{\sigma}^2 \tilde{O}_\mathbb{P} \left( \frac{\left\| T_\lambda^{-1} k(g, \cdot) \right\|_{L^2}^2}{n} \left( r_T + \frac{1 - r_T}{k} \right) \right).$$

*Proof.* We prove by the truncation technique. Denote $X_i = \sum_{j=1}^{k} T_\lambda^{-1} k(g, g_{ij}) \epsilon_{ij}$. We truncate by $\tau$ which will be determined later.

$$X_i = X_i \mathbb{I}_{\{|\epsilon_{ij}| \leqslant \tau, j=1,\dots,k\}} + X_i \mathbb{I}_{\{\exists j \in [k], |\epsilon_{ij}| > \tau\}}$$

$$= \underbrace{X_i \mathbb{I}_{\{|\epsilon_{ij}| \leqslant \tau, j=1,\dots,k\}} - \mathbb{E}\left[ X_i \mathbb{I}_{\{|\epsilon_{ij}| \leqslant \tau, j=1,\dots,k\}} \right]}_{X_i^{(1)}} + \underbrace{X_i \mathbb{I}_{\{\exists j \in [k], |\epsilon_{ij}| > \tau\}} - \mathbb{E}\left[ X_i \mathbb{I}_{\{\exists j \in [k], |\epsilon_{ij}| > \tau\}} \right]}_{X_i^{(2)}}$$

$$:= X_i^{(1)} + X_i^{(2)}.$$

Therefore,

$$\left| \frac{1}{n^2 k^2} \sum_{i_1=1}^{n} \sum_{i_2=1, i_2 \neq i_1}^{n} \sum_{j_1=1}^{k} \sum_{j_2=1}^{k} T_\lambda^{-1} k(g_{i_1 j_1}, g) T_\lambda^{-1} k(g_{i_2 j_2}, g) \epsilon_{i_1 j_1} \epsilon_{i_2 j_2} \right|$$

$$= \left| \frac{1}{n^2 k^2} \sum_{i_1=1}^{n} \sum_{i_2=1, i_2 \neq i_1}^{n} X_{i_1} X_{i_2} \right|$$

$$= \left| \frac{1}{n^2 k^2} \sum_{i_1=1}^{n} \sum_{i_2=1, i_2 \neq i_1}^{n} \left[ X_{i_1}^{(1)} X_{i_2}^{(1)} + X_{i_1}^{(1)} X_{i_2}^{(2)} + X_{i_1}^{(2)} X_{i_2}^{(1)} + X_{i_1}^{(2)} X_{i_2}^{(2)} \right] \right|.$$

We first bound the main term $\left| \frac{1}{n^2 k^2} \sum_{i_1=1}^{n} \sum_{i_2=1, i_2 \neq i_1}^{n} X_{i_1}^{(1)} X_{i_2}^{(1)} \right|$. Note that by Corollary C.2,

$$\left| X_i^{(1)} \right| \leqslant O\left( \lambda^{-\alpha} k \tau \right), \quad \mathbb{E} X_i^{(1)} = 0,$$

by Proposition D.3, we have

$$\mathbb{P}\left\{ \left| \sum_{i=1}^{n} X_i^{(1)} \right| \geqslant t \right\} \leqslant 2 \exp\left( -\frac{t^2/2}{O\left( \lambda^{-\alpha} k \tau \log n \right) t + \sum_{i=1}^{n} \mathbb{E} X_i^{(1)^2}} \right), \tag{B.4}$$

which implies that

$$\mathbb{P}\left\{ \left[ \sum_{i=1}^{n} X_i^{(1)} \right]^2 \geqslant t^2 \right\} \leqslant 2 \exp\left( -\frac{t^2/2}{O\left( \lambda^{-\alpha} k \tau \log n \right) t + \sum_{i=1}^{n} \mathbb{E} X_i^{(1)^2}} \right).$$

Set $\delta = 2 \exp\left( -\frac{t^2/2}{O(\lambda^{-\alpha} k \tau \log n) t + \sum_{i=1}^{n} \mathbb{E} X_i^{(1)^2}} \right)$ then we can solve that

$$t \leqslant O\left( \lambda^{-\alpha} k \tau \log n \log \frac{2}{\delta} \right) + O\left( \sqrt{\sum_{i=1}^{n} \mathbb{E} X_i^{(1)^2} \log \frac{2}{\delta}} \right),$$

and

$$t^2 \leqslant O\left( \lambda^{-2\alpha} k^2 \tau^2 \log^2 n \log^2 \frac{2}{\delta} \right) + O\left( \sum_{i=1}^{n} \mathbb{E} X_i^{(1)^2} \log \frac{2}{\delta} \right).$$

Therefore, considering the net $\mathcal{F}$ [8] constructed in Lemma B.3, it holds with probability at least $1 - \delta$, for any $f \in \mathcal{F}$,

$$\left| \frac{1}{n^2 k^2} \left[ \sum_{i=1}^{n} X_i^{(1)} \right]^2 \right|$$

$$\leqslant \frac{1}{n^2 k^2} \left[ O\left( \lambda^{-2\alpha} k^2 \tau^2 \log^2 n \log^2 \frac{2|\mathcal{F}|}{\delta} \right) + O\left( \sum_{i=1}^{n} \mathbb{E} X_i^{(1)^2} \log \frac{2|\mathcal{F}|}{\delta} \right) \right]$$

$$\leqslant \frac{1}{n^2 k^2} \left[ O\left( \lambda^{-2\alpha} k^2 \tau^2 \log^2 n \log^2 \frac{2nk}{\delta} \right) + O\left( n\tilde{\sigma}^2 \left\| T_\lambda^{-1} k(g, \cdot) \right\|_{L^2}^2 \left( k^2 r_T + k(1 - r_T) \right) \log \frac{2nk}{\delta} \right) \right],$$

where the last inequality uses the definition of net $\mathcal{F}$ and Definition 3.2. By setting $\tau = n^\ell \tilde{\sigma}$ with $\ell < \frac{1 - \alpha\theta}{2}$, we have with high probability, for any $f \in \mathcal{F}$,

$$\left| \frac{1}{n^2 k^2} \left[ \sum_{i=1}^{n} X_i^{(1)} \right]^2 \right| \leqslant \tilde{O}_{\mathbb{P}} \left( \tilde{\sigma}^2 \frac{\left\| T_\lambda^{-1} k(g, \cdot) \right\|_{L^2}^2}{n} \left( r_T + \frac{1 - r_T}{k} \right) \right).$$

We then consider $\left| \frac{1}{n^2 k^2} \sum_{i=1}^{n} \left( X_i^{(1)} \right)^2 \right|$. Applying Proposition D.3,

$$\mathbb{P} \left\{ \left| \sum_{i=1}^{n} \left( X_i^{(1)} \right)^2 - n\mathbb{E} \left[ \left( X_i^{(1)} \right)^2 \right] \right| \geqslant t \right\} \leqslant 2 \exp \left( -\frac{t^2/2}{O\left( \lambda^{-2\alpha} k^2 \tau^2 \log n \right) t + O\left( \lambda^{-2\alpha} k^2 \tau^2 \right) \sum_{i=1}^{n} \mathbb{E} \left[ \left( X_i^{(1)} \right)^2 \right]} \right),$$

Hence, with high probability, for any $f \in \mathcal{F}$,

$$\left| \frac{1}{n^2 k^2} \sum_{i=1}^{n} \left( X_i^{(1)} \right)^2 \right| \leqslant \frac{\mathbb{E} \left[ \left( X_i^{(1)} \right)^2 \right]}{nk^2} + \tilde{O}_{\mathbb{P}} \left( \frac{\lambda^{-\alpha} k\tau \sqrt{n\mathbb{E} \left[ \left( X_i^{(1)} \right)^2 \right]} + \lambda^{-2\alpha} k^2 \tau^2 \log n}{n^2 k^2} \right)$$

$$= O\left( \tilde{\sigma}^2 \frac{\left\| T_\lambda^{-1} k(g, \cdot) \right\|_{L^2}^2}{n} \left( r_T + \frac{1 - r_T}{k} \right) \right) + \tilde{O}_{\mathbb{P}} \left( \frac{\lambda^{-\alpha} \tau \tilde{\sigma} \sqrt{n\lambda^{-\alpha}} + \lambda^{-2\alpha} \tau^2}{n^2} \right)$$

$$= \tilde{O}_{\mathbb{P}} \left( \tilde{\sigma}^2 \frac{\left\| T_\lambda^{-1} k(g, \cdot) \right\|_{L^2}^2}{n} \left( r_T + \frac{1 - r_T}{k} \right) \right).$$

Combining these two bounds, with high probability, for any $f \in \mathcal{F}$, the main term

$$\left| \frac{1}{n^2 k^2} \sum_{i_1=1}^{n} \sum_{i_2=1, i_2 \neq i_1}^{n} X_{i_1}^{(1)} X_{i_2}^{(1)} \right| \leqslant \tilde{O}_{\mathbb{P}} \left( \tilde{\sigma}^2 \frac{\left\| T_\lambda^{-1} k(g, \cdot) \right\|_{L^2}^2}{n} \left( r_T + \frac{1 - r_T}{k} \right) \right).$$

We second bound the residual terms

$$\left| \frac{1}{n^2 k^2} \sum_{i_1=1}^{n} \sum_{i_2=1, i_2 \neq i_1}^{n} \left[ X_{i_1}^{(1)} X_{i_2}^{(2)} + X_{i_1}^{(2)} X_{i_2}^{(1)} + X_{i_1}^{(2)} X_{i_2}^{(2)} \right] \right|.$$

By Markov inequality,

$$\mathbb{P} \left\{ \exists j \in [k], |\epsilon_{ij}| > \tau \right\} \leqslant \sum_{j=1}^{k} \mathbb{P} \left\{ |\epsilon_{ij}| > \tau \right\} \leqslant \sum_{j=1}^{k} \frac{\|\epsilon_{ij}\|_{L^q}^q}{\tau^q},$$

---

[8] Here the net $\mathcal{F}$ is the $\epsilon$-net with $\epsilon = \frac{1}{nk}$.

then
$$\mathbb{P}\left\{\sum_{j=1}^{k}|\epsilon_{ij}| \leqslant k\tau, \ \forall i \in [1,n], \ \forall f \in \mathcal{F}\right\} \geqslant \left(1 - \sum_{j=1}^{k} \frac{\|\epsilon_{ij}\|_{L^q}^q}{\tau^q}\right)^n.$$

That said, if we set $\ell > \frac{1}{q}$ then $X_i\mathbb{I}_{\{\exists j\in[k],|\epsilon_{ij}|>\tau\}}$ vanishes for the reason that $\epsilon$ is (conditional) sub-Gaussian. Hence, with high probability, for any $f \in \mathcal{F}$

$$\left|\frac{1}{n^2k^2}\sum_{i_1=1}^{n}\sum_{i_2=1,i_2\neq i_1}^{n}\left[X_{i_1}^{(1)}X_{i_2}^{(2)} + X_{i_1}^{(2)}X_{i_2}^{(1)} + X_{i_1}^{(2)}X_{i_2}^{(2)}\right]\right|$$

$$= \left|\frac{1}{n^2k^2}\sum_{i_1=1}^{n}\sum_{i_2=1,i_2\neq i_1}^{n}\left[X_{i_1}^{(1)}\mathbb{E}\left[X_i\mathbb{I}_{\{\exists j\in[k],\epsilon_{ij}>\tau\}}\right] + \mathbb{E}\left[X_i\mathbb{I}_{\{\exists j\in[k],\epsilon_{ij}>\tau\}}\right]X_{i_2}^{(1)} + \mathbb{E}^2\left[X_i\mathbb{I}_{\{\exists j\in[k],\epsilon_{ij}>\tau\}}\right]\right]\right|$$

$$\leqslant 2\left|\mathbb{E}\left[X_i\mathbb{I}_{\{\exists j\in[k],|\epsilon_{ij}|>\tau\}}\right]\right| \cdot \left|\frac{1}{nk^2}\sum_{i=1}^{n}X_i^{(1)}\right| + \frac{\mathbb{E}^2\left[X_i\mathbb{I}_{\{\exists j\in[k],|\epsilon_{ij}|>\tau\}}\right]}{k^2}.$$

Firstly, by Cauchy-Schwarz inequality

$$\frac{1}{k^2}\mathbb{E}^2\left[X_i\mathbb{I}_{\{\sum_{j=1}^{k}|\epsilon_{ij}|>k\tau\}}\right] \leqslant \frac{1}{k^2}\mathbb{E}X_i^2\mathbb{P}\left\{\sum_{j=1}^{k}|\epsilon_{ij}| > k\tau\right\}$$

$$\leqslant \left(r_T + \frac{(1-r_T)}{k}\right)\tilde{\sigma}^2 O(\lambda^{-\alpha})\frac{\left\|\sum_{i=1}^{k}\epsilon_{ij}\right\|_{L^q}^q}{k^q\tau^q} \qquad \text{(B.5)}$$

$$= \left(r_T + \frac{(1-r_T)}{k}\right)\tilde{\sigma}^2 O\left(\frac{\lambda^{-\alpha}}{\tau^q}\right).$$

Secondly, by Equation (B.4),

$$\mathbb{P}\left\{\left|\sum_{i=1}^{n}X_i^{(1)}\right| \geqslant t\right\} \leqslant 2\exp\left(-\frac{t^2/2}{O\left(\lambda^{-\alpha}k\tau\log nk\right)t + \sum_{i=1}^{n}\mathbb{E}X_i^{(1)^2}}\right),$$

we have with high probability, $\forall f \in \mathcal{F}$,

$$\frac{1}{nk}\left|\sum_{i=1}^{n}X_i^{(1)}\right| \leqslant \tilde{O}_{\mathbb{P}}\left(\sqrt{\frac{\left(r_T + \frac{(1-r_T)}{k}\right)}{n}\tilde{\sigma}^2\left\|T_\lambda^{-1}k(g,\cdot)\right\|_{L^2}^2} + \frac{\lambda^{-\alpha}}{n}\right).$$

Further, by Equation (B.5),

$$\frac{\left|\mathbb{E}\left[X_i\mathbb{I}_{\{\sum_{j=1}^{k}|\epsilon_{ij}|>k\tau\}}\right]\right|}{k} \leqslant \sqrt{\left(r_T + \frac{(1-r_T)}{k}\right)}\tilde{\sigma}O\left(\frac{\lambda^{-\frac{\alpha}{2}}}{\tau^{\frac{q}{2}}}\right).$$

Therefore

$$2\left|\mathbb{E}\left[X_i\mathbb{I}_{\{\sum_{j=1}^{k}|\epsilon_{ij}|>k\tau\}}\right]\right|\frac{1}{nk^2}\sum_{i=1}^{n}\left|X_i^{(1)}\right| \leqslant \left(r_T + \frac{(1-r_T)}{k}\right)\tilde{\sigma}^2\tilde{O}_{\mathbb{P}}\left(\frac{\lambda^{-\alpha}}{n^{1/2}\tau^{\frac{q}{2}}}\right).$$

Note that under the condition $\ell > \frac{1}{q}$, it holds

$$\frac{\lambda^{-\alpha}}{\tau^q} < \frac{\lambda^{-\alpha}}{n}, \ \text{and} \ \frac{\lambda^{-\alpha}}{n^{1/2}\tau^{\frac{q}{2}}} < \frac{\lambda^{-\alpha}}{n}.$$

Then with high probability for all $f \in \mathcal{F}$, the residual terms

$$\left|\frac{1}{n^2k^2}\sum_{i_1=1}^{n}\sum_{i_2=1,i_2\neq i_1}^{n}\left[X_{i_1}^{(1)}X_{i_2}^{(2)} + X_{i_1}^{(2)}X_{i_2}^{(1)} + X_{i_1}^{(2)}X_{i_2}^{(2)}\right]\right| = \tilde{o}_{\mathbb{P}}\left(\frac{\left\|T_\lambda^{-1}k(g,\cdot)\right\|_{L^2}^2}{n}\right)\left(r_T + \frac{1-r_T}{k}\right)\tilde{\sigma}^2.$$

Jointly, if we select $\frac{1}{q} < \ell < \frac{1-\alpha\theta}{2}$ [9], then with high probability, for all $f \in \mathcal{F}$,

$$\left| \frac{1}{n^2 k^2} \sum_{i_1=1}^n \sum_{i_2=1, i_2 \neq i_1}^n \sum_{j_1=1}^k \sum_{j_2=1}^k T_\lambda^{-1} k(g_{i_1 j_1}, g) T_\lambda^{-1} k(g_{i_2 j_2}, g) \epsilon_{i_1 j_1} \epsilon_{i_2 j_2} \right|$$

$$\leqslant \tilde{\sigma}^2 \tilde{O}_{\mathbb{P}} \left( \frac{\left\| T_\lambda^{-1} k(g, \cdot) \right\|_{L^2}^2}{n} \left( r_T + \frac{1-r_T}{k} \right) \right).$$

At last, for any $g \in \mathcal{G}$, there exists $f \in \mathcal{F}$, such that

$$\left| T_\lambda^{-1} k(g_{i_1 j_1}, g) T_\lambda^{-1} k(g_{i_2 j_2}, g) - f(g_{i_1 j_1}) f(g_{i_2 j_2}) \right| \leqslant \varepsilon O(\left\| T_\lambda^{-1} k(g, \cdot) \right\|_{L^2}^2) = O \left( \frac{\left\| T_\lambda^{-1} k(g, \cdot) \right\|_{L^2}^2}{nk} \right).$$

Hence,

$$\left| \frac{1}{n^2 k^2} \sum_{i_1=1}^n \sum_{i_2=1, i_2 \neq i_1}^n \sum_{j_1=1}^k \sum_{j_2=1}^k \left[ T_\lambda^{-1} k(g_{i_1 j_1}, g) T_\lambda^{-1} k(g_{i_2 j_2}, g) - f(g_{i_1 j_1}) f(g_{i_2 j_2}) \right] \epsilon_{i_1 j_1} \epsilon_{i_2 j_2} \right|$$

$$\leqslant O \left( \frac{\left\| T_\lambda^{-1} k(g, \cdot) \right\|_{L^2}^2}{nk} \right) \frac{1}{n^2 k^2} \sum_{i_1=1}^n \sum_{i_2=1, i_2 \neq i_1}^n \sum_{j_1=1}^k \sum_{j_2=1}^k \left| \epsilon_{i_1 j_1} \epsilon_{i_2 j_2} \right|$$

$$\leqslant O \left( \frac{\left\| T_\lambda^{-1} k(g, \cdot) \right\|_{L^2}^2}{nk} \right) \tilde{O}_{\mathbb{P}} \left( \sigma^2 \right)$$

$$= \sigma^2 \tilde{O}_{\mathbb{P}} \left( \frac{\left\| T_\lambda^{-1} k(g, \cdot) \right\|_{L^2}^2}{nk} \right)$$

$$\leqslant \tilde{\sigma}^2 \tilde{O}_{\mathbb{P}} \left( \frac{\left\| T_\lambda^{-1} k(g, \cdot) \right\|_{L^2}^2}{nk} \right),$$

where the second inequality utilizes Lemma B.6.

Jointly, with high probability, for $g \in \mathcal{G}$ almost everywhere,

$$\left| \frac{1}{n^2 k^2} \sum_{i_1=1}^n \sum_{i_2=1, i_2 \neq i_1}^n \sum_{j_1=1}^k \sum_{j_2=1}^k T_\lambda^{-1} k(g_{i_1 j_1}, g) T_\lambda^{-1} k(g_{i_2 j_2}, g) \epsilon_{i_1 j_1} \epsilon_{i_2 j_2} \right| \leqslant \tilde{\sigma}^2 \tilde{O}_{\mathbb{P}} \left( \frac{\left\| T_\lambda^{-1} k(g, \cdot) \right\|_{L^2}^2}{n} \left( r_T + \frac{1-r_T}{k} \right) \right).$$

$\square$

**Lemma B.10.** *Under Assumption 1, if $\lambda \asymp n^{-\theta}, \theta \in (0, \beta)$, for $\alpha > \alpha_0$ being sufficiently close, with high probability, for $g \in \mathcal{G}$ almost everywhere,*

$$\left| \left\| T_G^{\frac{1}{2}} T_{G\lambda}^{-1} k(g, \cdot) \right\|_{\mathcal{H}} - \left\| T_G^{\frac{1}{2}} T_\lambda^{-1} k(g, \cdot) \right\|_{\mathcal{H}} \right| \leqslant \tilde{O}_{\mathbb{P}} \left( \lambda^{-\frac{\alpha}{2}} \sqrt{\frac{\lambda^{-\alpha}}{n}} \right).$$

*Proof.*

$$\left| \left\| T_G^{\frac{1}{2}} T_{G\lambda}^{-1} k(g, \cdot) \right\|_{\mathcal{H}} - \left\| T_G^{\frac{1}{2}} T_\lambda^{-1} k(g, \cdot) \right\|_{\mathcal{H}} \right| \leqslant \left\| T_G^{\frac{1}{2}} (T_{G\lambda}^{-1} - T_\lambda^{-1}) k(g, \cdot) \right\|_{\mathcal{H}}$$

$$= \left\| T_G^{\frac{1}{2}} T_{G\lambda}^{-1} (T - T_G) T_\lambda^{-1} k(g, \cdot) \right\|_{\mathcal{H}}$$

$$\leqslant \left\| T_G^{\frac{1}{2}} T_{G\lambda}^{-\frac{1}{2}} \right\| \left\| T_{G\lambda}^{-\frac{1}{2}} T_\lambda^{\frac{1}{2}} \right\| \left\| T_\lambda^{-\frac{1}{2}} (T - T_G) T_\lambda^{-\frac{1}{2}} \right\| \left\| T_\lambda^{-\frac{1}{2}} k(g, \cdot) \right\|_{\mathcal{H}}.$$

For the first term,

$$\left\| T_G^{\frac{1}{2}} T_{G\lambda}^{-\frac{1}{2}} \right\| \leqslant 1.$$

---

[9] $\alpha\theta < 1$ satisfies this condition.

For the second and the third term, by Lemma B.1,

$$\|T_{G\lambda}^{-\frac{1}{2}} T_\lambda^{\frac{1}{2}}\| \leqslant O_{\mathbb{P}}(1),$$

$$\|T_\lambda^{-\frac{1}{2}}(T - T_G)T_\lambda^{-\frac{1}{2}}\| \leqslant \tilde{O}_{\mathbb{P}}\left(\sqrt{\frac{\lambda^{-\alpha}}{n}}\right).$$

For the last term, by Corollary C.2,

$$\|T_\lambda^{-\frac{1}{2}} k(g, \cdot)\|_{\mathcal{H}} \leqslant M_\alpha \lambda^{-\frac{\alpha}{2}}.$$

Therefore

$$\left|\|T_G^{\frac{1}{2}} T_{G\lambda}^{-1} k(g, \cdot)\|_{\mathcal{H}} - \|T_G^{\frac{1}{2}} T_\lambda^{-1} k(g, \cdot)\|_{\mathcal{H}}\right| \leqslant \tilde{O}_{\mathbb{P}}\left(\lambda^{-\frac{\alpha}{2}} \sqrt{\frac{\lambda^{-\alpha}}{n}}\right).$$

$\square$

**Lemma B.11.** *If the conditional orthogonality holds, then*

$$\mathbb{E} T_\lambda^{-1} k(g_{ij_1}, g)\epsilon_{ij_1} T_\lambda^{-1} k(g_{ij_2}, g)\epsilon_{ij_2} \leqslant \tilde{\sigma}^2(r_e \vee r_0) O\left(\|T_\lambda^{-1} k(g, \cdot)\|_{L^2}^2\right).$$

*Proof.* By the optimality (3.1), we decompose:

$$\mathbb{E}\left[T_\lambda^{-1} k(g_{ij_1}, g)\epsilon_{ij_1} T_\lambda^{-1} k(g_{ij_2}, g)\epsilon_{ij_2}\right]$$
$$= \mathbb{E}\left[\left(T_\lambda^{-1} k(g_{ij_1}, g)\epsilon_{ij_1} - T_\lambda^{-1} k(g_{i'j_1}, g)\epsilon_{i'j_1}\right)\left(T_\lambda^{-1} k(g_{ij_2}, g)\epsilon_{ij_2} - T_\lambda^{-1} k(g_{i''j_2}, g)\epsilon_{i''j_2}\right)\right].$$

where $g_{i'j_1}$ and $g_{ij_1}$ share the same $u_{ij_1}$ but with independent $x_i$ and $x_i'$. $g_{i''j_2}$ and $g_{ij_2}$ share the same $u_{ij_2}$ but with independent $x_i$ and $x_i''$, $\epsilon_{i'j_1} := y_{ij_1} - f_\rho^*(g_{i'j_1}), \epsilon_{i''j_2} := y_{ij_2} - f_\rho^*(g_{i''j_2})$. Further,

$$\left(T_\lambda^{-1} k(g_{ij_1}, g)\epsilon_{ij_1} - T_\lambda^{-1} k(g_{i'j_1}, g)\epsilon_{i'j_1}\right)\left(T_\lambda^{-1} k(g_{ij_2}, g)\epsilon_{ij_2} - T_\lambda^{-1} k(g_{i''j_2}, g)\epsilon_{i''j_2}\right)$$
$$= \left(T_\lambda^{-1} k(g_{i'j_1}, g)(\epsilon_{ij_1} - \epsilon_{i'j_1}) + \epsilon_{ij_1}\left(T_\lambda^{-1} k(g_{ij_1}, g) - T_\lambda^{-1} k(g_{i'j_1}, g)\right)\right)$$
$$\left(T_\lambda^{-1} k(g_{i''j_2}, g)(\epsilon_{ij_2} - \epsilon_{i''j_2}) + \epsilon_{ij_2}\left(T_\lambda^{-1} k(g_{ij_2}, g) - T_\lambda^{-1} k(g_{i''j_2}, g)\right)\right)$$
$$= \underbrace{T_\lambda^{-1} k(g_{i'j_1}, g)(\epsilon_{ij_1} - \epsilon_{i'j_1}) T_\lambda^{-1} k(g_{i''j_2}, g)(\epsilon_{ij_2} - \epsilon_{i''j_2})}_{\Delta_1}$$
$$+ \underbrace{T_\lambda^{-1} k(g_{i'j_1}, g)(\epsilon_{ij_1} - \epsilon_{i'j_1})\epsilon_{ij_2}\left(T_\lambda^{-1} k(g_{ij_2}, g) - T_\lambda^{-1} k(g_{i''j_2}, g)\right)}_{\Delta_2}$$
$$+ \underbrace{\epsilon_{ij_1}\left(T_\lambda^{-1} k(g_{ij_1}, g) - T_\lambda^{-1} k(g_{i'j_1}, g)\right) T_\lambda^{-1} k(g_{i''j_2}, g)(\epsilon_{ij_2} - \epsilon_{i''j_2})}_{\Delta_3}$$
$$+ \underbrace{\epsilon_{ij_1}\left(T_\lambda^{-1} k(g_{ij_1}, g) - T_\lambda^{-1} k(g_{i'j_1}, g)\right) \epsilon_{ij_2}\left(T_\lambda^{-1} k(g_{ij_2}, g) - T_\lambda^{-1} k(g_{i''j_2}, g)\right)}_{\Delta_4}.$$

Then we bound each term under expectation respectively. For $\Delta_1$,

$$\mathbb{E}[\Delta_1] = \mathbb{E}\left[T_\lambda^{-1} k(g_{i'j_1}, g) T_\lambda^{-1} k(g_{i''j_2}, g)(\epsilon_{ij_1} - \epsilon_{i'j_1})(\epsilon_{ij_2} - \epsilon_{i''j_2})\right]$$
$$\leqslant \mathbb{E}\left[\left(T_\lambda^{-1} k(g_{i'j_1}, g)\right)^2 \left(T_\lambda^{-1} k(g_{i''j_2}, g)\right)^2\right]^{\frac{1}{2}} \mathbb{E}\left[(\epsilon_{ij_1} - \epsilon_{i'j_1})^2(\epsilon_{ij_2} - \epsilon_{i''j_2})^2\right]^{\frac{1}{2}}$$
$$= \|T_\lambda^{-1} k(g, \cdot)\|_{L^2}^2 \mathbb{E}\left[(\epsilon_{ij_1} - \epsilon_{i'j_1})^2(\epsilon_{ij_2} - \epsilon_{i''j_2})^2\right]^{\frac{1}{2}}.$$

Note that the kernel Hölder-continuous assumption implies that $f_\rho^* \in \mathcal{H}$ is Hölder-continuous with index $\frac{p}{2}$ [19, 18]. Hence, there exists $L_\epsilon > 0$, such that

$$|\epsilon_{ij_1} - \epsilon_{i'j_1}| \leqslant L_\epsilon \|g_{ij_1} - g_{i'j_1}\|^{\frac{p}{2}},$$

then

$$|\epsilon_{ij_1} - \epsilon_{i'j_1}|^2 \leqslant L_\epsilon^2 \|g_{ij_1} - g_{i'j_1}\|^p.$$

Therefore,

$$
\mathbb{E}[\Delta_1] \leqslant \mathbb{E}\left[\|g_{ij_1} - g_{i'j_1}\|^{2p}\right]^{\frac{1}{2}} \tilde{\sigma}^2 O\left(\|T_\lambda^{-1}k(g,\cdot)\|_{L^2}^2\right)
$$

$$
\leqslant \left(\mathbb{E}\left[\|g_{ij_1} - g_{i'j_1}\|^2\right]\right)^{\frac{p}{2}} \tilde{\sigma}^2 O\left(\|T_\lambda^{-1}k(g,\cdot)\|_{L^2}^2\right)
$$

$$
\leqslant r_0 \tilde{\sigma}^2 O\left(\|T_\lambda^{-1}k(g,\cdot)\|_{L^2}^2\right).
$$

For $\Delta_2$,

$$
\mathbb{E}[\Delta_2] = \mathbb{E}\left[T_\lambda^{-1}k(g_{i'j_1},g)(\epsilon_{ij_1} - \epsilon_{i'j_1})\epsilon_{ij_2}\left(T_\lambda^{-1}k(g_{ij_2},g) - T_\lambda^{-1}k(g_{i''j_2},g)\right)\right]
$$

$$
\leqslant \mathbb{E}\left[\left(T_\lambda^{-1}k(g_{i'j_1},g)\right)^2\left(T_\lambda^{-1}k(g_{ij_2},g) - T_\lambda^{-1}k(g_{i''j_2},g)\right)^2 \epsilon_{ij_2}^2\right]^{\frac{1}{2}} \mathbb{E}\left[(\epsilon_{ij_1} - \epsilon_{i'j_1})^2\right]^{\frac{1}{2}}
$$

$$
= \sqrt{\mathbb{E}\left[\left(T_\lambda^{-1}k(g_{i'j_1},g)\right)^2\right]\mathbb{E}\left[\left(T_\lambda^{-1}k(g_{ij_2},g) - T_\lambda^{-1}k(g_{i''j_2},g)\right)^2 \epsilon_{ij_2}^2\right]} \mathbb{E}\left[(\epsilon_{ij_1} - \epsilon_{i'j_1})^2\right]^{\frac{1}{2}}
$$

$$
= \|T_\lambda^{-1}k(g,\cdot)\|_{L^2}\sqrt{\mathbb{E}\left[\left(T_\lambda^{-1}k(g_{ij_2},g) - T_\lambda^{-1}k(g_{i''j_2},g)\right)^2 \epsilon_{ij_2}^2\right]} \mathbb{E}\left[(\epsilon_{ij_1} - \epsilon_{i'j_1})^2\right]^{\frac{1}{2}}
$$

$$
\leqslant \sigma_G \|T_\lambda^{-1}k(g,\cdot)\|_{L^2}\sqrt{\mathbb{E}\left[\left(T_\lambda^{-1}k(g_{ij_2},g) - T_\lambda^{-1}k(g_{i''j_2},g)\right)^2\right]} \mathbb{E}\left[(\epsilon_{ij_1} - \epsilon_{i'j_1})^2\right]^{\frac{1}{2}}.
$$

Similarly,

$$
\mathbb{E}\left[(\epsilon_{ij_1} - \epsilon_{i'j_1})^2\right]^{\frac{1}{2}} \leqslant \sqrt{r_0}\tilde{\sigma}O(1).
$$

We then focus on

$$
\mathbb{E}\left[\left(T_\lambda^{-1}k(g_{ij_2},g) - T_\lambda^{-1}k(g_{i''j_2},g)\right)^2\right]
$$

$$
= \mathbb{E}\left[\sum_{r,s}\frac{\lambda_s\lambda_r}{(\lambda+\lambda_r)(\lambda+\lambda_s)}e_r(g)e_s(g)\left(e_r(g_{ij_2}) - e_r(g_{i''j_2})\right)\left(e_s(g_{ij_2}) - e_s(g_{i''j_2})\right)\right]
$$

$$
= \sum_r \frac{\lambda_r^2}{(\lambda+\lambda_r)^2}e_r(g)^2\mathbb{E}\left[\left(e_r(g_{ij_2}) - e_r(g_{i''j_2})\right)^2\right]
$$

$$
- \sum_{r\neq s}\frac{\lambda_s\lambda_r}{(\lambda+\lambda_r)(\lambda+\lambda_s)}e_r(g)e_s(g)\mathbb{E}\left[e_r(g_{ij_2})e_s(g_{i''j_2}) + e_r(g_{i''j_2})e_s(g_{ij_2})\right].
$$

If the conditional orthogonality holds, then

$$
\mathbb{E}\left[\left(T_\lambda^{-1}k(g_{ij_2},g) - T_\lambda^{-1}k(g_{i''j_2},g)\right)^2\right] = \sum_r \frac{\lambda_r^2}{(\lambda+\lambda_r)^2}e_r(g)^2\mathbb{E}\left[\left(e_r(g_{ij_2}) - e_r(g_{i''j_2})\right)^2\right].
$$

By the definition of $r_e$,

$$
\mathbb{E}\left[\left(e_r(g_{ij_2}) - e_r(g_{i''j_2})\right)^2\right] \leqslant r_e O(1).
$$

Hence,

$$
\mathbb{E}[\Delta_2] \leqslant \tilde{\sigma}^2 O\left(\|T_\lambda^{-1}k(g,\cdot)\|_{L^2}^2\right)\sqrt{r_e r_0}.
$$

For $\Delta_3$ which is the same as $\Delta_2$, if the conditional orthogonality holds, then

$$
\mathbb{E}[\Delta_3] \leqslant \tilde{\sigma}^2 O\left(\|T_\lambda^{-1}k(g,\cdot)\|_{L^2}^2\right)\sqrt{r_e r_0}.
$$

For $\Delta_4$,

$$
\mathbb{E}[\Delta_4] = \mathbb{E}\left[\epsilon_{ij_1}\left(T_\lambda^{-1}k(g_{ij_1},g) - T_\lambda^{-1}k(g_{i'j_1},g)\right)\epsilon_{ij_2}\left(T_\lambda^{-1}k(g_{ij_2},g) - T_\lambda^{-1}k(g_{i''j_2},g)\right)\right]
$$

$$
\leqslant \sqrt{\mathbb{E}\left[\epsilon_{ij_1}^2\left(T_\lambda^{-1}k(g_{ij_1},g) - T_\lambda^{-1}k(g_{i'j_1},g)\right)^2\right]\mathbb{E}\left[\epsilon_{ij_2}^2\left(T_\lambda^{-1}k(g_{ij_2},g) - T_\lambda^{-1}k(g_{i''j_2},g)\right)^2\right]}
$$

$$
\leqslant \sigma_G^2\mathbb{E}\left[\left(T_\lambda^{-1}k(g_{ij_1},g) - T_\lambda^{-1}k(g_{i'j_1},g)\right)^2\right]
$$

$$
\leqslant \tilde{\sigma}^2 r_e O\left(\|T_\lambda^{-1}k(g,\cdot)\|_{L^2}^2\right).
$$

where the last inequality repeats the same procedure in bounding $\Delta_2$.

Jointly, if the conditional orthogonality $\delta_{rs} \approx 0, \forall r, s$,

$$\mathbb{E} T_\lambda^{-1} k(g_{ij_1}, g)\epsilon_{ij_1} T_\lambda^{-1} k(g_{ij_2}, g)\epsilon_{ij_2} \leqslant \tilde{\sigma}^2 (r_e \vee r_0) O \left( \left\| T_\lambda^{-1} k(g, \cdot) \right\|_{L^2}^2 \right).$$

$\square$

# C   Auxiliary Lemmas

## C.1   Key Lemmas

**Lemma C.1.** (Lemma A.5 in [39]) *Suppose $\mathcal{H}$ has embedding index $\alpha_0$. Let $p, \gamma \geqslant 0$, $\alpha > \alpha_0$ such that $0 \leqslant 2 - \gamma - \alpha \leqslant 2p$ then*

$$\left\| T_\lambda^{-p} k(g, \cdot) \right\|_{[\mathcal{H}]^\gamma}^2 \leqslant M_\alpha^2 \lambda^{2-2p-\gamma-\alpha}, \ g \in \mathcal{G} \ almost \ everywhere.$$

**Corollary C.2.** (Corollary A.6. in [39]) *Suppose $\mathcal{H}$ has embedding index $\alpha_0$ and $\alpha > \alpha_0$. Then the following holds for $g \in \mathcal{G}$ almost everywhere*

$$\left\| T_\lambda^{-1} k(g, \cdot) \right\|_{L^\infty}^2 \leqslant M_\alpha^4 \lambda^{-2\alpha},$$

$$\left\| T_\lambda^{-1} k(g, \cdot) \right\|_{L^2}^2 \leqslant M_\alpha^2 \lambda^{-\alpha},$$

$$\left\| T_\lambda^{-1/2} k(g, \cdot) \right\|_{\mathcal{H}}^2 \leqslant M_\alpha^2 \lambda^{-\alpha}.$$

**Lemma C.3.** (Proposition B.2 in [39]) *Under Assumption 1, if $\lambda \asymp n^{-\theta}$ and $\theta \in (0, \beta)$, then for any $p \geqslant 1$, we have*

$$\mathrm{tr} \left( TT_\lambda^{-1} \right)^p \asymp \lambda^{-\frac{1}{\beta}}.$$

**Lemma C.4.** (Lemma A.3 in [37]) *Under Assumption 1, for any $0 \leqslant \gamma \leqslant s, r = 1, \ldots, d$, we have*

$$\left\| f_\lambda^{(r)} - f_\rho^{*(r)} \right\|_{[\mathcal{H}]^\gamma}^2 \asymp \begin{cases} \lambda^{s-\gamma}, & s - \gamma < 2; \\ \lambda^2 \log \frac{1}{\lambda}, & s - \gamma = 2; \\ \lambda^2, & s - \gamma > 2. \end{cases}$$

**Lemma C.5.** (Lemma A.7 in [37]) *Under Assumption 1, for any $0 \leqslant \gamma < s + 2, r = 1, \ldots, d$, we have*

$$\left\| f_\lambda^{(r)} \right\|_{[\mathcal{H}]^\gamma}^2 \asymp \begin{cases} \lambda^{s-\gamma}, & s < \gamma; \\ \log \frac{1}{\lambda}, & s = \gamma; \\ 1, & s > \gamma. \end{cases}$$

**Lemma C.6.** (Lemma B.6 in [37]) *Let $A, B$ be two positive semi-definite bounded linear operators on separable Hilbert space $\mathcal{H}$. Then*

$$\|A^s B^s\|_{\mathcal{B}(\mathcal{H})} \leqslant \|AB\|_{\mathcal{B}(\mathcal{H})}^s, \ \forall s \in [0, 1].$$

**Lemma C.7.** *Denote $\xi(g) = T_\lambda^{-\frac{1}{2}} (T_g f_\rho^* - T_g f_\lambda)$ [10]. Under Assumption 1, if $s > \alpha_0, \alpha > \alpha_0$ then*

$$\|\xi(g)\|_{\mathcal{H}} \leqslant \tilde{O} \left( \lambda^{-\alpha + \frac{\tilde{s}}{2}} \right),$$

*where $\tilde{s} = \min(s, 2)$.*

*Proof.*

$$\|\xi(g)\|_{\mathcal{H}} = \| T_\lambda^{-\frac{1}{2}} k(g, \cdot)(f_\rho^*(g) - f_\lambda(g)) \|_{\mathcal{H}}$$

$$\leqslant \| T_\lambda^{-\frac{1}{2}} k(g, \cdot) \|_{\mathcal{H}} \| f_\rho^* - f_\lambda \|_{L^\infty}$$

$$\leqslant M_\alpha \lambda^{-\frac{\alpha}{2}} \| f_\rho^* - f_\lambda \|_{L^\infty},$$

where the last inequality is obtained by Corollary C.2. Then the proof is completed by

$$\|f_\rho^* - f_\lambda\|_{L^\infty} \leqslant M_\alpha \|f_\rho^* - f_\lambda\|_{[\mathcal{H}]^\alpha} \leqslant \tilde{O} \left( \lambda^{(\tilde{s}-\alpha)/2} \right),$$

where we use Lemma C.4. $\square$

---

[10]Here we state the lemma for any $r = 1, ..., d$. For simplicity, we ignore the script $r$.

**Lemma C.8.** (Theorem 3 in [54]) *If $\mathcal{X}$ is a Hilbert space, $X \in \mathcal{X}$, and $\mathbb{E}X_j = 0$ for all $j$, then*

$$\mathbb{E}\cosh\lambda\left|\sum_{j=1}^{n} X_j\right| \leqslant \prod_{j=1}^{n} \mathbb{E}\left[e^{\lambda|X_j|} - \lambda|X_j|\right].$$

**Lemma C.9.** (Lemma B.8 in [39]) *Assuming that $\mathcal{G} \subseteq \mathbb{R}^d$ is bounded and $k(\cdot, \cdot) \in C^{0,p}(\mathcal{G} \times \mathcal{G})$ for some $p \in (0,1]$. Denote $\mathcal{K}_\lambda = \left\{T_\lambda^{-1}k(g, \cdot)\right\}_{g \in \mathcal{G}}$. Then the $\varepsilon$-covering number of $\mathcal{K}_\lambda$*

$$\mathcal{N}\left(\mathcal{K}_\lambda, \|\cdot\|_\infty, \varepsilon\right) \leqslant C'''(\lambda\varepsilon)^{-\frac{2d}{p}},$$

*where $C'''$ is a positive constant not depending on $\lambda$ or $\varepsilon$.*

**Lemma C.10.** *Assuming that $\mathcal{G} \subseteq \mathbb{R}^d$ is bounded and $k(\cdot, \cdot) \in C^{0,p}(\mathcal{G} \times \mathcal{G})$ for some $p \in (0,1]$. Denote $\mathcal{K}_{G,\lambda} = \left\{\left(T_{G\lambda}^{-1} - T_\lambda^{-1}\right)k(g, \cdot)\right\}_{g \in \mathcal{G}}$. Then with high probability, the $\epsilon$-covering number of $\mathcal{K}_{G,\lambda}$*

$$\mathcal{N}\left(\mathcal{K}_{G,\lambda}, \|\cdot\|_\infty, \varepsilon\right) \leqslant C''''(\lambda\varepsilon)^{-\frac{2d}{p}},$$

*where $C''''$ is a positive constant not depending on $\lambda$ or $\varepsilon$.*

*Proof.* For any $a, b \in \mathcal{G}$,

$$\left\|\left(T_{G\lambda}^{-1} - T_\lambda^{-1}\right)k(a, \cdot) - \left(T_{G\lambda}^{-1} - T_\lambda^{-1}\right)k(b, \cdot)\right\|_{L^\infty}$$
$$= \sup_{g \in \mathcal{G}}\left|\left(T_{G\lambda}^{-1} - T_\lambda^{-1}\right)k(a, g) - \left(T_{G\lambda}^{-1} - T_\lambda^{-1}\right)k(a, g)\right|$$
$$= \sup_{g \in \mathcal{G}}\left|\left(T_{G\lambda}^{-1} - T_\lambda^{-1}\right)k(g, a) - \left(T_{G\lambda}^{-1} - T_\lambda^{-1}\right)k(g, b)\right|.$$

Note that by the properties of RKHS,

$$\left|\left(T_{G\lambda}^{-1} - T_\lambda^{-1}\right)k(g, a) - \left(T_{G\lambda}^{-1} - T_\lambda^{-1}\right)k(g, b)\right|$$
$$\leqslant \left\|\left(T_{G\lambda}^{-1} - T_\lambda^{-1}\right)k(g, \cdot)\right\|_{\mathcal{H}} \|k(a, \cdot) - k(b, \cdot)\|_{\mathcal{H}}$$
$$= \left\|\left(T_{G\lambda}^{-1} - T_\lambda^{-1}\right)k(g, \cdot)\right\|_{\mathcal{H}} \sqrt{k(a,a) - 2k(a,b) + k(b,b)}$$
$$\leqslant \left\|\left(T_{G\lambda}^{-1} - T_\lambda^{-1}\right)k(g, \cdot)\right\|_{\mathcal{H}} \sqrt{k(a,a) - k(a,b) + k(b,b) - k(a,b)}$$
$$\leqslant \sqrt{L}\left\|\left(T_{G\lambda}^{-1} - T_\lambda^{-1}\right)k(g, \cdot)\right\|_{\mathcal{H}} \|a - b\|^{\frac{p}{2}}$$
$$\leqslant \sqrt{L}\left(\left\|T_{G\lambda}^{-1}k(g, \cdot)\right\|_{\mathcal{H}} + \left\|T_\lambda^{-1}k(g, \cdot)\right\|_{\mathcal{H}}\right)\|a - b\|^{\frac{p}{2}}$$
$$\leqslant \sqrt{L}\left(\left\|T_{G\lambda}^{-1}T_\lambda\right\|\left\|T_\lambda^{-1}k(g, \cdot)\right\|_{\mathcal{H}} + \left\|T_\lambda^{-1}k(g, \cdot)\right\|_{\mathcal{H}}\right)\|a - b\|^{\frac{p}{2}}$$
$$\overset{\mathbb{P}}{\leqslant} \sqrt{L}\kappa\lambda^{-1}\|a - b\|^{\frac{p}{2}},$$

where the third inequality use the Hölder-continuity assumption on kernel function, the last inequality uses Lemma B.1

$$\left\|T_{G\lambda}^{-1}T_\lambda\right\| = \left\|T_\lambda^{1/2}T_{G\lambda}^{-1}T_\lambda^{1/2}\right\| = O_{\mathbb{P}}(1),$$

and the kernel function is bounded by $\kappa$

$$\sup_{g \in \mathcal{G}}\|k(g, \cdot)\|_{\mathcal{H}} \leqslant \kappa.$$

Therefore, to find an $\varepsilon$-net of $\mathcal{K}_{G,\lambda}$ with respect to $\|\cdot\|_{L^\infty}$, we only need to find an $\tilde{\varepsilon}$-net of $\mathcal{G}$ with respect to the Euclidean norm, where $\tilde{\varepsilon} = \left(\frac{\epsilon\lambda}{\sqrt{L}\kappa}\right)^{\frac{2}{p}}$. Hence, the covering number

$$\mathcal{N}\left(\mathcal{K}_{G,\lambda}, \|\cdot\|_\infty, \varepsilon\right) \leqslant \mathcal{N}\left(\mathcal{G}, \|\cdot\|_{\mathbb{R}^d}, \tilde{\varepsilon}\right) \overset{\mathbb{P}}{\leqslant} C''''(\lambda\varepsilon)^{-\frac{2d}{p}}.$$

$\square$

**Lemma C.11.** (Proposition A.9 in [37]) *Under Assumption 1, for any $0 < s \leqslant \alpha_0$ and $\alpha > \alpha_0$, we have embedding*

$$[\mathcal{H}]^s \hookrightarrow L^{q_s}(\mathcal{G}, d\mu), \quad q_s = \frac{2\alpha}{\alpha - s}.$$

**Lemma C.12.** (Proposition A.11 in [37]) *Let $\psi_\lambda = \lambda(T_G + \lambda)^{-1}$. Suppose $\lambda_1 \leqslant \lambda_2$, then for any $s, p \geqslant 0$*

$$\left\|T^s\psi_{\lambda_1}^p\right\| = \left\|\psi_{\lambda_1}^p T^s\right\| \leqslant \left\|T^s\psi_{\lambda_2}^p\right\| = \left\|\psi_{\lambda_2}^p T^s\right\|.$$

## C.2 Technical Lemmas for Concentration of $k$-gap Independent Data

**Lemma C.13.** (Lemma 4 in [3]) *Let $K$ be a finite subset of positive integers. Consider a family $(\mathbb{U}_k)_{k \in K}$ of $d \times d$ self-adjoint random matrices that are mutually independent. Assume that for any $k \in K$*

$$\mathbb{E}(\mathbb{U}_k) = \mathbf{0} \quad and \quad \lambda_{\max}(\mathbb{U}_k) \leqslant B \ a.s.,$$

*where $B$ is a positive constant. Then for any $t > 0$*

$$\mathbb{E}\mathrm{tr}\left(e^{t \sum_{k \in K} \mathbb{U}_k}\right) \leqslant d \exp\left(t^2 g(tB)\lambda_{\max}\left(\sum_{k \in K} \mathbb{E}[\mathbb{U}_k^2]\right)\right),$$

*where $g(x) = x^{-2}(e^x - x - 1)$.*

**Lemma C.14.** (The Intrinsic Dimension Lemma, [67]) *Let $\phi$ be a convex function on the interval $[0, \infty)$ with $\phi(0) = 0$. For any positive semi-definite matrix $A$:*

$$\mathrm{tr}\phi(A) \leqslant \mathrm{intdim}(A)\phi(\|A\|).$$

**Lemma C.15.** (Lieb inequality) *For a fixed symmetric $n \times n$ matrix $H$ and a $n \times n$ random matrix $Z$, it holds*

$$\mathbb{E}\left[\mathrm{tr}\exp\left(H + Z\right)\right] \leqslant \mathrm{tr}\exp\left(H + \log\mathbb{E}e^Z\right).$$

We extend Lemma C.13 as follow.

**Lemma C.16.** *Under the setting of Lemma C.13,*

$$\mathbb{E}\mathrm{tr}\left(e^{t \sum_{k \in K} \mathbb{U}_k} - \mathbf{I}\right) \leqslant \mathrm{intdim}\left(\mathbb{E}\left[\sum_{k \in K} \mathbb{U}_k^2\right]\right)\exp\left(t^2 g(tB)\lambda_{\max}\left(\sum_{k \in K} \mathbb{E}\left[\mathbb{U}_k^2\right]\right)\right).$$

*Proof.* Take $\phi(A) = e^A - \mathbf{I}$,

$$\begin{aligned}
\mathbb{E}\mathrm{tr}\left(e^{t \sum_{k \in K} \mathbb{U}_k} - \mathbf{I}\right) &= \mathbb{E}\mathrm{tr}\left(e^{t \sum_{k \in K} \mathbb{U}_k}\right) - \mathrm{tr}(\mathbf{I}) \\
&\leqslant \mathrm{tr}\exp\left(\sum_{k \in K} \log \mathbb{E}e^{t\mathbb{U}_k}\right) - \mathrm{tr}(\mathbf{I}) \\
&\leqslant \mathrm{tr}\exp\left(\sum_{k \in K} \log\left(1 + t^2 g(tB)\mathbb{E}\left[\mathbb{U}_k^2\right]\right)\right) - \mathrm{tr}(\mathbf{I}) \\
&\leqslant \mathrm{tr}\exp\left(\sum_{k \in K} \log\exp\left(t^2 g(tB)\mathbb{E}\left[\mathbb{U}_k^2\right]\right)\right) - \mathrm{tr}(\mathbf{I}) \\
&= \mathrm{tr}\exp\left(t^2 g(tB)\sum_{k \in K} \mathbb{E}\left[\mathbb{U}_k^2\right]\right) - \mathrm{tr}(\mathbf{I}) \\
&= \mathrm{tr}\left(\exp\left(t^2 g(tB)\sum_{k \in K} \mathbb{E}\left[\mathbb{U}_k^2\right]\right) - \mathbf{I}\right) \\
&= \mathrm{tr}\phi\left(t^2 g(tB)\sum_{k \in K} \mathbb{E}\left[\mathbb{U}_k^2\right]\right) \\
&\leqslant \mathrm{intdim}\left(\mathbb{E}\left[\sum_{k \in K} \mathbb{U}_k^2\right]\right)\exp\left(t^2 g(tB)\lambda_{\max}\left(\sum_{k \in K} \mathbb{E}\left[\mathbb{U}_k^2\right]\right)\right),
\end{aligned}$$

where the first inequality holds by iteratively using Lieb inequality (refer to Lemma C.15), the second inequality holds by Taylor expansion, and the last inequality uses Lemma C.14. $\square$

**Lemma C.17.** (Lemma 5 in [3]) *Let $\mathbb{U}_0, \mathbb{U}_1, \cdots$ be a sequence of $d \times d$ self-adjoint random matrices. Assume that there exists positive constants $\sigma_0, \sigma_1, ..., \sigma_n$ and $\kappa_0, \kappa_1, ..., \kappa_n$ such that for $i = 1, 2, ..., n$ and any $t \in [0, \frac{1}{\kappa_i}]$*

$$\log \mathbb{E}\mathrm{tr}\left(e^{t\mathbb{U}_i}\right) \leqslant C_d + (\sigma_i t)^2/(1 - \kappa_i t).$$

*Then for any $t \in [0, \frac{1}{\kappa}]$*

$$\log \mathbb{E}\mathrm{tr}\left(e^{t\sum_{k=0}^{n} \mathbb{U}_k}\right) \leq C_d + (\sigma t)^2/(1 - \kappa t).$$

*where $\sigma = \sigma_0 + \sigma_1 + ... + \sigma_n$ and $\kappa = \kappa_0 + \kappa_1 + \kappa_n$.*

We extend Lemma C.17 as follow.

**Lemma C.18.** *Under the setting of Lemma C.17, if*

$$\log \mathbb{E}\mathrm{tr}\left(e^{t\mathbb{U}_i} - \mathbf{I}\right) \leq C_{\mathrm{intd}} + (\sigma_i t)^2/(1 - \kappa_i t),$$

*Then*

$$\log \mathbb{E}\mathrm{tr}\left(e^{t\sum_{k=0}^{n} \mathbb{U}_k} - \mathbf{I}\right) \leq (n-1)\log 3 + C_{\mathrm{intd}} + (\sigma t)^2/(1 - \kappa t).$$

*Proof.*

$$
\begin{aligned}
\mathbb{E}\left[\mathrm{tr}\left(e^{t\mathbb{U}_0 + t\mathbb{U}_1} - \mathbf{I}\right)\right] &\leq \mathbb{E}\left[\mathrm{tr}\left(e^{t\mathbb{U}_0} e^{t\mathbb{U}_1} - \mathbf{I}\right)\right] \\
&= \mathbb{E}\left[\mathrm{tr}\left(\left(e^{t\mathbb{U}_0} - \mathbf{I}\right)\left(e^{t\mathbb{U}_1} - \mathbf{I}\right) + \left(e^{t\mathbb{U}_0} - \mathbf{I}\right) + \left(e^{t\mathbb{U}_1} - \mathbf{I}\right)\right)\right] \\
&\overset{(*)}{\leq} \exp\left(C_{\mathrm{intd}} + \frac{(\sigma t)^2}{1 - \kappa t}\right) + \exp\left(C_{\mathrm{intd}} + \frac{(\sigma_0 t)^2}{1 - \kappa_0 t}\right) + \exp\left(C_{\mathrm{intd}} + \frac{(\sigma_1 t)^2}{1 - \kappa_1 t}\right) \\
&= \exp\left(C_{\mathrm{intd}}\right) \cdot \left[\exp\left(\frac{(\sigma t)^2}{1 - \kappa t}\right) + \exp\left(\frac{(\sigma_0 t)^2}{1 - \kappa_0 t}\right) + \exp\left(\frac{(\sigma_1 t)^2}{1 - \kappa_1 t}\right)\right] \\
&\leq \exp\left(C_{\mathrm{intd}} + \frac{(\sigma t)^2}{1 - \kappa t}\right) \cdot 3,
\end{aligned}
$$

where (*) is the result of Lemma C.17. Hence,

$$\log \mathbb{E}[\mathrm{tr}(e^{t\mathbb{U}_0 + t\mathbb{U}_1} - \mathbf{I})] \leq \log 3 + C_{\mathrm{intd}} + \frac{(\sigma t)^2}{1 - \kappa t}.$$

By iteration, we complete the proof:

$$\log \mathbb{E}\mathrm{tr}\left(e^{t\sum_{k=0}^{n} \mathbb{U}_k} - \mathbf{I}\right) \leq (n-1)\log 3 + C_{\mathrm{intd}} + (\sigma t)^2/(1 - \kappa t).$$

$\square$

# D  Bernstein-type Concentration for $k$-gap Independent Data

In this section, we mainly present some useful propositions for undertaking Bernstein-type concentration for $k$-gap independent data, which are quite crucial for the concentration results in Section B. This novel technique can also be used for other weakly dependent processes assuming specific mixing property, i.e. structure of the $\alpha$-mixing or $\tau$-mixing coefficient decay [52, 3].

**Proposition D.1.** *Consider a $k$-gap independent sequence of random variables $(\mathbb{X}_i)_{i=1}^{nk}$ taking values of self-adjoint Hilbert-Schmidt operators. Suppose that there exists a positive constant $M$ such that for any $i \geq 1$,*

$$\mathbb{E}[\mathbb{X}_i] = \mathbf{0} \quad and \quad \lambda_{\max}(\mathbb{X}_i) \leq M \quad almost\ surely.$$

*Denote*

$$v^2 = \sup_{K \subseteq \{1,...,nk\}} \frac{1}{\mathrm{Card}K} \lambda_{\max}\left(\mathbb{E}\left[\left(\sum_{i \in K} \mathbb{X}_i\right)^2\right]\right),$$

*and*

$$\mathrm{intd} = \mathrm{intdim}(\mathbb{E}\mathbb{X}^2).$$

*Let $A$ be a positive integer larger than 2. Then there exists a subset $K_A$ of $\{1, ..., A\}$ with $\mathrm{Card}(K_A) \geq A/2$, such that for any positive $t$ such that $tM < \frac{2}{k}$,*

$$\log \mathbb{E}\mathrm{tr}\left[\left(e^{t\sum_{i \in K_A} \mathbb{X}_i}\right) - \mathbf{I}\right] \leq \log\left(\frac{A}{2}\mathrm{intd}\right) + \frac{4 \times 3.1 t^2 A v^2}{1 - \frac{Mkt}{2}}.$$

*Proof.* The key step is to construct $K_A$. As developed in [3], the set $K_A$ will be a finite union of $2^\ell$ disjoint sets of consecutive integers with same cardinality spaced according to a recursive 'Cantor'-like construction. Let

$$\delta := \frac{\log 2}{2\log A}, \quad \ell := \ell_A = \sup\left\{j \in \mathbb{N}^* : \frac{A\delta(1-\delta)^{j-1}}{2^j} \geqslant 2k \geqslant 2\right\}.$$

Let $n_0 = A$ and for $j \in \{1, 2, \ldots, \ell\}$, define

$$n_j = \left\lceil \frac{A(1-\delta)^j}{2^j}\right\rceil \text{ and } d_{j-1} = n_{j-1} - 2n_j.$$

To construct $K_A$ we proceed as follows. At the first step, we divide the set $\{1\ldots A\}$ into three disjoint subsets of consecutive integers: $I_{1,1}$, $I_{0,1}^*$ and $I_{1,2}$. These subsets are such that $\mathrm{Card}(I_{1,1}) = \mathrm{Card}(I_{1,2}) = n_1$ and $\mathrm{Card}(I_{0,1}^*) = d_0$. At the second step, each of the sets of integers $I_{1,i}$, $i = 1, 2$ is divided into three disjoint subsets of consecutive integers as follows: for any $i = 1, 2$, $I_{1,i} = I_{2,2i-1} \cup I_{1,i}^* \cup I_{2,2i}$ where $\mathrm{Card}(I_{2,2i-1}) = \mathrm{Card}(I_{2,2i}) = n_2$ and $\mathrm{Card}(I_{1,i}^*) = d_1$. Iterating this procedure we have constructed after $1 \leqslant j \leqslant \ell_A$ steps, $2^j$ sets of consecutive integers $I_{j,i}$, $i = 1, 2, \ldots, 2^j$. The set of consecutive integers $K_A$ is then defined by

$$K_A = \bigcup_{k'=1}^{2^\ell} I_{\ell, k'}.$$

Therefore

$$\mathrm{Card}\left(\{1, \ldots, A\}\backslash K_A\right) = \sum_{j=0}^{\ell-1}\sum_{i=1}^{2^j} \mathrm{Card}\left(I_{j,i}^*\right) = \sum_{j=0}^{\ell-1} 2^j d_j = A - 2^\ell n_\ell.$$

Note that

$$A - 2^\ell n_\ell \leqslant A\left(1 - (1-\delta)^\ell\right) = A\delta\sum_{j=0}^{\ell-1}(1-\delta)^j \leqslant A\delta\ell \leqslant \frac{A}{2},$$

then

$$A \geqslant \mathrm{Card}(K_A) \geqslant A/2.$$

For simplicity, for any $k' \in \{1, \ldots, \ell\}$ and any $j \in \{1, \ldots, 2^{k'-1}\}$, we define

$$K_{k',j} := K_{A,k',j} = \bigcup_{i=(j-1)2^{\ell-k'}+1}^{j2^{\ell-k'}} I_{\ell,i}, \quad \mathbb{S}_j^{(k')} = \sum_{i\in K_{k',j}} \mathbb{X}_i.$$

Then for the reason that $d_0 \geqslant \cdots \geqslant d_{\ell-1} \geqslant \frac{A\delta(1-\delta)^{\ell-1}}{2^{\ell-1}} - 2 \geqslant 2k$, we obtain for $k' = 0, \ldots, \ell-1$, for any $t > 0$

$$\mathbb{E}\mathrm{tr}\left(e^{t\sum_{j=1}^{2^{k'}}\mathbb{S}_j^{(k')}}\right) = \mathbb{E}\mathrm{tr}\left(e^{t\sum_{j=1}^{2^{k'+1}}\mathbb{S}_j^{(k'+1)}}\right).$$

Hence, by iteration,

$$\mathbb{E}\mathrm{tr}\exp\left(t\sum_{i\in K_A}\mathbb{X}_i\right) = \mathbb{E}\mathrm{tr}\exp\left(t\sum_{j=1}^{2^\ell}\mathbb{S}_j^{(\ell)}\right).$$

The rest of the proof consists of giving a suitable upper bound for $\mathbb{E}\mathrm{tr}\exp\left(t\sum_{j=1}^{2^\ell}\mathbb{S}_j^\ell\right)$. With this aim, let $p$ be a positive integer to be chosen later such that

$$p = \left\lceil\frac{2}{tM}\right\rceil \vee \left\lceil\frac{q}{2}\right\rceil,$$

where $q = n_\ell$. Let $m_{q,p} = \lfloor q/(2p)\rfloor$, for any $j \in \{1, \ldots, 2^\ell\}$, we divide $K_{\ell,j}$ into $2m_{q,p}$ consecutive intervals $\mathbb{Z}_{j,i}^\ell$, $1 \leqslant i \leqslant 2m_{q,p}$, each containing $p$ consecutive integers plus a remainder interval $\mathbb{Z}_{j,2m_{q,p}+1}^\ell$ containing $r$ consecutive integers with $r = q - 2pm_{q,p} \leqslant 2p - 1$. With this notation,

$$\mathbb{S}_j^{(\ell)} = \sum_{i=1}^{m_{q,p}+1}\mathbb{Z}_{j,2i-1}^{(\ell)} + \sum_{i=1}^{m_{q,p}}\mathbb{Z}_{j,2i}^{(\ell)}.$$

Since $\mathrm{tr} \circ \exp$ is convex, we get

$$\mathbb{E}\mathrm{tr}\exp\left(t\sum_{j=1}^{2^\ell}\mathbb{S}_j^{(\ell)}\right) \leqslant \frac{1}{2}\mathbb{E}\mathrm{tr}\exp\left(2t\sum_{j=1}^{2^\ell}\sum_{i=1}^{m_{q,p}+1}\mathbb{Z}_{j,2i-1}^{(\ell)}\right) + \frac{1}{2}\mathbb{E}\mathrm{tr}\exp\left(2t\sum_{j=1}^{2^\ell}\sum_{i=1}^{m_{q,p}}\mathbb{Z}_{j,2i}^{(\ell)}\right).$$
(D.1)

Note that the gap between $\{\mathbb{Z}_{j,2i-1}^{(\ell)}\}$ and $\{\mathbb{Z}_{j,2i}^{(\ell)}\}$ is $p$, if

$$\frac{2}{tM} \geqslant k \text{ and } \frac{q}{2} \geqslant k,$$
(D.2)

then $\{\mathbb{Z}_{j,2i-1}^{(\ell)}\}$ and $\{\mathbb{Z}_{j,2i}^{(\ell)}\}$ are mutually independent, respectively. For the first condition in (D.2), we set $tM \leqslant \frac{2}{k}$, while for the second condition in (D.2), note that

$$q = n^l \geqslant \frac{A}{2^{\ell+1}}, \quad \frac{A\delta(1-\delta)^{\ell-1}}{2^\ell} \geqslant 2k,$$

hence,

$$\frac{q}{2} \geqslant \frac{A}{4\cdot 2^\ell} \geqslant \frac{A\delta}{2\cdot 2^\ell} \geqslant \frac{A\delta(1-\delta)^{\ell-1}}{2\cdot 2^\ell} \geqslant k.$$

After undertaking the decomposition (D.1), we are going to bound $\mathbb{E}\mathrm{tr}\exp\left(t\sum_{j=1}^{2^\ell}\mathbb{S}_j^\ell\right)$ by bounding each term in (D.1) using Lemma C.16. To be specific, given that

$$2\lambda_{\max}(\mathbb{Z}_{j,2i-1}^{(\ell)}) \leqslant 2Mp \leqslant \frac{4}{t} \text{ almost surely,}$$

by Lemma C.16, we obtain

$$\mathbb{E}\mathrm{tr}\left(\exp\left(2t\sum_{i=1}^{2^\ell}\sum_{i=1}^{m_{q,p}+1}\mathbb{Z}_{j,2i-1}^{(\ell)}\right) - \mathbf{I}\right) \leqslant \mathrm{intdim}\left(\mathbb{E}\left[\sum\left(\mathbb{Z}_{j,2i-1}^{(\ell)}\right)^2\right]\right)\exp(4\times 3.1\times At^2v^2),$$

and

$$\mathbb{E}\mathrm{tr}\left(\exp\left(2t\sum_{i=1}^{2^\ell}\sum_{i=1}^{m_{q,p}+1}\mathbb{Z}_{j,2i}^{(\ell)}\right) - \mathbf{I}\right) \leqslant \mathrm{intdim}\left(\mathbb{E}\left[\sum\left(\mathbb{Z}_{j,2i}^{(\ell)}\right)^2\right]\right)\exp(4\times 3.1\times At^2v^2).$$

Note that $\mathrm{intdim}(A+B) \leqslant \mathrm{intdim}(A) + \mathrm{intdim}(B)$ and $\mathrm{intd} = \mathrm{intdim}(\mathbb{E}\mathbb{X}^2)$, we have

$$\mathbb{E}\mathrm{tr}\left(\exp\left(2t\sum_{i=1}^{2^\ell}\sum_{i=1}^{m_{q,p}+1}\mathbb{Z}_{j,2i-1}^{(\ell)}\right) - \mathbf{I}\right) \leqslant \frac{A}{2}\mathrm{intd}\exp(4\times 3.1\times At^2v^2),$$

and

$$\mathbb{E}\mathrm{tr}\left(\exp\left(2t\sum_{i=1}^{2^\ell}\sum_{i=1}^{m_{q,p}+1}\mathbb{Z}_{j,2i}^{(\ell)}\right) - \mathbf{I}\right) \leqslant \frac{A}{2}\mathrm{intd}\exp(4\times 3.1\times At^2v^2).$$

Therefore,

$$\mathbb{E}\mathrm{tr}\left[\exp\left(t\sum_{i\in K_A}\mathbb{X}_i\right) - \mathbf{I}\right]$$

$$=\mathbb{E}\mathrm{tr}\left[\exp\left(t\sum_{j=1}^{2^\ell}\mathbb{S}_j^{(\ell)}\right) - \mathbf{I}\right]$$

$$\leqslant\frac{1}{2}\mathbb{E}\mathrm{tr}\left[\exp\left(2t\sum_{j=1}^{2^\ell}\sum_{i=1}^{m_{q,p}+1}\mathbb{Z}_{j,2i-1}^{(\ell)}\right) - \mathbf{I}\right] + \frac{1}{2}\mathbb{E}\mathrm{tr}\left[\exp\left(2t\sum_{j=1}^{2^\ell}\sum_{i=1}^{m_{q,p}}\mathbb{Z}_{j,2i}^{(\ell)}\right) - \mathbf{I}\right]$$

$$\leqslant\frac{A}{2}\mathrm{intd}\exp(4\times 3.1\times At^2v^2).$$

$\square$

**Proposition D.2.** *Consider a $k$-gap independent sequence of random variables $(\mathbb{X}_i)_{i=1}^{nk}$ taking values of self-adjoint Hilbert-Schmidt operators. Suppose that there exists a positive constant $M$ such that for any $i \geqslant 1$,*

$$\mathbb{E}[\mathbb{X}_i] = \mathbf{0} \quad and \quad \lambda_{\max}(\mathbb{X}_i) \leqslant M \quad almost\ surely.$$

*Denote*

$$v^2 = \sup_{K \subseteq \{1,\ldots,nk\}} \frac{1}{\operatorname{Card}K} \lambda_{\max}\left(\mathbb{E}\left[\left(\sum_{i \in K} \mathbb{X}_i\right)^2\right]\right),$$

*and*

$$\operatorname{intd} = \operatorname{intdim}(\mathbb{E}\mathbb{X}^2).$$

*Then for any positive $t$ such that $tM < \frac{1}{k}\frac{1}{\log n}$,*

$$\log \mathbb{E}\operatorname{tr}\left(\exp\left(t\sum_{i=1}^{nk}\mathbb{X}_i\right) - \mathbf{I}\right) \leqslant \log n \log 3 + \log\left(\frac{nk}{2}\operatorname{intd}\right) + t^2 nk v^2 \frac{169}{1 - tMk\log n}.$$

*Proof.* Let $A_0 = A = nk$, and $\mathbb{Y}^{(0)}(i) = \mathbb{X}_i$, $i = 1, \ldots, A_0$. Let $K_{A_0}$ be the discrete Cantor type set as defined from Proposition D.1. Let $A_1 = A_0 - \operatorname{Card}(K_{A_0})$ and define for any $j = 1, \ldots, A_1$,

$$\mathbb{Y}^{(1)}(j) = \mathbb{X}_{i_j}, \text{ where } \{i_1, \ldots, i_{A_1}\} = \{1, \ldots, A\}\backslash K_A.$$

Now for $i \geqslant 1$, let $K_{A_i}$ be defined from $\{1, \ldots, A_i\}$ exactly as $K_A$ is defined from $\{1, \ldots, A\}$. Set $A_{i+1} = A_i - \operatorname{Card}(K_{A_i})$ and $\{j_1, \ldots, j_{A_{i+1}}\}\backslash K_{A_i}$. For $s = 1, \ldots, A_{i+1}$, define

$$\mathbb{Y}^{(i+1)}(s) = \mathbb{Y}^{(i)}(j_s).$$

Set $L = L_n = \inf\{j \in \mathbb{N}^*, A_j \leqslant 2k\}$. Then the following decomposition clearly holds,

$$\sum_{j=1}^{nk} \mathbb{X}_j = \sum_{i=0}^{L-1}\sum_{j \in K_{A_i}} \mathbb{Y}^{(i)}(j) + \sum_{j=1}^{A_L} \mathbb{Y}^{(L)}(j).$$

Let

$$\mathbb{U}_i = \sum_{j \in K_{A_i}} \mathbb{Y}^{(i)}(j) \text{ for } 0 \leqslant i \leqslant L-1 \text{ and } \mathbb{U}_L = \sum_{j=1}^{A_L} \mathbb{Y}^{(L)}(j),$$

By proposition D.1, for any positive $t$ such that $tM < \frac{2}{k}$,

$$\log \mathbb{E}\operatorname{tr}\left(\exp(t\mathbb{U}_i) - \mathbf{I}\right) \leqslant \log\left(\frac{nk2^{-i}}{2}\operatorname{intd}\right) + \frac{4 \times 3.1t^2 nk2^{-i}v^2}{1 - \frac{Mkt}{2}}, \ i = 0, 1, \ldots, L-1 \quad \text{(D.3)}$$

Note that

$$\lambda_{\max}(\mathbb{U}_L) \leqslant MA_L \leqslant 2kM,$$

By Lemma C.16, for any positive $t$ such that $tM < \frac{1}{k}$,

$$\log \mathbb{E}\operatorname{tr}\left(\exp(t\mathbb{U}_L) - \mathbf{I}\right) \leqslant \log(\operatorname{intdim}(\mathbb{E}\mathbb{U}_L^2)) + 2kt^2v^2 \leqslant \log(2k\operatorname{intd}) + \frac{2kt^2v^2}{1 - Mkt}. \quad \text{(D.4)}$$

At last, we aggregate Equation (D.3) and (D.4) by Lemma C.18. Let

$$\kappa_i = \frac{Mk}{2}, i = 0, \ldots, L-1; \ \kappa_L = Mk,$$

$$\sigma_i = 2^{1-\frac{i}{2}}v\sqrt{3.1nk}, i = 0, \ldots, L-1; \ \sigma_L = v\sqrt{2k}.$$

Note that $\frac{nk}{2^{L-1}} \geqslant A_{L-1} \geqslant 2k$, we have $L \leqslant \log n$. Then

$$\sum_{i=1}^{L} \kappa_i = \frac{Mk(L+1)}{2} \leqslant Mk\log n,$$

$$\sum_{i=1}^{L} \sigma_i \leqslant \frac{2\sqrt{3.1nkv}}{1 - \frac{1}{\sqrt{2}}} + v\sqrt{2k} \leqslant 13\sqrt{nkv}.$$

Therefore, for any positive $t$ such that $tM < \frac{1}{k}\frac{1}{\log n}$,

$$\log \mathbb{E}\mathrm{tr}\left(\exp\left(t\sum_{i=1}^{nk} \mathbb{X}_i\right) - \mathbf{I}\right) \leqslant \log n \log 3 + \log\left(\frac{nk}{2}\mathrm{intd}\right) + t^2 nkv^2\frac{169}{1 - tMk\log n}.$$

(D.5)

$\square$

**Proposition D.3.** *Consider the setting in Proposition D.2. Consider the case $d = 1$. There exists a constant $C''$ such that for any positive $t$ such that $tM < \frac{1}{k\log n}$,*

$$\log \mathbb{E}\exp\left(t\sum_{i=1}^{nk} \mathbb{X}_i\right) \leqslant \frac{C''t^2 nkv^2}{1 - tMk\log n},$$

*where*

$$v^2 = \sup_{K \subseteq \{1,\dots,nk\}} \frac{1}{\mathrm{Card}K}\lambda_{\max}\left(\mathbb{E}\left[\left(\sum_{i \in K} \mathbb{X}_i\right)^2\right]\right).$$

*Proof.* This result can be obtained by (D.5) in Proposition D.2, where we replace Lemma C.16 and Lemma C.18 with Lemma C.17 and Lemma C.13, and take $d = 1$. $\square$

# E  Conditional Orthogonality Condition

In this section, we exemplify the conditional orthogonality condition. For simplicity, we merely prove the scalar case, that is, $d = 1$, where the vector case can be generalized trivially. We present some examples as follows.

**Example E.1.** (Additive Gaussian distribution and Hermite polynomial)

$$x \sim N(0, \sigma_x^2), \ u \sim N(0, 1), \ g = \sqrt{\alpha_t}x + \sqrt{1 - \alpha_t}u, \ e_i(g) = \frac{H_i\left(\frac{g}{\sigma_g}\right)}{\sqrt{i!}},$$

*where $\sigma_g^2 = \alpha_t\sigma_x^2 + 1 - \alpha_t$ and $H_i(\cdot)$ is the Hermite polynomial.*

*Proof.* Denote $Z = \frac{g - \sqrt{1-\alpha_t}u}{\sqrt{\alpha_t}\sigma_x}$ then

$$Z|u \sim N(0, 1).$$

Hence,

$$\mathbb{E}\left[e_i(g)|u\right] = \frac{1}{\sqrt{i!}}\mathbb{E}\left[H_i\left(\frac{\sqrt{1-\alpha_t}u}{\sigma_g} + \frac{\sqrt{\alpha_t}\sigma_x}{\sigma_g}Z\right)|u\right].$$

We then focus on computing

$$\mathbb{E}\left[H_i\left(\frac{\sqrt{1-\alpha_t}u}{\sigma_g} + \frac{\sqrt{\alpha_t}\sigma_x}{\sigma_g}Z\right)|u\right] := \mathbb{E}\left[H_i\left(a + bZ\right)\right],$$

where

$$a = \frac{\sqrt{1-\alpha_t}u}{\sigma_g}, b = \frac{\sqrt{\alpha_t}\sigma_x}{\sigma_g}, Z \sim N(0, 1).$$

By the definition of Hermite polynomial,

$$\sum_{n=0}^{\infty} \frac{t^n}{n!}H_n(a + bZ) = e^{t(a+bZ) - \frac{t^2}{2}}.$$

Taking expectation over $Z$, we have

$$RHS = e^{at + \frac{b^2-1}{2}t^2}.$$

By Taylor expansion,

$$e^{at + \frac{b^2-1}{2}t^2} = \sum_{n=0}^{\infty} \frac{t^n}{n!} a^n \sum_{m=0}^{\lfloor \frac{n}{2} \rfloor} (b^2-1)^m a^{-2m} \frac{n!}{m!(n-2m)!2^m}.$$

Hence, by matching with

$$\sum_{n=0}^{\infty} \frac{t^n}{n!} H_n(a+bZ),$$

we obtain

$$\mathbb{E}\left[H_i(a+bZ)\right] = \sum_{m=0}^{\lfloor \frac{i}{2} \rfloor} (b^2-1)^m a^{i-2m} \frac{i!}{m!(i-2m)!2^m}.$$

Setting

$$c^2 = 1 - b^2 = 1 - \frac{\alpha_t \sigma_x^2}{\alpha_t \sigma_x^2 + 1 - \alpha_t} = \frac{1-\alpha_t}{\alpha_t \sigma_x^2 + 1 - \alpha_t} = \frac{a^2}{u^2},$$

then

$$\mathbb{E}[e_i(g)|u] = \frac{1}{\sqrt{i!}} \mathbb{E}\left[H_i(a+bZ)\right]$$

$$= \frac{1}{\sqrt{i!}} \sum_{m=0}^{\lfloor \frac{i}{2} \rfloor} (b^2-1)^m a^{i-2m} \frac{i!}{m!(i-2m)!2^m}$$

$$= \frac{c^i}{\sqrt{i!}} \sum_{m=0}^{\lfloor \frac{i}{2} \rfloor} (-1)^m u^{i-2m} \frac{i!}{m!(i-2m)!2^m}.$$

Note that by the definition of Hermite polynomial,

$$H_i(x) = \sum_{m=0}^{\lfloor \frac{i}{2} \rfloor} (-1)^m x^{i-2m} \frac{i!}{m!(i-2m)!2^m}.$$

Hence,

$$\mathbb{E}[e_i(g)|u] = \frac{c^i}{\sqrt{i!}} H_i(u).$$

At last,

$$\mathbb{E}_u \left[\mathbb{E}[e_i(g)|u]\mathbb{E}[e_j(g)|u]\right] = \frac{c^{i+j}}{\sqrt{i!j!}} \mathbb{E}_u \left[H_i(u)H_j(u)\right] = 0.$$

$\square$

**Example E.2.** (Additive Uniform distribution on a cyclic group and discrete Fourier basis) $x, u$ *satisfy the Uniform distribution on a cyclic group* $\mathbb{Z}_n = \{0, 1, 2, \ldots, n-1\}$, $g = x + u \bmod n$, $e_j(g) = \frac{1}{\sqrt{n}} w^{jg}, w = e^{2\pi i/n}, j = 0, 1, \ldots, n-1$.

*Proof.* For $j \neq 0$

$$\mathbb{E}_x[e_j(g)] = \mathbb{E}_x \left[\frac{1}{\sqrt{n}} w^{j(x+u)}\right] = \frac{w^{ju}}{\sqrt{n}} \frac{1}{n} \sum_{x=0}^{n-1} w^{jx} = 0.$$

$\square$

# F Discussion on Assumptions

## F.1 Polynomial-decay Kernel Spectrum

The polynomial spectrum assumption makes our bound clearer and facilitates direct comparison with established results in the i.i.d. setting (consistent with prior works [37, 39]). This makes the core theoretical insights more accessible. While the assumption simplifies presentation, our framework

is readily extensible to general spectra on the technical level. In particular, the key terms (e.g. $\text{tr}(TT_\lambda^{-1})^p$, $\left\|f_\lambda^{(r)}\right\|[\mathcal{H}]^{\gamma^2}$ and $\left\|f_\lambda^{(r)} - f_\rho^{*(r)}\right\|[\mathcal{H}]^{\gamma^2}$ in Lemma C.3-C.5) can be expressed directly in terms of individual eigenvalues $(\lambda_1, \lambda_2, \dots)$ rather than the decay rate $\beta$. For instance, the norm $\|f_\lambda\|_{[\mathcal{H}]^\gamma}^2$ is fundamentally given by a series (shown below) that depends on the full eigenvalue sequence: $\|f_\lambda\|_{[\mathcal{H}]^\gamma}^2 \asymp \sum_{i=1}^\infty \left(\frac{\lambda_i^p}{\lambda_i + \lambda}\right)^2 i^{-1}$, $p = (s + 2 - \gamma)/2$. Therefore, the polynomial decay is primarily a tool for deriving clearer, more interpretable bounds without fundamentally limiting the scope of our technical approach. We believe it best serves the goal of presenting our core theoretical contributions transparently.

## F.2 Relative Smoothness

In deriving our general theoretical bound (4.1), we can relax $s > 1$ to $s > 0$ and obtain exactly the same result, as we have technically leveraged assumptions and properties of interpolation spaces for refinements. While for the specified bounds under conditional orthogonality ((4.2) and Theorem 4.4), $s > 1$ is required to estimate the relevance parameter $r_T$ (Lemma B.11) for providing a concise bound and clear insights, where the relevance parameter is explicitly related to $\alpha_t$. Technically, the smoothness on $f_\rho^*$ enables continuity to convert the relevance in the function space into the relevance $r_0$ in the data space. We consider this specific case to make our conclusion more clear and understandable. Indeed, without the strong smoothness $s > 1$, we can also provide estimation (less concise expression) for $r_T$. We merely need to replace $r_0$ with $r_\rho := \frac{\mathbb{E}\left[\left(f_\rho^{(r)*}(g_{ij}) - f_\rho^{(r)*}(g_{i'j})\right)^2\right]}{4\mathbb{E}\left[f_\rho^{(r)*}(g_{ij})^2\right]} \in$ $[0, 1]$, maintaining all convergence guarantees.

## F.3 Hölder continuity

The primary purpose of the Hölder continuity assumption is to eliminate the need for the often unrealistic sub-Gaussian design assumption in deriving our general bounds [37, 39]. Technically, it is essential for establishing a uniform concentration bound via covering number estimates (Lemma C.9 and C.10). Furthermore, this assumption allows us to derive concise estimates for the relevance parameter in specific scenarios, such as when conditional orthogonality holds. We note that Hölder continuity is naturally satisfied by important kernel classes like the Laplace kernel, Sobolev kernels, and Neural Tangent Kernels [37, 39].

# G Experiment

## G.1 Real Image Diffusion Training

To demonstrate the applicability of our method beyond toy examples, we conducted an ablation study on the CIFAR-10 dataset. We trained a diffusion model using a dataset of 1024 samples for 100 epochs with a batch size of 1024, optimized using Adam with a learning rate of 2e-3. The model architecture was a two-layer U-Net, and time conditioning was implemented by expanding the time variable $t$ and concatenating it as an additional input channel to the image.

We report the diffusion loss on the test set with a size of 1024 at $t = 1.0$ and $t = 0.1$ across different values of $k$ (number of noisy realizations per data). Each configuration was evaluated over 100 parallel runs to ensure robustness.

As shown in Figure 3a, increasing $k$ consistently improves performance at $t = 1$, indicating better fitting of the complex score function. However, it also leads to degradation at $t = 0.1$, consistent with our empirical findings on the MoG settings in the paper. Both our empirical findings on the MoG settings and CIFAR-10 align well with our theory that when $t$ is large (i.e., noise dominates), increasing $k$ is beneficial to generalization.

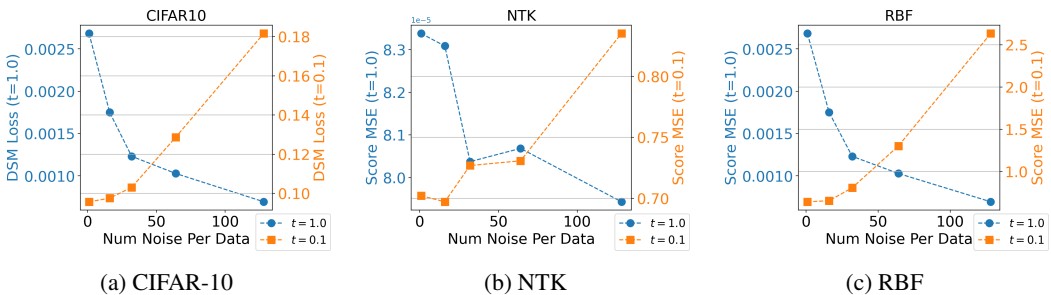

| (a) CIFAR-10 | (b) NTK | (c) RBF |

Figure 3: Score estimation error versus *the number of noise per data*, i.e., $k$, for two noise levels.

## G.2 Kernel Ridge Regressor

To supplement, we conducted the same experiments as the **Numerical Experiments** in Section 4 using both NTK and RBF kernel regressor. As shown in Figure 3b and Figure 3c, the results show consistent trends with our MLP findings.

## G.3 Experiment Details

The paper fully discloses all the information, including training and testing details, needed to reproduce the main experimental results of the paper to the extent that it affects the main conclusions of the paper, as described in Section G.1, Section G.2 and **Numerical Experiments** in Section 4. All data are either synthetically generated with detailed description or publicly open dataset. The experimental results report statistical information including mean and standard deviation in Fig.2. All experiments are conducted using a single 4090 GPU.

# H Broader Impacts

- Our theory provides a general framework to characterize the learnability of different data distributions. Practitioners can leverage this framework as follows: first, select a kernel appropriate to the problem domain; second, check the decay rate of the kernel's spectrum; and finally, apply Theorem 4.4 to rigorously determine (i) whether the distribution can be learned efficiently and (ii) the sample complexity required for convergence.

- Our results provide practical insights for optimizing the training efficiency of diffusion models, suggesting that adaptive noise-sample pairing strategies may offer significant computational benefits.

- Our general-purpose concentration technique advances the theoretical toolkit for dependent data analysis and may find applications beyond our current setting, which is of independent interest to the community.

