# OpenReview forum: "Kernel Regression in Structured Non-IID Settings: Theory and Implications for Denoising Score Learning"
_NeurIPS.cc/2025/Conference — NeurIPS 2025 poster_

### Official Review · Reviewer_APPF · 2025-06-06

**Clarity:** 3
**Significance:** 3
**Originality:** 3
**Rating:** 5
**Confidence:** 3

**Summary:**

This paper considered the model $y_{i}^{j} = f^*(g_{i}^{j}) + \epsilon_{i}^{j}$, $i \leq n$, $j \leq k$. The authors further assumed that there exists a function $g$, such that for each group $j$, the inputs are generated as $g_{i}^{j} = g( x_{i}, u_{i}^{j})$, where i.i.d. signals $\{x_i\}$ are independent of i.i.d. noises $\{u_{i}^{j}\}$.

This paper then obtained an upper bound on the convergence rate of learning curve (i.e., the excess risk w.r.t. different choices of regularization parameter $\lambda$) of KRR under common assumptions.

**Questions:**

- 1) What is the motivation of setting $r \leq d$ in line 172? I notice that $r$ systems are trained independently, and both the theoretical analysis and the example in Section 4.1 consider the special case $r = 1$.

- 2) There might be typos in (2) and possibly other equations: The bias convergence rate should be $n^{-\min (s, 2)\theta}$; and the $\Theta$ notation should be $O$ unless both upper and lower bounds are established.

- 3) The paper would benefit from a more comprehensive discussion of relevant prior work in several areas:
    - (i) Optimal rate of KRR and kernel gradient flow (deep neural networks, that the authors mentioned, are more closely related to kernel gradient flow). E.g., Optimal rates for the regularized least-squares algorithm; On the optimality of misspecified kernel ridge regression; Generalization Error Curves for Analytic Spectral Algorithms under Power-law Decay.
    - (ii) saturation effect of KRR (the appearance of $\min$\{$s$, 2\} in Theorem 4.1 is related to it). E.g., Optimal rates for the regularized least-squares algorithm; ON THE SATURATION EFFECT OF KERNEL RIDGE REGRESSION.
    - (iii) high-dimensional KRR and KGF. E.g., Just interpolate: Kernel “Ridgeless” regression can generalize; linearized two-layers neural networks in high dimension; on the Saturation Effects of Spectral Algorithms in Large Dimensions; On the Pinsker bound of inner product kernel regression in large dimensions

**Ethical Concerns:**

["NO or VERY MINOR ethics concerns only"]

**Final Justification:**

I believe the authors have addressed my main concerns.

**Limitations:**

yes

**Quality:**

3

**Strengths And Weaknesses:**

Strengths:
- i) The authors considered the non-i.i.d. data settings, which is new within the KRR literature. They also provide two real-world examples (Examples 3.1 and 3.2) to illustrate the relevance of this setting.
- ii) The paper is carefully written and overall enjoyable to read.
- iii) The proof is novel compared with existing works. For example, the authors used a blockwise decomposition method for $k$-gap independent random sequence as an alternative to Bernstein-type concentration inequalities typically used for i.i.d. data (e.g., Lemma B.4 in [31]).
- iv) The results in this paper recovered known results in the i.i.d. case when $k = 1$.

Weaknesses:
- i) The authors assume that the target function $f^* \in L^2$ lies in the interpolation space $[\mathcal{H}]^s$ with $s \geq 1$. In particular, when $s = 1$, this reduces to $f^* \in \mathcal{H}$, i.e., the RKHS itself. This assumption may be restrictive when dimension $d$ is large. For example,
    - Consider the Sobolev RKHS $W^{m,2}(\mathcal{X})$ with $\mathcal{X} \subset \mathbb{R}^{d}$ and $m > d/2$. Its interpolation space with $s>0$ is given by $ [W^{m,2}(\mathcal{X})]^s \cong W^{m s , 2}(\mathcal{X})$, and functions in $W^{ms,2}(\mathcal{X})$ are weak differentiable of order $m s$.
    - On one hand, if we assume that $f_{\star} \in \mathcal{H}$, then it requires that $f_{\star}$ is weak differentiable of order $m>d/2$, which becomes increasingly unrealistic as $d$ grows.
    - On the other hand, if we consider the interpolation space with $s>0$, then we can adopt a weaker assumption $f_{\star} \in W^{m s , 2}(\mathcal{X})$, allowing the results to cover less smooth functions.

    It would be helpful if the authors could discuss the challenges associated with analyzing the case $s < 1$, and whether these challenges are analogous to those in the i.i.d. setting.

- ii) The experiments involve a three-layer ReLU MLP. Since there is currently no theoretical approximation framework between MLPs and kernel gradient flow (KGF) / KRR in the non-i.i.d. setting, it is better to use KRR instead.

---

> ### Author Rebuttal · Authors · 2025-07-31
>
> Thanks for your time and efforts reviewing our paper. We now address raised questions as follows.
>
> * Q1: The target function $f^*$ lies in the interpolation space $[\mathcal{H}]^s$ with $s\geq 1$ may be restrictive. / Discuss the challenges associated with analyzing the case $s < 1$, and whether these challenges are analogous to those in the i.i.d. setting.
>
>     Thanks for your comment. The smoothness condition on $f_\rho^*$ appears in some existing work (Assumption (B) in [1], Lemma A.12 in [4]). In fact, for the derivation of our general theoretical bound (the first statement in Theorem 4.1, Eq.(2)), **we can relax our assumption to $s>0$** and obtain exactly the same result, as we leverage assumptions and properties of interpolation spaces for refinements on the technical level.
>
>     While for the specified bounds under conditional orthogonality (the second statement in Theorem 4.1, Eq.(3) and Theorem 4.4), $s\geq 1$ is required to estimate the relevance parameter $r_T$ (Lemma B.11) for providing a concise bound and clear insights, where the relevance parameter is explicitly related to $\alpha_t$. Technically, the smoothness on $f_\rho^* $ enables continuity to convert the relevance in the function space into the relevance $r_0$ in the data space (line 813-815). We consider this specific case to make our conclusion more clear and understandable. Indeed, without the strong smoothness $s\geq 1$, we can also provide estimation (less concise expression) for $r_T$. We merely need to replace $r_0$ with $r_\rho:=\frac{\mathbb{E}\left[\left(f _ \rho^{(r) * }(g_{ij})-f_\rho^{(r)* }(g_{i'j})\right)^2\right]}{4\mathbb{E}\left[f_\rho^{(r)* }(g_{ij})^2\right]}\in [0,1]$, maintaining all convergence guarantees.
>
> * Q2: The experiments involve a three-layer ReLU MLP. Since there is currently no theoretical approximation framework between MLPs and kernel gradient flow (KGF) / KRR in the non-i.i.d. setting, it is better to use KRR instead.
>
>     We thank the reviewer for the insightful comment regarding the use of MLPs. To supplement，we conducted the same experiments using both **NTK** and **RBF** kernels. The results show consistent trends with our MLP findings:
>
>     | **k**   | **NTK Loss at t = 1.0** | **NTK Loss at t = 0.1** | **RBF Loss at t = 1.0** | **RBF Loss at t = 0.1** |
>     |--------:|------------------------:|-------------------------:|------------------------:|-------------------------:|
>     | 1       | 8.3379e-05              | 0.7020                   | 0.00268199              | 0.6412                   |
>     | 16      | 8.3086e-05              | 0.6971                   | 0.00175084              | 0.6532                   |
>     | 32      | 8.0374e-05              | 0.7268                   | 0.00123094              | 0.8061                   |
>     | 64      | 8.0684e-05              | 0.7307                   | 0.00102835              | 1.3014                   |
>     | 128     | 7.9432e-05              | 0.8351                   | 0.00069417              | 2.6334                   |
>
>     These results reinforce both our theoretical insights and the empirical findings from the MLP experiments presented in the paper: **increasing \( k \)** consistently improves performance at \( t = 1.0 \), where the target function is simpler, but leads to degraded accuracy at \( t = 0.1 \), where the function exhibits greater complexity. These empirical findings align well with our theory that when $t$ is large (i.e., noise dominates), increasing $k$ is beneficial to generalization.
>
> * Q3: What is the motivation of setting $r \leq d$ in line 172? I notice that $r$ systems are trained independently, and both the theoretical analysis and the example in Section 4.1 consider the special case $r = 1$.
>
>     Thanks for your comment. Considering $r = 1, \dots, d$ allows our framework to handle real-world vector-output tasks like denoising score learning (where outputs correspond to spatial/gradient dimensions). This generalizes prior scalar-output analyses to broader applications. While on the technical level, the vector output analysis can be reduced to $r=1$ for the reason that noise is independent per dimension. This enables provably equivalent analysis via parallel single-dimensional KRR. Focusing on $r=1$ in analysis maintains clarity in proofs and notation, and preserves all key insights.
>
> * Q4: There might be typos in (2) and possibly other equations: The bias convergence rate should be $n^{-\min (s, 2)\theta}$; and the $\Theta$ notation should be $O$ unless both upper and lower bounds are established.
>
>     Thanks for your comment. The bias convergence rate is indeed $n^{-\min (s, 2)\theta}$, we will modify these typos and the usage of asymptotic notations in the revised manuscript.
>
> * Q5: A more comprehensive discussion of relevant prior work.
>
>     Thanks for your suggestion. We will include the following discussion of related prior work in the revised manuscript.
>
>     * (i) Optimal rate of KRR and kernel gradient flow.
>
>         There are rich literature showing the optimal rate of KRR and kernel gradient flow. [3] demonstrates the minimax optimality of KRR when the regression lies in the RKHS and proves the best upper bound of the generalization error is $n^{-\frac{2\beta}{2\beta+1}}$. [4] further extends the result to the misspecified cases when the regression function does not lie in the RKHS. More recently, [5] studies the generalization error curves of a large class of analytic spectral algorithms, including the kernel gradient method, where the regressor $\hat{f}_\lambda$ is obtained by a filter function. Our paper initials the study of the generalization behavior of KRR in the structured non-i.i.d. setting and defers the study of analytic spectral algorithms to future.
>
>     * (ii) Saturation effect of KRR.
>
>         The saturation effect refers to the phenomenon that the kernel ridge regression (KRR) fails to achieve the information theoretical lower bound when the smoothness of the underground truth function exceeds certain level. To be specific, [2] shows that the information theoretical lower bound of the generalization error is $n^{-\frac{s\beta}{s\beta+1}}$ while [3] shows that the best upper bound of the generalization error is $n^{-\frac{2\beta}{2\beta+1}}$. This gap has been widely observed in practices: no matter how carefully one tunes the KRR, the rate of the generalization error can not be faster than $n^{-\frac{2\beta}{2\beta+1}}$. More recently, [1] rigoriously prove the saturation effect by providing a lower bound matching the existing upper bound $n^{-\frac{2\beta}{2\beta+1}}$. In our paper, let $\tilde{r}+\frac{1-\tilde{r}}{k} \asymp n^{-l}$, then the optimal $\theta$ for generalization error is $\theta_{opt}=\frac{(1+l)\beta}{\beta \mathrm{min}(s,2)+1}$ and the optimal generalization error rate is $n^{-\frac{(1+l)\beta \mathrm{min}(s,2)}{\beta \mathrm{min}(s,2)+1}}$. On the one hand, when $l>0$ (implying that $\tilde{r}$ is extremely small and $k$ is extremely large), the saturation effect still holds but with a faster rate. On the other hand, when $l=0$ (e.g. k=1), our result matches the previous result in the i.i.d. setting.
>
>     * (iii) high-dimensional KRR and KGF.
>
>         While kernel regression with a fixed data dimension $d$ has been extensively studied, there are rising attentions to the case when dealing with large-dimensional data. In the large-dimensional setting where $n \asymp d^\gamma$ with $\gamma>0$, [6] finds that for the square-integrable regression function, KRR and kernel gradient flow are consistent if and only if the regression function is a polynomial with a low degree. Another line of work e.g. [7], considers the overfitting behavior of kernel interpolation. On the technical level, [7] utilizes the argument that high-dimensional random kernel matrices can be approximated in spectral norm by linear kernel matrices plus a scaled identity. Regarding the saturation effect, [8] explores this effects for a large class of spectral algorithms (including the KRR, gradient descent, etc.) in large dimensional settings, by providing minimax lower bound and the exact convergence rates. Besides, [9] explores the Pinsker bound problem for kernel regression models that incorporate large-dimensional inner product kernels defined on the sphere $\mathbb{S}^d$, by addressing the scenario where the sample size $n$ is given by $\alpha d^\gamma (1+o_d(1))$ for some $\alpha,\gamma >0$. Our paper defers the analysis in high-dimensional data scnario to future.
>
> We hope above response can address your concern and we are open to discuss more if any question still hold.
>
> [1] ON THE SATURATION EFFECT OF KERNEL RIDGE REGRESSION.
>
> [2] Optimal rates for the regularized learning algorithms under general source condition.
>
> [3] Optimal rates for the regularized least-squares algorithm.
>
> [4] On the optimality of misspecified kernel ridge regression.
>
> [5] GENERALIZATION ERROR CURVES FOR ANALYTIC SPECTRAL ALGORITHMS UNDER POWER-LAW DECAY.
>
> [6] Linearized Two-Layers Neural Networks in High Dimension.
>
> [7] Just Interpolate: Kernel “Ridgeless” Regression Can Generalize.
>
> [8] On the Saturation Effects of Spectral Algorithms in Large Dimensions.
>
> [9] ON THE PINSKER BOUND OF INNER PRODUCT KERNEL REGRESSION IN LARGE DIMENSIONS.

---

> > ### Comment · Reviewer_APPF · 2025-08-01
> >
> > Thank you for your response. I believe you have addressed my main concerns:
> >
> > - You claimed that Theorem 4.1 can be extended to interpolation spaces with $s < 1$ (the original statement only covers $s \ge 1$). Consequently, Theorem 4.1 matches the convergence rates for KRR under i.i.d. setting as given in [31]. I attempted to verify this claim:
> >     - Under i.i.d. setting, most steps hold for any $s>0$, expect for a key bias‐term estimate (see, e.g., Lemma A.5 of [31] for $s\ge1$ and Lemma A.10 of [31] for $s<1$), since $\|f_\rho^*\|_\infty$ is only guaranteed finite when $s \ge 1$.
> >     - In this manuscript, an analogous estimate appears in Lemma B.2, and its proof indeed distinguishes the cases \(s\ge1\) and \(s<1\).
> >
> > - I see you have added two KRR experiments that convincingly validate your theoretical findings.
> >
> > - You have answered my other questions and enriched the KRR literature review.
> >
> > Moreover, I reviewed the other reviewers’ comments and your replies, and I have no further concerns.  Nevertheless, I agree with Reviewer aGuy that the assumptions deserve further explanation, and the manuscript should undergo one more proofreading pass to catch any remaining typos.
> >
> > Based on the above, I decide to raise my score from 4 to 5.  I hope the authors would implement all promised modifications in the revised manuscript.

---

> > > ### Author Response · Authors · 2025-08-01
> > >
> > > Dear Reviewer APPF,
> > >
> > > We are glad to hear that our rebuttal has addressed your concerns, and we sincerely appreciate your decision to raise the score to 5. In particular, thank you for emphasizing the key detalis in the smoothness assumption $s>0$. We will improve this part, provide more comprehensive explanations and discussions (for our assumptions, related work, experiment, setup, technique, etc.), and check the typos carefully in the revised manuscript.
> > >
> > > Thank you again for your effort in reviewing our work. We will implement all promised modifications in the revised manuscript.
> > >
> > > Best,
> > >
> > > Authors

---

### Official Review · Reviewer_aGuy · 2025-06-23

**Clarity:** 1
**Significance:** 3
**Originality:** 3
**Rating:** 4
**Confidence:** 2

**Summary:**

This paper proposes upper bounds on the risk of scalar kernel ridge regression in the misspecified setting (meaning that the optimal regressor is in a specific power space of the RKHS) with block-independent data. The first theorem (4.1) provides the upper bound, showing polynomial decrease rates of the population risk if the regularization parameters decreases slowly enough. In contrast, the second theorem establishes that the risk is lower bounded by a positive constant if the regularization parameter decreases too sharply, establishing the absence of generalization (which I understand as consistency of the estimator).
The paper then applies the obtained rates to conclude on the successful learning of the denoising score algorithm under theoretical assumptions. This use case is the motivational application for the theory, and is illustrated in experiments on a toy problem.
On a technical level, the main novelty seems to reside in a block argument to recover the case of iid variables (cf. Section 5.1).

**Questions:**

My questions are detailed in the section "Strengths and weaknesses". In particular:
1. How do your results relate to the works mentioned?
2. Is there a reason to considering the "mild" non-iid-ness of block-independent data compared to mixing data, and how do your proof techniques relate to those for mixing data?
3. Can you clarify Assumption 1 and the assumptions of $f_\rho$? In particular, is it assumed to lie in $\mathcal H$ and, if not, how is this consistent with Assumption 1?
4. Can you provide guidelines or related works on how to design neural networks to abide by the assumptions of Theorem 4.4?

**Ethical Concerns:**

["NO or VERY MINOR ethics concerns only"]

**Final Justification:**

The authors have addressed my technical concerns, and I trust that they will implement the promised changes (mainly, regarding Assumption 1).
Nonetheless, I believe the exposition of this paper can be thoroughly improved and recommend rewriting it to emphasize different parts of the contribution to improve readability; the current emphasis to denoising score learning could be kept but relegated to an appendix. I also find the proofs very hard to follow and was not able to check them in details given my limited familiarity with the tools they leverage. This is the reason for a **borderline** accept.

**Limitations:**

The paper does not satisfyingly identify the limitations of its Assumption 1, leading to an overblown claim of lack of assumptions on the ground truth, and the limitations of Theorem 4.4 of the difficulty/impossibility for neural networks to abide by the assumptions of the theorem, leading to an overblown claim of practicality.
Other than that, the authors adequately address the potential negative societal impact of their work.

**Quality:**

3

**Strengths And Weaknesses:**

This paper is well-motivated and generally well-written (modulo the issues raised below). It proposes interesting ideas, including in the extensive appendix. However, I have significant concerns regarding several aspects: the completeness of the related work discussion, the meaningfulness of the theoretical setup, the adequacy of the assumptions relative to the claimed contributions, and the validity of the application to denoising score learning. I also raise concerns about some aspects of the writing.

Given the scope and depth of these issues, I believe that fully addressing them within the rebuttal period may be challenging. That said, I remain open to discussion and would welcome a constructive exchange.

1. **Related work** The paper does not sufficiently discuss its relation to the following bodies of work:
	1. Classical literature on statistical learning of support vector machines (SVM) [R1]: a central goal in statistical learning is to derive generalization guarantees in the form of high confidence bounds on the excess risk. While the paper does acknowledge that it is a central question in KRR (which is a special case of SVMs), it does not acknowledge the classical theory like that of [R1]
	2. Learning with SVMs from non-iid data [R2,R3,R4]: there is a rich literature on learning from non-iid data, either asymptotically for general losses [R2,R3] or in the finite-sample case with mixing assumptions [R4,R5] or without (under assumptions on the loss and kernel) [R6,R7]. I believe the results on mixing are of particular interest to the authors, because block-independent processes _are_ mixing. This connects to one of my concern below about the fact that the non-iid-ness considered in this work is mild.
	3. Time-uniform concentration inequalities in bandits and Bayesian optimization: there is a class of concentration results bounding the error between the target function and the KRR estimate pointwise in the input [R8, Corollary 3.17], [R9, R10]. While these results originate from a different community and are not immediately applicable to the present use case (well-specified case, no asymptotic analysis), they should be mentioned in the related work and may be of interest to the authors.
2. **Setup** The non-iid-ness considered in the general paper is mild, in the sense that simple subsampling recovers iid data, or more generally that it implies any form of mixing, which is a very standard assumption to handle dependent data in learning theory.
In particular, what seems to be the main novelty in the proof (decomposing the data in independent blocks) reminds me of standard arguments in $\beta$-mixing tracing back to [R11] (see [R12] for a more recent reference on this applied to learning theory, and [R13] for an application to kernel methods). The more restricted setup considered in the paper may be sufficient for denoising score matching, but I would encourage the authors to justify why they consider this particular form of non-iid-ness compared to a form of mixing.
3. **Assumptions**
	1. The paper provides insufficient details on the assumptions imposed on the kernel function. In particular, the assumptions for RKHS elements to be square-integrable and for the Mercer decomposition to exist, be well-behaved, and yield an ONB of $\mathcal H$ instead of an ONS [48] are not stated. Furthermore, the domain and range of the operator $T$ (and related operators) need to be specified. Are they operators on $L_2$? On $\mathcal H$? Similarly, the paper should make explicit at the beginning of 3.2 already what set $f_\rho$ is assumed to lie in, as this may impose further restrictions on the kernel (e.g., if $f_\rho$ is assumed square integrable at this point). Additionally, the space $[\mathcal H]^t$ (which the paper calls "interpolation spaces" but is actually the power RKHS, cf. [48, Section 4]) is not always well-defined; only the actual interpolation space is [48, Eq. (36)] (and it is not a space of functions, but one of equivalence classes). I do not think these imprecisions impact the results, but they need to be addressed.
	2. I would like to emphasize that the classical convention [48] in the Mercer decomposition is that the family $(\lambda^{1/2}e_i)_i$ is an ONS in $\mathcal H$, and **not** the family $(e_i)$ (which is an ONS when embedded in $L_2$, but is not normalized in $\mathcal H$). A different convention is used in the main text (l. 178). This is not critical as it is ultimately a convention, and I also believe this is a typo and that the authors meant to use the classical convention (given comments such as l.183). I will assume this from now on, as it is important for my next point.
	3. More importantly, I have significant concerns regarding the first two points of Assumption 1. Indeed, I claim that the combination of the first and second points imply that $f_\rho\in\mathcal H$ if $s>1$ (see below). As a result, Assumption 1 imposes a **very strong** smoothness condition on $f_\rho$ (RKHS membership), which directly contradicts the storyline and claims of the paper and makes unnecessary all the refinements leveraging interpolation spaces. Technically, the membership results from the additional term $i^{-1/2}$ assumed in the expansion of $f_\rho$, which deviates from standard source conditions in the misspecified setting for KRR (interpolation space membership, which is usually expressed via the expansion in Assumption 1 without the $i^{-1/2}$ factor and assuming that $a_i$ is square summable instead of bounded above and below, cf. [48, Eq. (36)]). Finally, the assumption that $a_i\geq c>0$ is also nonstandard, and further restricts possible values for $f_\rho$ (in particular, $f_\rho$ cannot be $0$, as no coefficient may be $0$ in the expansion). I encourage the authors to revise this assumption or the claims of the paper.
	4. I am also surprised by the form of the polynomial eigenvalue decay in Assumption 1.
	In particular, I have only encountered the condition $c_\beta i^{-\beta}\leq \lambda_i$ to show _lower bounds_ on the excess risk [R14, Section 5]. A discussion of why this lower bound is needed for the present results would be welcome.
4. **Application** The application to denoising score learning in Theorem 4.4 reduces to applying Theorem 4.1 with a particular form of measurement model ($g$). Yet, the classical method (and the present numerical experiment) relies on a neural network for the regressor, and _not_ on KRR. In other words, Theorem 4.1 is only valid if the chosen neural network can represent all functions of the RKHS $\mathcal H$ (and only those); i.e., if it behaves as KRR. This is highly unrealistic, and yet the paper does not discuss this aspect, and instead claims that the result can be applied for practitioners to perform rigorous kernel engineering. I believe this is an overclaim, as it would require an analysis of how to design the neural network should be chosen to approximate the RKHS and of the incurred error due to the approximation. I am not convinced such an analysis would be interesting, however, as one would certainly be better off using KRR rather than approximating it with a neural network. Summarizing, I believe that the claims on the relevance of Theorem 4.4 in practice and the discussion of its limitations should be adapted in light of this discussion.

**Proof of $f_\rho\in\mathcal H$** Define $a_i := a_i^{(r)}$, and assume that $(\lambda_{i}^{1/2}e_{i})$ is an ONB of $\mathcal H$ and that $s>1$. Then, the series used to define $f_\rho$ converges in $\mathcal H$. Indeed: $$
\begin{split}\sum_{i}\lVert a_i \lambda_i^{s/2} e_i\rVert^2_k
&= \sum_i a_i^2\lambda_i^si^{-1}\lambda_i^{-1}\\
&\leq \max(c^2,C^2)\sum_i \lambda_i^{s-1}i^{-1}\\
&\leq \max(c^2,C^2)\cdot C_\beta^{s-1}\cdot \sum_i i^{-\beta(s-1)-1}.\end{split}
$$
The RHS is finite as long as $\beta(s-1)+1>1$, which holds since $s>1$.
As a result, the general term of the series is square summable, implying that the series converges in norm in $\mathcal H$ and that $f_\rho\in\mathcal H$.


**Minor comments**
* The terminology "the signal" to designate the input is nonstandard; I recommend using "the input"
* I recommend avoiding self-congratulatory expressions like "advanced technique" (l. 67)
* I am not fond of notations like $R \leq \Theta(\dots)$, $R \geq \Omega(\dots)$, or $R\leq \mathcal O(\dots)$. Indeed, the symbols $\Theta$, $\Omega$, and $\mathcal O$ are _already_ inducing orderings, as $f\geq g$ and $g = \mathcal O(h)$ implies that $f = \mathcal O(h)$. While this is merely style for $\mathcal O$ and $\Omega$, it becomes misleading for $\Theta$, as the notation $f \leq \Theta(g)$ may have a different meaning than $f=\Theta(g)$. I recommend using the equality symbol everywhere.
* You should avoid informal statements like "roughly" (l.71) in formal statements if they are not rigorously defined/made precise later.
* The notation $\mu_\mathcal G$ (l. 141) is undefined (I suspect it should be $\rho$). I recommend redefining $\mu_\mathcal G:= \rho$, however.
* The change of notations in Example 3.2 seems unnecessary
* It would be helpful to clarify why KRR is applied independently per dimension rather than using a multi-output variant (possibly with a diagonal kernel if the proofs don't generalize).
* l.203: I believe that "$f_\rho^{(r)}\in\mathcal H$" is a typo, as the paper tries to address the misspecified case (up to the points raised in my comment 3.2). This is also present l.813
* Other undefined symbols: $\lambda$ in Theorem 1 (undefined at this point) and $\sigma_\epsilon^2$,  $\sigma_{\epsilon_{1,2}}$, $\sigma$, and $\sigma_G$ ll.191--193
* The notation $g$ is overloaded and ambiguous. It is sometimes used as a random variable on which expectations are taken (cf. (1)), sometimes as a function (l.145), sometimes as an element of $\mathcal X\times\mathcal U$, and sometimes as multiple of these options (cf. the equation ll.192--193). I encourage the authors to clarify this.
* l.141: I don't understand what is meant by $\mu_\mathcal G(g(x,u)) = \mu_\mathcal X(x)\mu_\mathcal U(u)$, as measures should take sets as inputs.

[R1] Steinwart and Christmann, Support vector machines, 2008

[R2] Steinwart et al., Learning from dependent observations, 2009

[R3] Massiani et al., On the consistency of kernel methods with dependent observations, 2024

[R4] Steinwart and Christmann, Fast learning from non-iid observations, 2009

[R5] Hang and Steinwart, Fast learning from $\alpha$-mixing observations, 2014

[R6] Ziemann and Tu, Learning with little mixing, 2022

[R7] Simchowitz et al., Learning without mixing (...), 2018

[R8] Abbasi-Yadkori, Online Learning for Linearly Parametrized Control Problems, 2013

[R9] Chowdhury and Gopalan, On kernelized multi-armed bandits, 2017

[R10] Chowdhury and Gopalan, No-regret algorithms for multi-task bayesian optimization, 2021

[R11] Yu, Rates of convergence for empirical processes of stationary mixing sequences, 1997

[R12] Mohri and Rostamizadeh, Stability Bounds for Stationary ϕ-mixing and β-mixing Processe, 2010

[R13] Kuznetsov and Mohri, Generalization bounds for non-stationary mixing processes, 2017

[R14] Li et al., Optimal learning rates for regularized conditional mean embedding, 2022

---

> ### Author Rebuttal · Authors · 2025-07-31
>
> Thanks for your time and efforts reviewing our paper. We now address raised questions as follows.
>
> * Q1: Related works.
>
>     Thanks for your question. We will add statistical learning of SVM [R1] and time-uniform concentration inequalities [R8-R10] in our revised manuscript and focus on discussing the non-i.i.d. analysis [R2,R3,R6,R7] here (deferring the discussion of learning under mixing assumptions [R4,R5,R11-R13] to Q2).
>
>     * **Setting**
>
>         A line of work established the consistency of SVMs or kernel methods under some specific assumptions e.g. processes satisfying a law of large numbers [R2], or empirical weak convergence [R3]. While the consistency can be characterized under such strong non-i.i.d.-ness, it remains unclear of the convergence speed. Another line of work focuses on the regression over trajectories generated by a dynamic system, both linear [R6] and non-linear [R7]. Though these approaches are without mixing, the surrogate trajectory assumptions limit its applicability to other settings. Our work aim to capture the asymptotic convergence rate for $k$-gap independent data.
>
>     * **Technique**
>
>         [R6,R7] utilize single-layer block decomposition and handle each blocks by Mendelson’s small-ball method, eschewing the use of standard mixing-time arguments. Regarding the specific $k$-gap independence, our technique relies on a new-designed block scheme which is free of neither Mendelson’s small-ball method nor standard mixing-time arguments. Our novel technique enables us to eliminate the approximation error between dependent blocks and their independent surrogates, revealing the benefit of data relevance (detailed in Q2).
>
> * Q2: Setup.
>
>     * Q2.1: Reason to consider the block-independence.
>
>         Thanks for your question. We clarify the difference from those related works [R4,R5,R11-R13] as follows, whose techniques cannot resolve our setting.
>
>         * **Setting**
>
>             While many works have derived concentration high-probability bounds for non-i.i.d. data under the geometrically mixing assumption [R4,R5] or the algebraically mixing assumption [R11-R13], our $k$-gap independent case **cannot be covered by these works**, as the correlation between data points will remain high as long as they are from the same group with size $k$. Therefore, our $k$-gap independent case requires a fundamantally different non-i.i.d. data analysis.
>
>         * **Intuition**
>
>             Most of the existing results [R11-R13] perform analysis under the intuition to control the approximation error between independent blocks and dependent blocks by mixing coefficients. That being said, the data dependency is treated as an bad effect in their analysis and will finally appear as an error term. However, in our setting, we do not treat the data dependency a bad effect but aim to discover some benefits for reducing the error, otherwise it would be intractable to prove the vanishing generalization error in our setting for general $k$. To this end, we perform refined study in decomposing blocks to eliminate direct error approximation.
>
>         * **Insight**
>
>             Our analysis demonstrates the benefit of data relevance, which is in totally contrast to current non-i.i.d. data analysis. We believe some of our techniques and methodologies can be potentially extended to a broader settings (e.g., the size of each group could vary).
>
>     * Q2.2: Proof techniques.
>
>         Thanks for your question. Our proof technique differs fundamentally from mixing data approaches [R11-R13] in both decomposition strategy and error handling. While all methods employ block decomposition, the **block scheme** is different. Prior work does not perform refined study on the **intra-block randomness**, as they use single-layer decomposition that requires constructing independent surrogate blocks and controlling approximation errors. The crude study of the intra-block randomness **fails to characterize the role of data dependence** in our $k$-gap independent case. Besides, **the size of block** in existing methods **fails to capture the underlying independence** in our specific $k$-gap independent setting, limiting its applicability.
>
>         In contrast, to overcome the above weakness, we introduce a more refined decomposition, yielding multiple partitions with several independent blocks respectively. This enables us to apply standard symmetrization techniques for each partition individually and resolve cross-partition dependencies through our novel incorporating lemma (Lemma C.20). Only by this technique, can we fully utilize the underlying data independence and intra-block dependence, and derive a preciser Bernstein-type high probability bound consistent with the i.i.d. case (up to some logarithm terms).
>
> * Q3: Assumptions.
>     * Q3.1: Kernel function / Convention(l. 178).
>
>         Thanks for your good points. Operators related to $T$ are mappings: $L^2\to L^2$. We will provide clear characterizations of other concepts according to [48] in the revised manuscript.
>
>     * Q3.2: The smoothness condition on $f_\rho^*$.
>
>         Thanks for your comment. The smoothness condition on $f_\rho^*$ appears in some existing work (Lemma A.12 in [31]). In fact, In deriving our general theoretical bound Eq.(2), **we can relax $s\geq 1$ to $s>0$** and obtain exactly the same result, as we  have technically leveraged assumptions and properties of interpolation spaces for refinements.
>
>         While for the specified bounds under conditional orthogonality (Eq.(3) and Theorem 4.4), $s\geq 1$ is required to estimate the relevance parameter $r_T$ (Lemma B.11) for providing a concise bound and clear insights, where the relevance parameter is explicitly related to $\alpha_t$. Technically, the smoothness on $f_\rho^* $ enables continuity to convert the relevance in the function space into the relevance $r_0$ in the data space (line 813-815). We consider this specific case to make our conclusion more clear and understandable. Indeed, without the strong smoothness $s\geq 1$, we can also provide estimation (less concise expression) for $r_T$. We merely need to replace $r_0$ with $r_\rho:=\frac{\mathbb{E}\left[\left(f_\rho^{(r)* }(g_{ij})-f_\rho^{(r)* }(g_{i'j})\right)^2\right]}{4\mathbb{E}\left[f_\rho^{(r)* }(g_{ij})^2\right]}\in [0,1]$, maintaining all convergence guarantees.
>
>         Regarding the terms $i^{-\frac{1}{2}}$ and $a_i\geq c>0$ in this assumption, we follow the expression in the existing result [4] to provide a direct and clear comparison with the i.i.d. setting.
>
>     * Q3.3: The polynomial eigenvalue decay assumption.
>
>         Thanks for your comment. The polynomial spectrum makes our bound clearer and facilitates direct comparison with established results in the i.i.d. setting [31]. This makes the core theoretical insights more accessible. While the assumption simplifies presentation, our framework is readily extensible to general spectra on the technical level. The key terms (e.g. Lemma C.3-C.5) can be expressed directly in terms of individual eigenvalues ($\lambda_1, \dots$) rather than the decay rate $\beta$. For instance, the norm $\Vert f_\lambda\Vert_{[\mathcal{H}]^\gamma}^2$ is fundamentally given by a series (shown below) that depends on the full eigenvalue sequence: $\Vert f_\lambda\Vert_{[\mathcal{H}]^\gamma}^2\asymp\sum_{i=1}^\infty\left(\frac{\lambda_i^p}{\lambda_i+\lambda}\right)^2i^{-1}.$ Therefore, the polynomial decay is primarily a tool for deriving clearer, more interpretable bounds without fundamentally limiting the scope of our technical approach.
>
> * Q4: Application to denoising score learning / Design neural networks.
>
>     Thanks for your question. Our theoretical bounds are derived on KRR to provide insights and inspirations for practical data usage and training. In particular, our theoretical findings that noise multiplicity $k$ should scale with the noise level align well with the intuition that when the noise dominates in the observation, practitioners should add more noise rather than the raw data. Therefore, we believe this finding holds whatever learner is and apply our theory on practical neural network training. / The main contributions and insights of our theoretical results are to provide guidance for data sampling instead of designing neural network to approximate KRR. With our theoretical insights on KRR and empirical validations on neural network training, we believe our work can inspire practical training broadly.
>
> * Q5：Minor Comments
>
>     Thanks for your comments. We apologize for some typos and misunderstandings. For some inappropriate expressions such as "signal", "advanced technique", "roughly", typos "$f_\rho^* \in \mathcal{H}$" and some confused notations such as the asymptotic notations, $\lambda$ in Theorem 1, $\mu_G$, Example 3.2, $g$, we will modify those in the revised manuscript. We clarify other misunderstandings as follows.
>
>     * Q5.1: KRR is applied independently per dimension.
>
>         Thanks for your comment. On the one hand, applying KRR independently per dimension yields cleaner proofs and more interpretable bounds (with standard i.i.d. results [31]) and insights. On the other hand, in practical denoising score learning example, noise is standardly assumed independent per dimension, where multi-output variant case can be reduced to ours.
>
>     * Q5.2: Second moment terms.
>
>         Thanks for your good points. $\sigma_\epsilon$ and $\sigma^2$ control the decay of the population noise conditionally on data, with the former controlling the exponential decay (sub-Gaussian norm) and the latter controlling the quadratic decay (second moment). In parallel, for technical reasons to analyze dependent data, we extend $\sigma_\epsilon$ and $\sigma$ which are single-data conditioning to $\sigma_{\epsilon_{1,2}}$ and $\sigma_G^2$ which are dependent-data conditioning.
>
> We hope above response can address your concern and we are open to discuss more if any question still hold.

---

> ### Author Response · Authors · 2025-08-05
> **Further Response 1/2**
>
> We are glad that our rebuttal has addressed most of your other concerns. Thank you once again for your thoughtful follow-up questions. We would like to take this chance to address raised questions as follows.
>
> * **Q1: Are you claiming that Theorem 4.1 holds under $s>0$?**
>
>     Thanks for your question. Under the relaxation to $s>0$, the first statement (Eq. (2)) holds true and the second statement (Eq.(3)) holds by replacing $r_0$ with $r_\rho:=\frac{\mathbb{E}\left[\left(f_\rho^{(r)* }(g_{ij})-f_\rho^{(r)* }(g_{i'j})\right)^2\right]}{4\mathbb{E}\left[f_\rho^{(r)* }(g_{ij})^2\right]}\in [0,1]$. Previous assumption $s\geq 1$ is used to estimate the relevance parameter $r_T$ (Lemma B.11) for providing a concise bound and clear insights.
>
> * **Q2: Can you elaborate on why your $k$-gap independence assumption is not covered by standard mixing assumptions?**
>
>     Thanks for your question. For simplicity, we take $\phi$-mixing as an example. We would like to clarify that $\phi$-mixing is a very general assumption, which asssumes $\lim_{i\to \infty} \phi(i)=0$. Our $k$-independent setting can be covered by such general mixing condition with $\phi(i)=0,\ i\geq k$. However, **our analysis derives tighter concentration bound** over $k$-gap dependent data compared with previous work, yielding a generalization bound that **fully captures the benefit of data relevance**.
>
>     In the bias-variance excess risk analysis, the key to bound the variance term is to perform concentration for $V=\frac{1}{n^2k^2 }\left[\sum _ {i=1}^{nk} Z_i \right]^2=\frac{1}{n^2k^2 }\sum_ {i,j} Z_iZ_j  $, where $Z_i $ are some correlated random variables. To bound the associated differences under $\phi$-mixing conditions, previous concentration techniques treat the data dependency as a bad effect, yielding the following concentration inequality (Theroem 8 in [R12]) for random variables $W_1,...,W_m$ satisfing mixing condition with $\phi(i)$: $$P\left[|\sum_{i=1}^m W_i-E[\sum_{i=1}^m W_i]|\geq\epsilon\right]\leq 2\exp(\frac{-2\epsilon^2}{mc^2[1+\sum_{i=1}^{m}\phi(i)]^2}).$$ Applying this result on $V$ and let $\phi(i)$ denote the mixing coefficient of the sequence $\\{Z_i Z_j\\} _ {i=1...,nk; \ j=1,...,nk}$, we have $P(|V-E[V]|\geq \epsilon)\leq 2\exp(\frac{-2\epsilon^2n^4k^4}{n^2k^2c^2[1+\sum_{i=1}^{n^2k^2}\phi(i)]^2})$, i.e., $|V-E[V]|\leq O_P(\frac{1+\sum_{i=1}^{n^2k^2}\phi(i)}{nk})$.
>
>     In contrast, to uncover the precise dependency among data points, we develop a **new concentration inequality** for the random variables $W_1,\dots,W_m$ under our setting (i.e., $k$-gap dependent data) by (1) **capturing the underlying data independence** and (2) **utilizing the intra-block randomness**, $$P\left[|\sum_{i=1}^m W_i-E[\sum_{i=1}^m W_i]|\geq\epsilon\right]\leq 2\exp(\frac{-\epsilon^2/2}{mc'v^2}),$$where $c'$ is some constant and $v^2=\sup_{K\subset\\{1,...,m\\}}\frac{1}{{\rm Card}K}\mathbb{E}\left[\big(\sum_{i\in K} W_i-E[\sum_{i\in K} W_i]\big)^2\right]$(see Proposition D.3). Furthermore, to better use our concentration inequality and develop a sharper bound on the variance error, we **decompose** $V$ into $V_1,V_2$ and $V_3$ given the specific data structure and apply our technique separately to bound each concentration error $E_i=P(|V_i-E[V_i]|\geq \epsilon), i=1,2,3$. In particular, our results show that $E_1 \leq 2\exp(\frac{-\epsilon^2}{c''/n^3k^2})$, $E_2\leq 2\exp(\frac{-\epsilon^2 }{c''(k-1)^2/n^3k^2})$, $E_3\leq 2\exp(\frac{-\epsilon }{c''(kr_T+1-r_T)/nk})$, where $c''$ is some constant. Overall, the dominating error $2\exp(\frac{-\epsilon }{c''(kr_T+1-r_T)/nk})$ yields the high probability bound $|V-E[V]|\leq O_P(\frac{kr_T+1-r_T}{nk})$, where $r_T$ characterizes the correlation of data (see Definition 3.2).
>
>     For comparison, note that $\phi(i)$ may be difficult to estimate precisely, we could merely bound $\phi(i)$ as a **constant level** under general condition (see [1]). In this sense, **our high probability bound $O_P(\frac{kr_T+1-r_T}{nk})$ is tighter than the one derived from existing technique, i.e., $O_P(nk)$**, which is nearly meaningless. Moreover, we notice that **even the previous concentration technique is applied to our decomposition method**, the final high probability bound $|V-E[V]| \leq O_P(\frac{(1+\sum_{i=1}^{k} \phi(i))^2}{nk})\leq O_P(\frac{k}{n})$ is yet looser than our current result $O_P(\frac{kr_T+1-r_T}{nk})$, as $r_T\in[0,1]$ (here $\phi(i)$ is the mixing coefficient of the sequence $\\{Z_i\\}_{i=1}^{nk}$).
>
>     In conclusion, our analysis provides tighter concentration bound over $k$-gap dependent data compared with previous work and develops a novel decomposion of the variance error analysis, yielding a generalization bound that can precisely captures the data relevance. We believe our techniques can also be potentially extended to a broader settings and we will discuss the comparison between our technique and prior work in the revised version.

---

> ### Author Response · Authors · 2025-08-05
> **Further Response 2/2**
>
> * **Q3: Theorem 4.4 should mention that the algorithm used is KRR / The theory holds for KRR / The conclusions for NNs are only heuristic based on this understanding.**
>
>     Thanks for your comment. We will modify Theorem 4.4 by mentioning that the algorithm used is KRR and the conclusions for NNs are heuristic based on this understanding in the revised manuscript. In numerical experiment, we have added the experiments using both **NTK** and **RBF** kernels. The results show consistent trends with our MLP findings:
>
>     | **k**   | **NTK Loss at t = 1.0** | **NTK Loss at t = 0.1** | **RBF Loss at t = 1.0** | **RBF Loss at t = 0.1** |
>     |--------:|------------------------:|-------------------------:|------------------------:|-------------------------:|
>     | 1       | 8.3379e-05              | 0.7020                   | 0.00268199              | 0.6412                   |
>     | 16      | 8.3086e-05              | 0.6971                   | 0.00175084              | 0.6532                   |
>     | 32      | 8.0374e-05              | 0.7268                   | 0.00123094              | 0.8061                   |
>     | 64      | 8.0684e-05              | 0.7307                   | 0.00102835              | 1.3014                   |
>     | 128     | 7.9432e-05              | 0.8351                   | 0.00069417              | 2.6334                   |
>
>     These results reinforce both our theoretical insights and the empirical findings from the MLP experiments presented in the paper: **increasing \( k \)** consistently improves performance at \( t = 1.0 \), where the target function is simpler, but leads to degraded accuracy at \( t = 0.1 \), where the function exhibits greater complexity. We will revise our theorem 4.4 and include these new experimental results in the revision for more consistent explanation.
>
> We hope above response can address your concern and we are open to discuss more if any question still hold.
>
> [1] On the Computation of Mixing Coefficients
> Between Discrete-Valued Random Variables.

---

> > ### Comment · Reviewer_aGuy · 2025-08-05
> >
> > I thank the authors for their extensive follow-up answer, particularly regarding Q2.
> > This discussion leads me to saying that the main claims of the article should be revised to account for it.
> > In particular, the first contribution point (l.88) should reflect that the contribution is on the **modified bound** compared to analysis under classical non-iid assumptions such as mixing.
> > Furthermore, I thoroughly encourage them to include some of the above discussion in the main text to highlight the qualitative difference between the bounds they obtain and those obtained via mixing arguments.
> > Furthermore, I trust the authors that they will implement the promised changes regarding Assumption 1.
> >
> > That said, I still have some concerns on the paper; mainly expository. In particular, I believe that the new Bernstein bound should occupy a more prominent place in the paper, and come before its application to KRR. Additionally, the application to denoising score learning seems anecdotical and can be relegated to the appendix in my opinion (and also not be a part of the title). This would enable the authors to discuss more their new bound and its implications, beyond only a summary of its proof. It would also greatly simplify notations, by removing the need for the multi-indexing present in most of the paper (by focusing, for instance, on scalar KRR for the application to KRR). Finally, the proofs in the appendix are hard to follow (references to multiple lemmas, very heavy notation) and would benefit from being streamlined (though I realize I am not their target audience given my limited familiarity with the tools involved).
> >
> > I also have the following last question:
> > * You use repeatedly the expression "$\alpha>\alpha_0$ being sufficiently close" in Appendix B. What does it mean? Sufficiently close to what? I recommend making this explicit at least once in the paper.
> >
> > Summarizing, the above comments lead me to raise my score to a borderline accept (4), as I believe the paper is technically solid and contains new results, but its exposition can be substantially improved.

---

> > > ### Author Response · Authors · 2025-08-05
> > >
> > > Dear Reviewer aGuy,
> > >
> > > We are glad that our rebuttal has addressed most of your other concerns. Thank you once again for your thoughtful follow-up questions and comments and we sincerely appreciate your decision to raise the score.
> > >
> > > We would like to clarify the condition "$\alpha>\alpha_0$ being sufficiently close" follows from previous settings and assumptions [1,2]. In line 183-185, we introduce the embedding index $\alpha_0$ of $\mathcal{H}$, which characterizes the embedding property whether $[\mathcal{H}]^{\alpha}$ can be continuously embedded into $L^\infty(\mathcal{G},\mu_\mathcal{G})$:
> > > $$\alpha_0=\inf\left\\{\alpha: \left\|[\mathcal{H}]^\alpha\hookrightarrow L^\infty(\mathcal{G},\mu_\mathcal{G})\right\|:=\operatorname{\mathrm{ess~sup}}_ {g\in\mathcal{G},\mu_\mathcal{G}}\sum_{i\in N}\lambda_i^\alpha e_i(g)^2=M_\alpha<\infty\right\\}.
> > > $$
> > > Generally, with this definition, we can derive bounds for some norms (e.g., $\left \Vert T_{\lambda}^{-p}k(g,\cdot)\right \Vert_{[\mathcal{H}]^{\gamma}}^{2}$ in Lemma C.1) where these upper bounds generally hold for some $\alpha>\alpha_ 0$. As **the bound is tighter when $\alpha$ is smaller**, we claim "$\alpha>\alpha_0$ being sufficiently close" in some of our lemmas to guarantee a sharper bound. Such a condition enables us to obtain a final bound which can be expressed by $O^{\rm poly}(\lambda^{-\alpha_0}(\frac{r_T}{n}+\frac{1-r_T}{nk}))$.
> > >
> > > For other comments, we will follow your suggestions to modify the first contribution, include our discussions on mixing conditions and assumptions, adjust the descriptions of the new Bernstein bound and the application to denoising score learning in the revised manuscript. Moreover, we will also streamline our notations and proofs to make them more easy to follow.
> > >
> > > Thank you again for your effort in reviewing our work. We will implement all promised modifications in the revised manuscript.
> > >
> > > Best,
> > >
> > > Authors
> > >
> > > [1] On the Asymptotic Learning Curves of Kernel Ridge Regression under Power-law Decay.
> > >
> > > [2] Kernel interpolation generalizes poorly.

---

### Official Review · Reviewer_imqV · 2025-06-29

**Clarity:** 3
**Significance:** 3
**Originality:** 3
**Rating:** 4
**Confidence:** 3

**Summary:**

This paper analyzes kernel ridge regression under a structured non-i.i.d. setting where each latent signal generates multiple noisy observations.  A k-block model captures blockwise intra-dependencies with inter-block independence.  The authors derive excess risk bounds using a blockwise concentration approach.  The results describe how sampling and dependency affect generalization in denoising score learning.

**Questions:**

1. The theoretical contribution mainly extends known bias-variance analysis to a blockwise non-i.i.d. setting. Could the authors clarify what key steps or insights make this extension substantially novel compared to existing kernel generalization theory?

2. The paper’s findings on the optimal noise sampling schedule provide some guidance for diffusion models. Could the authors comment on how these results might inspire practical training or data usage strategies more broadly?

**Ethical Concerns:**

["NO or VERY MINOR ethics concerns only"]

**Final Justification:**

I appreciate the overall completeness and clarity of the analysis. The derivation is solid and well-aligned with existing literature. I maintain my overall score of 4.

**Limitations:**

Yes.

**Paper Formatting Concerns:**

No.

**Quality:**

3

**Strengths And Weaknesses:**

**Strengths:**

1. The paper provides an explicit excess risk bound for kernel ridge regression under a structured non-i.i.d. block setting, which clarifies how the causal data model and sampling scheme affect generalization.
2. The analysis connects directly to denoising diffusion probabilistic models, showing how the optimal noise sampling ratio should depend on the time-varying noise-to-signal ratio, which is practically relevant.
3. The paper develops a Bernstein-type concentration inequality tailored for block-dependent samples, offering a reusable tool for analyzing learning with dependent data.

**Weaknesses：**

1. The paper does not propose a new algorithm but focuses mainly on theoretical bounds, which is not a novel direction by itself given that excess bias-variance bounds for kernel methods have been widely studied.
2. The new Bernstein-type concentration inequality mainly builds on existing techniques, including the Intrinsic Dimension Lemma and standard $\phi(a) = e^a - 1$ tricks from related literature, which limits its originality.
3. In Assumption 1, the notation for sub-Gaussian variance proxies is not used consistently: $\sigma^2_\epsilon$, $\sigma^2_{\epsilon_{1,2}}$, $\sigma^2$, and $\sigma^2_G$ appear without clear distinction. Since this assumption underpins the main risk bound, clearer notation or explicit definitions for each variance term would improve clarity and rigor.
4. The paper states “Yes” for open access to data and code, but only synthetic data generation details are described; no actual code link or repository is provided, which limits full reproducibility.

---

> ### Author Rebuttal · Authors · 2025-07-31
>
> Thanks for your time and efforts reviewing our paper. We now address raised questions as follows.
>
> * Q1: The paper does not propose a new algorithm but focuses mainly on theoretical bounds, which is not a novel direction by itself given that excess bias-variance bounds for kernel methods have been widely studied.
>
>     Thanks for your comment. While excess risk bounds for kernel methods are well-studied in i.i.d. settings, no prior works have studied the excess risk and even learnability in the structured non-i.i.d. setting which is ubiquitous and important in applications like diffusion models. This paper contributes the analysis to the novel structured non-i.i.d. setting. Our theoretical bound explicitly characterizes the learnability in the structured non-i.i.d. setting and demonstrates how multiple noisy observations ($k$) affect generalization.
>
>     Beyond theoretical bounds, our theory also guides noise-sample pairing strategies when training diffusion models: for general $t$, using large $k$ when $t$ increases.
>
> * Q2: The new Bernstein-type concentration inequality mainly builds on existing techniques, including the Intrinsic Dimension Lemma and standard $\phi(a) = e^a - 1$ tricks from related literature, which limits its originality.
>
>     Thanks for your comments. The Intrinsic Dimension Lemma and standard $\phi(a) = e^a - 1$ tricks are used in the standard i.i.d. Bernstein inequality. However, for $k$-gap independent data which is non-i.i.d. in our setting, exisiting techniques cannot handle data dependence. Therefore, we introduce **a refined block decomposition**, yielding multiple partitions with several independent blocks respectively. The novel technique enables us to apply standard symmetrization techniques for each partition individually and resolve cross-partition dependencies through our **novel incorporating lemma** (Lemma C.20). Our technique can also extend to more general cases (e.g. varying $k$ per signal, approximately independent block gaps), serving as a new tool in dependent data analysis for machine learning community.
>
>
> * Q3: Could the authors clarify what key steps / insights make this extension substantially novel compared to existing kernel generalization theory?
>
>     Thanks for your question. Our extension advances kernel generalization theory through two key innovations. One key step lies in the development of **the new Bernstein inequality** tailored for KRR analysis in the structured non-i.i.d. setting. The other key step is providing **additional concentration analysis** for $V_3$ (line 547), where we utilize the truncation technique and the novel Bernstein inequality (Lemma B.9). Both the two steps are essential for excess risk analysis in dependent cases. / To existing kernel generalization theory, we provide novel insights for **whether data dependencies benefit or hinder the generalization performance of KRR**. Our theoretical bounds explicitly blends $\frac{1}{n}$ and $\frac{1}{nk}$, revealing a critical trade-off between relevance and noise sample size: when the correlation level $\tilde{r}$ is large, i.e., the signal dominates in the observed noisy data, increasing $k$ offers little benefits while increasing $k$ helps generalization when the noise component prevails.
>
> * Q4: Clearer notation or explicit definitions for each variance term.
>
>
>     Thanks for your good points. $\sigma_\epsilon$ and $\sigma^2$ control the decay of the population noise conditionally on data, with the former controlling the exponential decay (sub-Gaussian norm) and the latter controlling the quadratic decay (second moment). In parallel, for technical reasons to analyze dependent data, we extend $\sigma_\epsilon$ and $\sigma$ which are single-data conditioning to $\sigma_{\epsilon_{1,2}}$ and $\sigma_G^2$ which are dependent-data conditioning.
>     We provide explicit definitions as follows. For $(g,y)\in \mathcal{G}\times \mathcal{Y}$, for any dimension $r=1,\dots, d$, denote $\epsilon^{(r)}:=y^{(r)}-f_\rho^{(r)* }(g)$. Then $\sigma_\epsilon$ is the upper bound of the sub-Gaussian norm of $\epsilon^{(r)}$ conditionally on $g$ and $\sigma_{\epsilon_{1,2}}$ is the upper bound of the sub-Gaussian norm of $\epsilon^{(r)}$ conditionally on $g,g'$ where $g\neq g'\in \mathcal{G}$. To be specific, $$\Vert\epsilon^{(r)}|g\Vert _ {\psi_2}=\inf\left\\{t>0{:}\ \mathbb{E}\left[\exp\left(\left(\epsilon^{(r)}\right)^2/t^2\right)|g\right]\leq2\right\\}\leq\sigma_\epsilon,\ g,g'\in \mathcal{G} \ \text{almost everywhere}.$$ $$\left\Vert\epsilon^{(r)}|g,g'\right\Vert_{\psi_2}=\inf\left\\{t>0{:}\ \mathbb{E}\left[\exp\left(\left(\epsilon^{(r)}\right)^2/t^2\right)\big|g,g'\right]\leq2\right\\}\leq\sigma_{\epsilon_{1,2}}, \ g,g'\in \mathcal{G} \ \text{almost everywhere}.$$ Similarly, $\sigma^2$ and $\sigma_G^2$ are the upper bounds of the second moment of $\epsilon^{(r)}$ conditionally on $g$ and $g,g'$ respectively. $$\mathbb{E}\left[{\epsilon^{(r)}}^2|g\right]\leq\sigma^2,\ \mathbb{E}\left[{\epsilon^{(r)}}^2|g,g'\right]\leq \sigma_G^2, \ g,g' \in \mathcal{G} \ \text{almost everywhere}.$$
>
>
> * Q5: The paper states “Yes” for open access to data and code, but only synthetic data generation details are described; no actual code link or repository is provided, which limits full reproducibility.
>
>     Thanks for your comment. As we can not provide any code link in the rebuttal stage, we will provide the actual code link or repository in the revised manuscript.
>
> * Q6: The paper’s findings on the optimal noise sampling schedule provide some guidance for diffusion models. Could the authors comment on how these results might inspire practical training or data usage strategies more broadly?
>
>     Thanks for your question. The optimal noise sampling schedule provides inspirations for practical training of denoising score learning. Consider training with a **fixed batch size** $nk$, for varying $t$ (or equivalently, $\alpha_t$), practitioners can adaptively choose $k$ for optimizing the training efficiency: if signal dominates, then setting $k=1$ is enough; while when noise dominates, practitioners are encouraged to increase $k$ to $\Theta\left((1-\alpha_t^{p/2})/\alpha_t^{p/2}\right)$. This adaptive design for noise multiplicity $k$ may advance the empirical study for denoising score learning.
>
>     Additionally, our insights can also extend beyond denoising score learning to **flow-based diffusion model** (e.g. flow matching, rectified flow) and other noise schedules (e.g. Variance-Exploding (VE), Variance-Preserving (VP), sub-VP, Linear, EDM), unifying noise sampling principles across diffusion paradigms. Overall, our $k$-adaptation principle provides the first theoretical grounding for noise scheduling decisions - enabling systematic design rather than empirical guesswork.
>
>
> We hope above response can address your concern and we are open to discuss more if any question still hold.

---

> > ### Comment · Reviewer_imqV · 2025-08-05
> >
> > Thank you to the authors for their response. Lemma C.20 appears relatively straightforward. The authors utilize Lieb’s inequality—frequently used in Tropp’s work such as [R1]—to extend classical Bernstein-type concentration results under dependence structures (e.g., [R2]) to the matrix setting, as developed in [R3].
> >
> > While I find the technical challenge of this lemma moderate, given the maturity of the underlying tools, I appreciate the overall completeness and clarity of the analysis. The derivation is solid and well-aligned with existing literature. I maintain my overall score of 4.
> >
> > References:
> >
> > [R1] Tropp, J. A. An Introduction to Matrix Concentration Inequalities. Foundations and Trends® in Machine Learning, 8(1-2), 1–230, 2015.
> >
> > [R2] Merlevède, F., Peligrad, M., & Rio, E. Bernstein inequality and moderate deviations under strong mixing conditions. In High Dimensional Probability V: The Luminy Volume, Volume 5, pp. 273–293. Institute of Mathematical Statistics, 2009.
> >
> > [R3] Banna, M., Merlevède, F., & Youssef, P. Bernstein-type Inequality for a Class of Dependent Random Matrices. Random Matrices: Theory and Applications, 5(02), 1650006, 2016.

---

> > > ### Author Response · Authors · 2025-08-05
> > >
> > > Dear Reviewer imqV,
> > >
> > > We extend our thanks for your time, and we sincerely appreciate your positive feedback on our work. In particular, thanks for your appreciation to the overall completeness and clarity of the analysis.
> > >
> > > Thank you again for your effort in reviewing our work.
> > >
> > > Best,
> > >
> > > Authors

---

### Official Review · Reviewer_Vqf9 · 2025-07-05

**Clarity:** 3
**Significance:** 3
**Originality:** 3
**Rating:** 5
**Confidence:** 3

**Summary:**

The paper develops excess‐risk bounds for kernel ridge regression when the training data consist of multiple noisy observations of shared latent signals, a setting that breaches the usual i.i.d. assumption. A new blockwise decomposition technique yields a Bernstein‐type concentration inequality for k-gap independent sequences, leading to bounds that expose how the number of noises per signal k and a relevance parameter r jointly govern variance. The theory is specialized to denoising score learning in diffusion models, where it predicts an optimal k that grows as noise dominates, and small simulations corroborate the trend.

**Questions:**

Please clarify whether the blockwise concentration still holds if k varies per signal or if gaps are only approximately independent.

Could the relevance parameters be estimated from data to guide k adaptively in practice?

How sensitive are the bounds to the Hölder index p; can slower continuity be handled?

An ablation on real image diffusion training would strengthen claims; what computational overhead would varying k incur?

**Ethical Concerns:**

["NO or VERY MINOR ethics concerns only"]

**Final Justification:**

I thank the authors for their thorough rebuttal. My primary concern in my initial review was the limited empirical support, which made the practical impact of the theory speculative. The authors have now addressed this by providing new experiments on CIFAR-10, not only with an MLP but also with KRR.

While it is not possible to fully vet the code in the rebuttal phase, the specificity of the experimental setup and the coherence of the results give me confidence in their claims. The authors have strengthened the paper and demonstrated the relevance of their work. Based on these improvements, and with the expectation that the full details of these new experiments will be incorporated into the final paper, I will raise my score.

**Limitations:**

The authors include a brief limitations section but should discuss the restrictiveness of the equal‑k assumption and the challenge of estimating r and spectral decay in high dimensions.

**Quality:**

3

**Strengths And Weaknesses:**

Quality: proofs appear rigorous, with explicit assumptions and detailed appendices, and the novel concentration inequality is broadly useful. Clarity: notation is heavy but consistent; proofs are sketched in the main text with full details deferred, and key intuitions are conveyed. Significance: the work fills a gap by treating structured dependence that arises in modern generative modeling, giving actionable guidance for noise‑pairing schedules.
Originality: the blockwise Bernstein bound and its application to kernel regression under causal noise structure are new.

Main weaknesses are strong spectral and Hölder continuity assumptions, reliance on equal k across signals, no comparison with alternative dependent‑data analyses, and only toy simulations on a 1‑D mixture that limit empirical support. Practical impact on diffusion training remains speculative without large‑scale experiments.

---

> ### Author Rebuttal · Authors · 2025-07-31
>
> Thanks for your time and efforts reviewing our paper. We now address raised questions as follows.
>
> * Q1: Reliance on strong spectral assumption.
>
>     Thanks for your comment. We consider the **polynomial spectrum** to make our bound clearer and facilitates direct comparison with established results in the i.i.d. setting (consistent with prior works [1,2]), which can make the core theoretical insights more accessible. While the assumption simplifies presentation, our framework is readily extensible to general spectra on the technical level. The key terms (e.g. $tr(TT_\lambda^{-1})^p$, $\Vert f_\lambda^{(r)}\Vert_{[\mathcal{H}]^\gamma}^2$ and $\Vert f_\lambda^{(r)}-f_\rho^{*(r)}\Vert_{[\mathcal{H}]^\gamma}^2$ in Lemma C.3-C.5) can be expressed directly in terms of individual eigenvalues ($\lambda_1, \lambda_2, \dots$) rather than the decay rate $\beta$. For instance, the norm $\Vert f_\lambda\Vert_{[\mathcal{H}]^\gamma}^2$ is fundamentally given by a series (shown below) that depends on the full eigenvalue sequence: $\Vert f_\lambda\Vert_{[\mathcal{H}]^\gamma}^2\asymp\sum_{i=1}^\infty\left(\frac{\lambda_i^p}{\lambda_i+\lambda}\right)^2i^{-1}.$ Therefore, the polynomial decay is primarily a tool for deriving clearer, more interpretable bounds without fundamentally limiting the scope of our technical approach.
>
> * Q2: Reliance on the Hölder continuity assumption.
>
>     Thanks for your comment. The primary purpose of **the Hölder continuity assumption** is to eliminate the need for the often unrealistic sub-Gaussian design assumption in deriving our general bounds [1,2]. Technically, it is essential for establishing a uniform concentration bound via covering number estimates (Lemmas C.10 and C.11). Furthermore, this assumption allows us to derive concise estimates for the relevance parameter $r_T$ in specific scenarios, such as when conditional orthogonality holds. We note that Hölder continuity is naturally satisfied by important kernel classes like the Laplace kernel, Sobolev kernels, and Neural Tangent Kernels [1,2].
>
> * Q3: Please clarify whether the blockwise concentration still holds if k varies per signal or if gaps are only approximately independent.
>
>     Thanks for your question. The blockwise concentration remains valid **when the number of noisy realizations $k_i$ varies per signal $x_i$**. A simple method to handle is replacing $k$ with $k_\mathrm{max}:=\max_i k_i$. The key insight is that the entire training observations $G$ satisfies $k_{\mathrm max}$-gap independent (see definitions in line 146-148), preserving the concentration properties. **If gaps are only approximately independent**, e.g., some weakly dependent process assuming specific mixing property, our framework can also be readily extended to accomodate the dependency. The core technique is to incorporate the specific dependence structure (e.g., mixing coefficients) to modify the concentration bounds by adding terms quantifying block dependence. Established methods for handling such dependencies can be adapted from [3,4].
>
> * Q4: No comparison with alternative dependent‑data analyses.
>
>     Thanks for your comment. Current literature on dependent-data analyses rely heavily on mixing conditions, which characterize the correlation between random variables from certain distance in the sequence. Differently, we consider the $k$-gap independent data, where the random variables are categorized multiple groups with size $k$, the data point will be highly correlated/dependent with the data in the same group, but is independent of the data from other groups. Then we would like to explain the difference between our work and these related works in the following aspects:
>
>     * **Setting and Assumption**
>
>         While many works have derived tight concentration high-probability bounds for non-i.i.d. data under the geometrically regular mixing assumption [5,6] or the algebraically mixing assumption [7], which characterize the decay of correlation between random varaiables as the distance increases, our $k$-gap independent case **cannot be covered by these works**, as the correlation between data points will remain high as long as they are from the same group with size $k$. Therefore, our $k$-gap independent case requires a fundamantally different non-i.i.d. data analysis.
>
>     * **Analysis Intuition**
>
>         Most of the existing results [5-7] perform analysis under the intuition to control the approximation error between independent blocks and dependent blocks by mixing coefficients. That being said, the data dependency is treated as an bad effect in their analysis and will finally appear as an error term. However, in our setting, we do not treat the data dependency a bad effect but aim to discover some benefits for reducing the error, otherwise it would be intractable to prove the vanishing generalization error in our setting for general $k$. To this end, we perform refined study in decomposing blocks to eliminate direct error approximation.
>
>     * **Insight**
>
>         Our analysis demonstrating the benefit of data relevance stands in contrast to a long line of work on learning from dependent data, providing insights for understanding data dependence. We believe some of our techniques and methodologies can be potentially extended to a broader settings (e.g., the size of each group could vary).
>
>     On the technical level, prior work [5-7] use **single-layer decomposition** that requires constructing independent surrogate blocks and carefully controlling approximation errors. On the contrary, we introduce **a more refined decomposition**, yielding multiple partitions with several independent blocks respectively, which enables us to apply standard symmetrization techniques for each partition individually and resolve cross-partition dependencies through our **novel incorporating lemma** (Lemma C.20).
>
> * Q5: Could the relevance parameters be estimated from data to guide k adaptively in practice?
>
>     Thanks for your question. In the case when $\alpha_t \to 0$, i.e., the data $x_t$ becomes more noisy, the optimal $k$ can be selected as $k^*=\Theta\left(\frac{1-\alpha_t^{p/2}}{\alpha_t^{p/2}}\right)$. For general $t$, the suggestion from our theory is to use larger $k$ when $t$ increases. Nevertheless, it still requries a bit hyper-parameter tuning for different practical tasks.
>
>
> * Q6: How sensitive are the bounds to the Hölder index p; can slower continuity be handled?
>
>     Thanks for your question. Under the specific conditional orthogonality, our **general upper bound** is $R(\lambda) \leq \tilde{\Theta} _ {\mathbb{P}}\left({n}^{-\operatorname*{min}(s,2)\theta}\right) + \tilde{\sigma}^2 O _ \mathbb{P}^{\rm poly}\left(n^{\alpha_0\theta}\left(\frac{\tilde{r} _ 0^{p/2}\vee r_e }{n}+\frac{(1-\tilde{r} _ 0^{p/2}) \wedge (1-r_e)}{nk}\right)\right)$. Overall, $p$ affects both the tightness of our bounds and the benefit of multiple observations ($k$). To be specific, larger $p$ implies the domination of $\frac{1}{nk}$, showing the underlying benefit to generalization by increasing $k$. In contrast, smaller $p$ (**slow continuity**) implies the generalization error is up to $O_\mathbb{P}^{\mathrm{poly}}(n^{\alpha_0\theta}/n)$.
>
> * Q7: Only toy simulations on a 1‑D mixture that limit empirical support. Practical impact on diffusion training remains speculative without large‑scale experiments. An ablation on real image diffusion training would strengthen claims.
>
>     Thanks for your comment. To demonstrate the applicability of our method beyond toy examples, we conducted an ablation study on the CIFAR-10 dataset. We trained a diffusion model using a dataset of 1024 samples for 100 epochs with a batch size of 1024, optimized using Adam with a learning rate of 2e-3. The model architecture was a two-layer U-Net, and time conditioning was implemented by expanding the time variable `t` and concatenating it as an additional input channel to the image.
>
>     We report the diffusion loss on the test set with a size of 1024 at \( t = 1.0 \) and \( t = 0.1 \) across different values of \( k: number of noisy realizations per data\). Each configuration was evaluated over **100 parallel runs** to ensure robustness:
>
>     | **k**  | **Loss at t = 1.0** | **Loss at t = 0.1** |
>     |-------:|--------------------:|---------------------:|
>     | 1      | 0.00268199          | 0.09571435           |
>     | 16     | 0.00175084          | 0.09752702           |
>     | 32     | 0.00123094          | 0.10304873           |
>     | 64     | 0.00102835          | 0.12863263           |
>     | 128    | 0.00069417          | 0.18158743           |
>
>     As shown, increasing $k$ consistently improves performance at $t=1$, indicating better fitting of the complex score function. However, it also leads to degradation at $t=0.1$, consistent with our empirical findings on the MoG settings in the paper. Both our empirical findings on the MoG settings and CIFAR-10 align well with our theory that when $t$ is large (i.e., noise dominates), increasing $k$ is beneficial to generalization.
>
> * Q8: What computational overhead would varying k incur?
>
>     Thanks for your question. Varying $k$ will not incur any computational overhead, as we always conside a fixed batchsize in experiments. No matter how $k$ varies, the computation is the same as the case $k=1$.
>
> We hope above response can address your concern and we are open to discuss more if any question still hold.
>
> [1] On the asymptotic learning curves of kernel ridge regression under power-law decay.
>
> [2] Kernel interpolation generalizes poorly.
>
> [3] Bernstein-type inequality for a class of dependent random matrices.
>
> [4] Bernstein inequality and moderate deviations under strong mixing conditions.
>
> [5] Fast learning from non-iid observations.
>
> [6] Fast learning from $\alpha$-mixing observations.
>
> [7] Rates of convergence for empirical processes of stationary mixing sequences.

---

> > ### Comment · Reviewer_Vqf9 · 2025-08-04
> >
> > I thank the authors for their thorough rebuttal. My primary concern in my initial review was the limited empirical support, which made the practical impact of the theory speculative. The authors have now addressed this by providing new experiments on CIFAR-10, not only with an MLP but also with KRR.
> >
> > While it is not possible to fully vet the code in the rebuttal phase, the specificity of the experimental setup and the coherence of the results give me confidence in their claims. The authors have strengthened the paper and demonstrated the relevance of their work. Based on these improvements, and with the expectation that the full details of these new experiments will be incorporated into the final paper, I will raise my score.

---

> > > ### Author Response · Authors · 2025-08-05
> > >
> > > Dear Reviewer Vqf9,
> > >
> > > We are glad to hear that our rebuttal has addressed your concerns, and we sincerely appreciate your decision to raise the score. In particular, thank you for emphasizing the strength in providing new experiments and non-i.i.d. data analysis. We will incorporate the full details of those new experiments and non-i.i.d. data analysis discussion into the revised manuscript.
> > >
> > > Thank you again for your effort in reviewing our work.
> > >
> > > Best,
> > >
> > > Authors

---

### Decision · Program_Chairs · 2025-09-17

**Decision:**

Accept (poster)

**Comment:**

This paper investigates the excess risk of kernel ridge regression under a structured non-i.i.d. setting, where each latent signal generates multiple noisy observations. This setting goes beyond the classical i.i.d. assumption and is modeled via blockwise independence, which differs from mixing processes. The work has a natural application to diffusion models. The authors derive excess risk bounds using a blockwise concentration approach under standard assumptions (source and capacity conditions). However, it remains unclear why the additional Hölder-continuous kernel assumption is required.

After the rebuttal, all reviewers were satisfied and recommended acceptance. The authors are expected to address the reviewers’ comments in the final version. From my own high-level reading, the intuition is that the $k$ noisy observations per latent signal do not affect the bias term, while they can help reduce variance. This formulation shares similarities with pairwise learning, and there are relevant works in the kernel literature that should be discussed for completeness. Furthermore, the introduction to be rewritten would benefit from a clearer motivation, for instance emphasizing the connection to diffusion models, as highlighted by multiple reviewers.